# Inertial drag and lift forces for coarse grains on rough alluvial beds measured using in-grain accelerometers

Georgios Maniatis[1], Trevor Hoey[2], Rebecca Hodge[3], Dieter Rickenmann[4], and Alexandre Badoux[4]

[1]School of Environment and Technology, University of Brighton, UK
[2]Department of Civil and Environmental Engineering, Brunel University London, UK
[3]Department of Geography, Durham University, UK
[4]Swiss Federal Institute WSL

**Correspondence:** Georgios Maniatis (g.maniatis@brighton.ac.uk)

**Abstract.** Quantifying the force regime that controls the movement of a single grain during fluvial transport has historically proven to be difficult. Inertial Micro Mechanical and Electrical Sensors (MEMS) (sensor-assemblies that mainly comprise micro-accelerometers and gyroscopes) can used to address this problem using a "smart-pebble": a mobile Inertial Measurement Unit (IMU) enclosed in a stone-like assembly that can measure directly the forces on a particle during sediment transport.

Previous research has demonstrated that measurements using MEMS sensors can be used to calculate the dynamics of single grains over short time periods, despite limitations in the accuracy of the MEMS sensors that have been used to date. This paper develops a theoretical framework for calculating drag and lift forces on grains based on IMU measurements. IMUs were embedded a spherical and an ellipsoidal grain and used in flume experiments in which flow was increased until the grain moved. Acceleration measurements along three orthogonal directions were then processed to calculate the threshold force for

entrainment resulting in a statistical approximation of inertial impulse thresholds for both the lift and drag components of grain inertial dynamics. The ellipsoid IMU was also deployed in a series of experiments in a steep stream (Erlenbach, Switzerland). The inertial dynamics from both sets of experiments provide direct measurement of the resultant forces on sediment particles during transport which quantifies: a) the effect of grain shape; and b) the effect of varied intensity hydraulic forcing on the motion of coarse sediment grains during bed-load transport. Lift impulses exert a significant control on the motion of the

ellipsoid across hydraulic regimes, despite the occurrence of higher magnitude and duration drag impulses. The first order statistical generalisation of the results suggests that the kinetics of the ellipsoid are characterised by low or no mobility states and that the majority of mobility states are controlled by lift impulses.

## 1 Introduction

River sediment transport is a critical process in landscape evolution (Tucker and Hancock, 2010), controls river morphology

and ecology (Recking et al., 2015) and affects river engineering (Van Rijn, 1984). The study of the two-way relationship between transport processes and the corresponding morphology has a long history (e.g. Gilbert and Murphy, 1914), using approaches and mathematical conceptualisations that range from deterministic (e.g. Gilbert and Murphy, 1914; Shields, 1936; Ali and Dey, 2016) to probabilistic (e.g. Einstein, 1937; Grass, 1970; Ancey et al., 2008).

Fluvial sediment transport is a complex two-phase flow defined by: (a) hydraulics (Kline et al., 1967; Nelson et al., 1995; Papanicolaou et al., 2002); (b) sediment properties and arrangement (Ashida and Michiue, 1971; Komar and Li, 1988; Kirchner et al., 1990; Buffington et al., 1992; Hodge et al., 2013; Prancevic and Lamb, 2015); (c) flow history across time scales (Shvidchenko and Pender, 2000; Diplas et al., 2008; Valyrakis et al., 2010; Phillips et al., 2018; Masteller et al., 2019); and, (d) biological and chemical processes that can rearrange or stabilise sediment (Johnson et al., 2011; Vignaga et al., 2013; Johnson, 2016).

To analyse the motion of a grain resting on a riverbed that is sheared by a turbulent flow (Dey and Ali, 2018), a large group of laboratory and theoretical studies use an implicit (fixed) reference frame. Historically, such analyses have been deterministic (implementing a single threshold shear stress or force at which grains are entrained (Gilbert and Murphy, 1914; Shields, 1936; Yalin, 1963; Iwagaki, 1956; Ikeda, 1982; Dey, 1999). However, stochastic descriptions arguably capture better the complex particle-fluid interplay since near-bed turbulence which drives gain motion is inherently stochastic (Einstein, 1937; Grass, 1970; Papanicolaou et al., 2002; Marion and Tregnaghi, 2013). Coupled with advances in monitoring techniques (e.g. Papanicolaou et al., 2002; Fathel et al., 2016), stochastic treatments have led to Lagrangian, primarily numerical, formulations being applied to the full range of motion (McEwan et al., 2004; Bialik et al., 2015). The term Lagrangian means that sediment flow is observed from the perspective of individual mobile sediment grains and not a fixed time and space domain (as in traditional Eulerian approaches using fixed x,y,z co-ordinates, Ballio et al., 2018). The turbulence impulse approach (Diplas et al., 2008; Valyrakis et al., 2010; Celik et al., 2010) accounts for both the magnitude and the duration of the hydraulic forcing and is often categorised as a stochastic approach (Dey and Ali, 2018) since the stochastic nature of local turbulence is accounted for by the integration of turbulent forces acting on the grain over time. However, the grain forces are treated deterministically through a detailed treatment of the force balance during incipient motion. Finally, the spatio-temporal approach (Coleman and Nikora, 2008) is different as the equations of motion are applied separately for the fluid (in a spatially averaged domain) and the sediment particles, linking the mode of transport with the scales of turbulence (Bialik et al., 2015).

Field experiments tracing individual sediment grains are in principle Lagrangian (Hassan and Roy, 2016). Lagrangian analytical models have been developed following advances in monitoring techniques that allow tracking of individual grains, including magnetic (e.g. Schmidt and Ergenzinger, 1992; Hassan et al., 2009) and RFID tracers (e.g. Schneider et al., 2014; Tsakiris et al., 2015). An important milestone in the development of Lagrangian approaches for sediment transport was the introduction of Discrete Particle Modelling techniques in simulations (McEwan et al., 2001, 2004; Schmeeckle and Nelson, 2003) which opened up the prospect for upscaling the Lagrangian metrics.

Lagrangian measurements find direct application in coarse grain gravel bed and bedrock river environments (e.g. Hassan et al., 1992, 2009; Ferguson et al., 2002; Hodge et al., 2011; Liedermann et al., 2012) and the morphological impact of Lagrangian dynamics in those environments is pronounced (Hodge et al., 2011). For example, the inertia of the typically larger particles transported in these streams has been identified as one of the factors contributing to non-accurate predictions of transport rates (Buffington and Montgomery, 1997; Bunte et al., 2004; Singh et al., 2009). Equally important is the lack of information on the energy transfer between these large particles and the river bed, particularly during impact. Recent experiments (Gimbert et al., 2019) show how this energy transfer can be inferred from seismic measurements, opening the way for

testing hypotheses that relate to river reach scale processes (eg. Burtin et al., 2014). Finally, for a complete understanding of these interactions, the rotational component of grain movement cannot be ignored (Niño and García, 1998).

A particular advance in monitoring technology has been the development of sediment grain - scale inertial sensors which record at high frequency the accelerations and angular velocities experienced by grains during entrainment and motion (Kularatna et al., 2006; Akeila et al., 2010; Frank et al., 2015; Maniatis et al., 2013; Gronz et al., 2016; Maniatis et al., 2017). These applications became possible after the development of compact MEMS (Micro Electrical Mechanical Sensors) Inertial Measurement Units (IMUs), assemblies of 3D MEMS accelerometers and 3D MEMS gyroscopes, which overcome many technical difficulties posed by older instrumentation (mainly caused by limited storage capacity, e.g. Ergenzinger and Jupner, 1992; Spazzapan et al., 2004). The focus of those works is the development of an IMU based sensor assembly (IMU enclosed in a grain or a purpose specific grain shaped artificial enclosure) that can successfully measure grain dynamics.

MEMS-IMU sensors measure the acceleration and angular velocity of the grain, which can be used to calculate the net force acting on the grain. For the most accurate measurement of this force, sensors should be located at the grain's centre of mass. Data collected from within grains undergoing transport has potential to describe the timing of motion, forces acting on the grain and grain location. As a grain moves, its centre of mass moves and so the reference point for the force measurements is mobile. The latter means that the IMU measurements need to be transformed to a frame of reference that can be understood by an observer. Generally, an IMU accelerometer is a non-inertial frame fixed within the mobile body frame of the sensor assembly.

In theory, the accelerations recorded by the IMU could be integrated to calculate grain velocity and integrated again to reveal location, a process referred to as dead-reckoning. One long-term goal for this approach would be to use IMUs to track a large number of grains through fluvial systems. However, real fixed ('strap-down') IMUs based on MEMS are not suitable for these integrations since the data contain several sources of uncertainty including signal noise and nano-scale mis-alignment of sensor axes. With sensors that are cheap enough to be deployed in large numbers, the accumulation of errors means that they cannot be used for 3D tracking of long term unconstrained motions (Woodman, 2007; Kok et al., 2017; VectorNav, 2016). This problem is well known in the fields of navigation and electrical engineering and the modelling of IMU errors is a significant research area (Zekavat and Buehrer, 2011) since the applications of this technology are numerous (Gebre-Egziabher et al., 1998; Grewal et al., 2007).

Despite the limitations, IMU sensors have been considered to be a suitable technique for measuring grain motion (e.g. Gronz et al., 2016). Recently Gimbert et al. (2019) used accelerometers to measure particle bed impacts in order to complement seismic measurements. However, for field deployments, relevant sensors have only been used as start and stop motion sensors (Olinde and Johnson, 2015).

The first goal of this paper is to introduce a simple rigid body model that connects measurements derived from an idealised IMU with existing models for grain motion. For this model to be successful, it is necessary to map the IMU body frame dynamics to the reference frame of motion (flume or riverbed). This resolution allows the inertial measurements to be related to the forcing on the particle and defines an explicit and unambiguous threshold for particle motion. The second goal is to introduce the calculation of inertial impulses above the drag and lift thresholds of motion, following the example of Diplas

et al. (2008), for grain entrainments and short transport events. The calculations are performed for observations made in a set of
flume entrainment experiments using two sensor assemblies: one spherical and one ellipsoidal. We apply the same analysis to
a time series of successive transport events measured with the ellipsoid sensor in a steep Alpine river (Erlenbach, Switzerland),
calculating the force regime and the generated impulses during grain motion. Finally, we discuss how the combined dataset
(flume and field experiments) for the ellipsoid sensor can be used for bootstrap calculations leading to the generalisation of the
derived measurements.

## 2  Force measurements and a Newton - Euler regime for sediment motion

An IMU accelerometer records accelerations of the grain, which can be converted to forces acting on the grain by multiplying
by grain mass. However, those accelerations are both fixed (e.g due to gravity) and variable (e.g due to fluid forces applied
on the particle). The measurements are difficult to interpret because fixed and variable accelerations cannot be decoupled
after the accelerometer begins to move and these accelerations are perceived differently from different reference frames. If the
sensor is static and gravity is recorded from the reference frame of the sensor it is then possible to derive a reliable orientation
measurement. If the sensor moves and the sensor frame accelerations are compensated for gravity (by removing gravity from
the raw accelerometer measurement) then the sensor will record the 3D components of the resultant or net force that mobilises
the particle. This resultant is the force that can be observed from an observer who is static in relation to the particle. In Appendix
A, we provide a technical explanation of the differences between non-inertial (mobile sensor frame) and inertial (observed from
an external to the sensor fixed frame) accelerations and demonstrate the necessary transformations.

The physics of IMU sensors define also the main difference between IMUs and other force sensors that have been used to
monitor grain motion, such as load cells (e.g. Schmeeckle et al., 2007 and Lamb et al., 2017). Load cells measure the total
vertical and horizontal forces applied to the grain prior to the onset of motion, but their application is also limited because
they prevent the grain from being fully entrained. In the the case of a grain sitting on a horizontal bed and being subjected to
an increasing lift force, a load cell would record an increasing force up until the point at which the lift force was equal to the
weight and the grain was entrained. In contrast, the gravity compensated IMU will record zero acceleration (and therefore zero
net force) until the point of entrainment after which the grain starts to accelerate. In flowing water, grains do not transition
instantly from being completely stationary to being fully entrained (Garcia et al., 2007), and the IMU will also record the
movement of grain vibrations prior to entrainment. Consequently, load cells and IMUs measure different parts of the transport
process; load cells record the forces that are applied to the grain prior to entrainment, i.e. to the point where the forces balance
the weight of the grain, whereas IMUs measure the forces that accelerate the grain from this balanced position.

The following derivations rely on the transformation of the IMU accelerations from the mobile reference frame of the
monitored particle (frame $b$) to a local static frame where the applied hydraulic, particle and gravitational forces are analysed
(frame $r$ hereafter). The $x$ and $y$ directions of the $r$ frame ($r_x$ and $r_y$ hereafter) are, respectively, parallel and perpendicular to
the mean direction of the flow and the river or flume bed. The $z$ direction ($r_z$ hereafter) is normal to the flow and the riverbed. In
addition, an independent static frame $i$ is used, the origin of which coincides with the centre of the Earth, as a stable reference

for the initial alignment of frames $b$ and $r$ (Appendix A). $b$ - $r$ transformations rely on the successive tracking of the relative orientation between the two frames which are represented here using quaternions (Appendices A and G1).

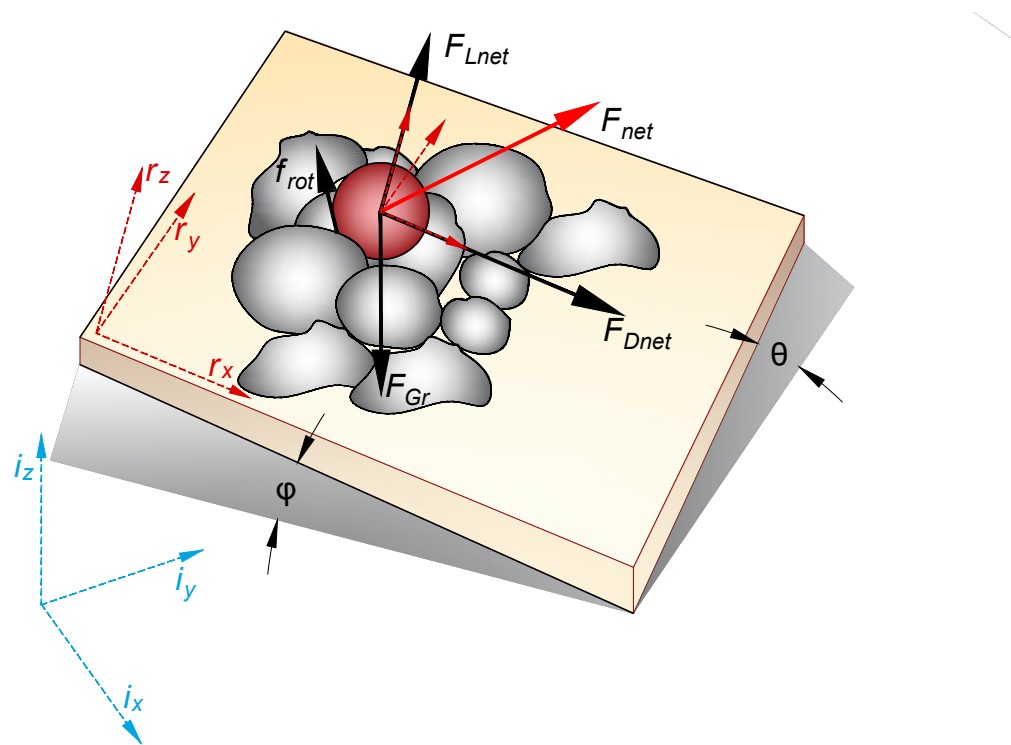

**Figure 1. Forces in the Newton-Euler model.** The diagram shows the linear forces applied on the centre of the mass of the target particle. $F_{Dnet}$ is the net drag force, $F_{Lnet}$ is the net lift force and $F_G$ is the force of gravity (equation 1) all analysed in the local static frame $r$. The rotational component (torques generated by the surface force $f_{rot}$ around the centre of the mass of the target particle) is analysed on the body frame of the particle ($b$, here depicted as aligned with the static frame $r$, the general case is discussed in Appendix A). $\phi$ is the downstream slope. For the results presented in this work $r_x$ is the downstream direction. Also, for the flume experiments there is no cross-stream slope ($\theta = 0$)

    Combined measurements of grain acceleration and angular velocity allow direct calculation of the forces and turning mo-
ments acting at the grain's centre of mass. This type of model formulation is the Newton-Euler model in the rigid body

dynamics literature (O'Reilly, 2008). For a spherical particle resting on an inclined bed, irrespective of the degree of exposure to the flow, the Newton-Euler regime is defined as follows:

$$\boldsymbol{F_{net}} = \boldsymbol{f} + \boldsymbol{F_{Gr}} = m \begin{bmatrix} a_{rx} \\ a_{ry} \\ a_{rz} \end{bmatrix} \tag{1}$$

$$\boldsymbol{T_{net}} = \boldsymbol{f_{rot}} \, R = I_{cm} \begin{bmatrix} \alpha_{bx} \\ \alpha_{by} \\ \alpha_{bz} \end{bmatrix} \tag{2}$$

$\boldsymbol{F_{net}}$ and $\boldsymbol{T_{net}}$ are net forces and torques applied on and around the centre of mass of the particle. For the right hand elements of equations 1 and 2, $m$ is the particle mass and $I_{cm}$ is its moment of inertia (Appendix C). $a_{rx}$, $a_{ry}$, $a_{rz}$ are linear accelerations resolved in the static frame $r$ (Figure A1) measured using accelerometer measurements and equation A4. $\alpha_{bx}$, $\alpha_{by}$, $\alpha_{bz}$ are rotational accelerations resolved in the mobile but fixed on the particle frame $b$ such that:

$$\begin{bmatrix} \alpha_{bx} & \alpha_{by} & \alpha_{bz} \end{bmatrix}^T = \begin{bmatrix} \dfrac{d\boldsymbol{\omega_{bx}}}{dt} & \dfrac{d\boldsymbol{\omega_{by}}}{dt} & \dfrac{d\boldsymbol{\omega_{bz}}}{dt} \end{bmatrix}^T \tag{3}$$

where $\omega_{bx}, \omega_{by}$ and $\omega_{bz}$ are angular velocities within the body frame reference frame as recorded by gyroscope measurements.

The left hand elements of equations 1 and 2 describe the forces applied on the particle. Hereafter, the lower case vectors and scalars (e.g. terms $\boldsymbol{f}$ and $\boldsymbol{f_{rot}}$) will refer to the interactions between hydraulic forces (turbulence) and particle forces (support forces and friction) that are not measured directly by an IMU and which cannot be decoupled using IMU measurements. The

vector $\boldsymbol{f}$ is the linear component of those interactions, applied on the centre of mass of the particle. The vector $\boldsymbol{f_{rot}}$ is the component of those interactions applied on the surface of the particle and it generates the torques around the centre of its mass (tangential to the particle's radius). The vector $\boldsymbol{F_{Gr}}$ defines the force of gravity rotated in the $r$ frame and compensated for hydrodynamic effects (e.g buoyancy) as:

$$\boldsymbol{F_{Gr}} = R(_i^r q) \boldsymbol{W_{si}} f_v = \begin{bmatrix} F_{Grx} \\ F_{Gry} \\ F_{Grz} \end{bmatrix} \tag{4}$$

The matrix $R(_i^r q)$ denotes the orientation of the slope in 3D (the relative orientations between the frames $i$ and $r$, equation A5). $W_{si}$ is the immersed weight of the spherical particle equal to $m_b g$, where $m_b$ is its immersed mass (Papanicolaou et al., 2002) and $g$ is the acceleration of gravity, both acting at its centre of mass. $\boldsymbol{W_{si}}$ has a constant direction in frame $i$ and is rotated in $r$ using $R(_i^r q)$. $f_v = 1 + [0.5\rho/(\rho_s - \rho)]$ accounts for the hydrodynamic mass effect (Papanicolaou et al., 2002;

Celik et al., 2010) and $\rho$, $\rho_s$ are the densities of the water and the particle, respectively. For our ellipsoid, we calculate $F_{Gr}$ assuming a sphere with the same volume as the ellipsoid, since resolving the hydrodynamic mass coefficient for an ellipsoid in 3D is beyond the scope of this work. The magnitude of $\boldsymbol{F_{Gr}}$ can be measured by deploying an IMU as an orientation sensor (Appendix A1).

The magnitude of $\boldsymbol{F_{Gr}}$ components along the drag direction ($r_x$) is given by the vector:

$$FG_D = F_{Grx} \tag{5}$$

For the direction normal to the bed ($r_z$, direction of lift hydraulic force) the $\boldsymbol{F_{Gr}}$ component is the vector:

$$FG_L = F_{Grz} \tag{6}$$

Similarly, we can use the linear accelerations of equation 1 to separate the components of the net force ($F_{net}$) along the drag and lift directions as:

$$F_{Dnet} = m\, a_{rx} \tag{7}$$

and

$$F_{Lnet} = m\, a_{rz} \tag{8}$$

## 2.1 Inertial measurements and the threshold of motion

The linear force threshold of motion is defined for $a_r = [0\ \ 0\ \ 0]^T$, representing the explicit state where the forces are balanced (a resultant force $\boldsymbol{F_{net}}$ of equation 1 equal to 0) and the particle is not moving. Since all the linear forces are transformed in the $r$ frame, exceeding this threshold relates to either a motion along the $r_x$ drag direction (threshold defined at $\boldsymbol{F_{Dnet}} = 0$) and/or a motion along the $r_z$ lift direction (threshold defined at $\boldsymbol{F_{Lnet}} = 0$). Given the force balance of equation 1, the threshold of motion is also the point where the resultant of the combined forces represented by $\boldsymbol{f}$ (which includes the hydraulic lift and drag forces) balance the rotated gravity forces. For the drag direction, the component of $\boldsymbol{f}$ is given by the vector:

$$\boldsymbol{f_D} = \boldsymbol{F_{Dnet}} - \boldsymbol{FG_D} \tag{9}$$

and the threshold of motion is at the point where $\boldsymbol{f_D} = -\boldsymbol{FG_D}$

If $\boldsymbol{F_{Dnet}}$ exceeds 0 then $\boldsymbol{f_D}$ exceeds the component of gravity $\boldsymbol{FG_D}$ towards the positive (+) drag direction (downstream). Consequently, the condition $\boldsymbol{F_{Dnet}} > 0$ is the exact equivalent of the condition $\boldsymbol{f_D} > -\boldsymbol{FG_D}$. We use the second description (for both the drag and the lift components) in order to highlight the fact that the $\boldsymbol{f}$ interaction primarily balances the components of gravity defined by the mean orientation of the bed.

For the lift direction, the component of $\boldsymbol{f}$ is given by:

$$\boldsymbol{f_L} = \boldsymbol{F_{Lnet}} - \boldsymbol{FG_L} \tag{10}$$

Similarly to the drag components, the condition $\boldsymbol{F_{Lnet}} > 0$ is the exact equivalent of the condition $\boldsymbol{f_L} > -\boldsymbol{FG_L}$ (the particle moving towards the $r_z$ positive direction, upwards).

The rotational threshold of motion is defined by the state where the balance of torques around the centre of mass of the particle are balanced ($\boldsymbol{T_{net}} = 0$). In the Newton-Euler model introduced here, the sum of torques captures a spinning rotational component defined by the product of the moment of inertia of the rotating particle ($I_{cm}$) and the body frame angular accelerations (equations 2 and 3). It is important to note that this rotation differs from the *orbital* rotation defined around the centre of mass of a supporting particle which is a common description in the hydraulic literature (e.g. Papanicolaou et al., 2002).

The torques are analysed in the body frame of the particle ($b$) and a non-directional description of the rotation threshold is given by the norm of angular accelerations ($\|\boldsymbol{\alpha_b}\|$) exceeding 0. After equations 2 and 3 the critical condition for particle rotation is given by:

$$T_{net} = I_{cm}\|\boldsymbol{\alpha_b}\| = I_{cm}\|\frac{d\boldsymbol{\omega_b}}{dt}\| = f_{rot}\, R \geq 0 \tag{11}$$

It is useful to re-state here that all the derivations so far are for a spherical particle (the equivalent of equation 11 for an ellipsoid is given in Appendix C). Using equation 11, it is possible to estimate the magnitude of the tangential component $f_{rot}$ (Appendix C). The calculation reveals that, for the scale of the particles discussed here, the effect of the tangential force, in terms of force magnitude applied on the particle, is negligible in comparison to the linear forces. This is explained by the dependency of the sum of torques ($\boldsymbol{T_{net}}$) on the moment of inertia $I_{cm}$ of the particle which is generally a very small number even for relatively coarse particles (e.g. for the spherical particle introduced in section 3 it is equal to $0.00085$ kg.m$^2$). For this reason, we will only focus on the linear net force and the interaction $\boldsymbol{f}$ which we calculate as the difference between the net force and the gravitational components. However, we demonstrate the scale difference between the rotational and the linear components of particle motion in Figure 3 and Appendix C.

## 2.2 General kinematics and impulse

For completeness we note that if $\boldsymbol{a_r} = [a_{rx}\, a_{ry}\, a_{rz}]^T$ is the 3D vector of linear acceleration of equation 1 in the $r$ reference frame, grain linear velocities can be calculated by single integration. The velocity in the $r$ frame is $v_r(t) = v_r(0) + \int_0^t a_r(t)\, dt$) and the total kinetic energy can by calculated from these velocities as $K = \frac{1}{2}m\|v_r\|^2 + \frac{1}{2}I_{cm}\|\omega_b\|^2$ with $\frac{1}{2}m\|v_r\|^2$ being the translational and $\frac{1}{2}I_{cm}\|\omega_b\|^2$ the rotational component. The condition $K = 0$ represents the threshold of rolling for any rigid body. However, this is not a direct equivalent to the typical rolling mode of entrainment because of the differences in the definition of rotation (spinning vs orbital) and the fact that here there is no assumption about the slipping condition as $K$ can be used to describe a rolling with slipping motion.

To account for both the duration and the magnitude of a force, the impulse $I$ for duration $\delta t$ starting from the time $t_i$ is defined as:

$$I = \int_{t_i}^{t_i+\delta t} F(t)dt \tag{12}$$

The subsequent analysis focuses on the calculation of impulses for specific time durations $\delta t$ (Diplas et al., 2008; Celik et al., 2010; Valyrakis et al., 2010). We calculate the impulse of the resultant interaction $\boldsymbol{f}$ above the threshold of motion and towards the positive drag and lift direction. This happens when the balancing gravity components $\boldsymbol{F_{Gr}}$ (equations 5 and 6) are exceeded, thus for the conditions :

$$f_D > -FG_D \tag{13}$$

or

$$f_L > -FG_L \tag{14}$$

which, as explained above, can only be satisfied when the magnitude of the net force exceeds 0 ($F_{net} > 0$, net force measured using gravity compensated accelerometer data, equation A4). In practice, we only account for the net forces that exceed the noise threshold of the accelerometer sensor (Figure 3, YEI, 2014; Maniatis, 2016). Finally, it is important to note that the calculated impulses are transferred to the particle from fluid turbulence and coherent flow structures, however this transfer is not described in this work. Here, the impulses capture directly the flow-particle interaction. The terms impulse, net impulse and inertial impulse are used interchangeably hereafter and they relate to impulses of the non zero net force.

## 3 Laboratory and field experiments

Two sensor assemblies were deployed, one sphere and one ellipsoid (described in Maniatis, 2016). The 90 mm diameter, 1.019 kg, sphere is solid aluminium with a symmetrical cavity for the IMU centred at the origin of the sphere. The ellipsoid (axes 100, 70, 30mm), made of the same material, weighs 0.942 kg. The cavity in the ellipsoid was designed to ensure that the IMU axes align with the principal axes of the whole device. The density of both devices after the cavity cut is $2670 \pm 3$ kg.m$^{-3}$ approximating the density of quartz (2650 kg.m$^{-3}$). The measuring unit is the TSS-DL-HH-S sensor from YEI-Technologies$^{\text{TM}}$ (YEI, 2014), equipped with a gyroscope ($\pm 2000°$.s$^{-1}$ sensitivity) and an accelerometer with a maximum range of $\pm 400$ g. The acceleration range is one of the main reasons for the selection of this IMU as lower range accelerometers, particularly those in the very common $\pm 20$ g range, are not suitable for capturing the forcing in natural environments (Maniatis et al., 2013). The factory maximum sampling frequency is 250 Hz. The nominal sampling frequency of the sensor (used for all the flume and field experiments presented in this work) is 50 Hz which permits constant use for approximately 5 hours (LiPo rechargeable battery). This frequency was chosen after considering displacements of 15 particle diameters per second reported by Drake et al. (1988). This is adequate for capturing the dynamics of the particles if collisions and strong interactions with the bed are excluded (different type and higher frequency piezoelectric sensors are more suitable for the full measurement of impacts). The measuring unit was calibrated through a series of shaking table and and rolling drop experiments which are described in Maniatis (2016, Chapter 6), along with the corresponding filtering workflow. During this calibration the noise threshold for the accelerometer was defined at $1.1 \pm 0.23$ N and the value of 1.2 N is used for the following presentation. The gyroscope has a much lower noise threshold which was calculated to be $< 0.0001$ °.s$^{-1}$

## 3.1 Laboratory entrainment experiments and separation of impulsive events

All flume experiments took place in an Armfield$^{TM}$ flume of 7 m effective length and 0.9 m internal width and at a fixed bed slope of $S = 0.02$. This slope corresponds to an orientation by the quaternion $^r_i q = [0.92, 0.14, -20, 028]$ (Appendix A). $^r_i q$ was measured by aligning the X-IMU axis with the centreline of the flume, the Y-IMU axis with the transverse direction and the Z-IMU axis with the direction normal to the bed. The positive X-IMU axis coincides with cross-section averaged flow direction. A bed of plastic hemispheres of the same diameter (90 mm) as the spherical device was constructed. The hemispheres were glued to form a 0.5 m (L) x 0.9 m (W) section and the whole section was placed at the point which allowed for the test particle to be at 4.5 m from the upstream boundary of the flume. A thin layer of 1.5 mm uniform sand was glued to both the hemispheres section and the upstream flume surface. The section upstream of the hemispheres was filled with very densely packed non-uniform, rounded gravel ($D_{50} = 0.015$ m) enabling the development of turbulent flow. No sediment transport occurred from the upstream gravel section during the entrainment experiments. The hemispheres were glued in positions that produced a 0.045 m protrusion of the sphere above the top of the hemispheres, although the sand layer made the protrusion higher, by partially filling the gaps between the hemispheres, and non uniform around the test particle ($\approx 0.050$ m). The ellipsoid was only supported by the hemispheres and was fully exposed to flow, thus having a protrusion equal to its c-axis ($\approx 0.03$, Figure 2).

For the experiments, the spherical device was placed on the flume centreline in a saddle position between four bed hemi-spheres and the three sensor axes were aligned with the inertial frame $i$ ($^b_i q = [1,0,0,0]$, Appendix B). After positioning the sensor, the discharge was increased at a constant rate of $0.028$ l.s$^{-2}$ until the particle was entrained. Acceleration and rotation were measured for the duration of the flow increase, throughout the sensor movement, and for 10 further seconds after it stopped moving. All the experiments were videoed at 60 fps using a standard GoPro Hero 7. The same experimental protocol was followed for the ellipsoid device, differing only in that the particle was initially aligned to the frame of reference of the flume bed ($^b_i q = {}^r_i q = [0.92, 0.14, -20, 028]$). This resulted in the X - sensor long axis coinciding with the X flume direction (the direction of the flow, Figure 2).

Ten entrainment experiments were conducted with each device. We define entrainment as when the particle moves by one particle diameter, or b-axis length for the ellipsoid. Having identified the time when the grain has moved by this distance from the video, the timing of the vibrations which directly and continuously preceded entrainment was also determined from the video. Many periods of vibration which do not lead to entrainment were recorded by the sensor and are visible on the videos. Drag and lift forces as well as the duration and the inertial impulses for cases where the grain started to move ($f_D > -FG_D$ and/or $f_L > -FG_L$) were calculated using the derivations of section 2.1. For the spherical sensor, the gravitational components are $FG_D = 3.99$ N and $FG_L = -7.25$ N, respectively (Figure 3). The equivalent values for the ellipsoid, using the geometry of a sphere of equal volume, were $FG_D = 5.11$ N and $FG_L = -9.28$ N. The critical discharge for the sphere was $24.8 \pm 1.8$ l.s$^{-1}$ which corresponds to a measured depth of $0.095 \pm 0.015$ m, measured from the top of the supporting hemispheres. The critical discharge for the ellipsoid was $45.2 \pm 2.2$l.s$^{-1}$ which corresponds to a measured depth of $0.12 \pm 0.02$ m ($\tau_* = 0.01$ and 0.02 respectively, $\tau^* = \frac{H S}{(\rho_p - \rho_f) D}$, Appendix D). The hydraulic parameters at mean critical discharges

are calculated in Table D1. The flume experiments have particle diameter to flow depth ratio close to 1 ($d/H \approx 1$) despite the fact that the particles were fully submerged at the critical discharge for all the experiments (Figure 2). The tested conditions are relevant to the entrainment of coarse particles in steep mountain streams but they should not be directly generalised to other bedload transport regimes despite the generality of the Newton-Euler model presented in section 2.

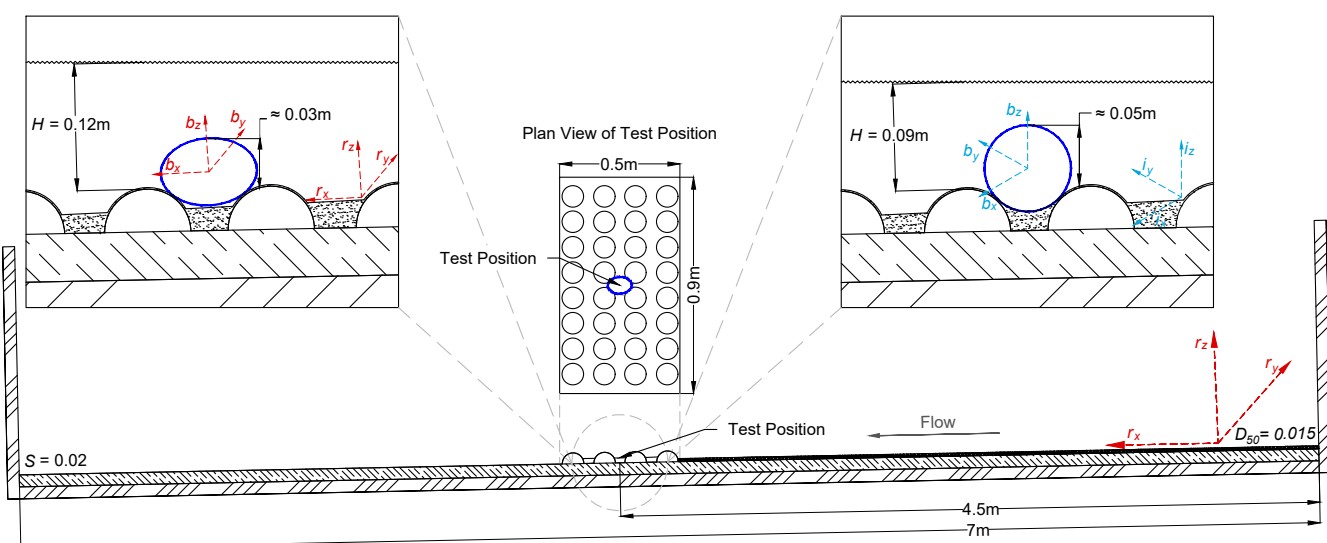

**Figure 2. Laboratory setting and initial alignment** The diagram shows the arrangement of the bed of hemispheres and the test position (4.5 m downstream the entrance of the flume). Upstream of the hemispheres bed, the flume was filled with densely packed gravel ($D_{50}$ = 0.015 m, black layer upstream of the hemisphere section in the graph) to enable the development of the flow. The sphere diameter is 0.09 m, and ellipsoid b-axis is 0.07 m. The height of centroid above the crest of the glued hemispheres is approximately 0.02 m for the sphere and 0.015 m for the ellipsoid. The depths correspond to the hydraulics at the point of entrainment, summarised in Table D1. $r$ stands for river-bed (flume in this case) reference frame, $b$ for body frame and $i$ for the inertial reference frame ($_i^r q$ = [0.92, 0.14, -20, 028]), Appendix A). $S$ is the slope of the channel (0.02). The sketch also depicts the initial alignment for each device ($b - i$ for the sphere and $b - r$ for the ellipsoid). While the spherical particle was in full contact with the bed (hemispheres and coating/filling gravel) the setting could result to a film of water flowing underneath the ellipsoid.

## 3.2 Probabilistic impulse threshold for motion

Entrainment was observed independently from video recordings which were synchronised with the experiments from the start of the flow increase (section 3.1). These observations are used to calculate statisticallty the probability of entrainment as a function of the impulse of the interaction $f$ exceeding gravity. Following the framework presented in Maniatis et al. (2017), the exact time of entrainment was noted in the video recording and the derived inertial impulses were separated into a binary, pre- and post-entrainment, data set. Logistic regression was used to describe the probability of entrainment, with $Pr > 0.5$ defining the threshold of motion. Following the conceptualisation of Grass (1970), exceeding that threshold relates to impulses that are able to fully dislodge the particle, in contrast to the conditions that relate to pre-entrainment vibrations. The difference here is that this conceptualisation is applied to events in which impulses exceed the thresholds defined in equations 5 and 6). Video recording was not possible in the field setting, so this calculation is only presented for the laboratory experiments.

## 3.3 Field testing

Field experiments took place within a 5 m long straight and confined reach of the Erlenbach mountain stream in Switzerland, approximately 15 m upstream of the concrete channel section and 55 m upstream of the sediment retention basin in which continuous bedload transport measurements have been made during the past 30 years (Turowski et al., 2011; Rickenmann et al., 2012). The stream has a step-pool morphology allowing the sensor to be retrieved from pools, so the ellipsoid sensor was submerged on a bare bedrock section close to the edge of a step (average slope $S = 0.1$, cross-averaged flow depth $H = 0.1$ m, $\tau_* = 0.095$), aligned to the same orientation as the riverbed ($_i^b q = {_i^r q} = [0.50\ -0.39\ 0.34\ -0.68$, assumed parallel to the banks and the cross-averaged flow direction, at the approximate centre line of the stream ) and allowed to be transported until it stopped moving and remained immobile for at least 10 seconds (Figure 3c). In all the experiments the sensor was entrained fully and immediately as there was no vibration *in situ*. The first one second of each transport event was removed from the data as the effect of holding and releasing the sensor were still present. Ten transport events were recorded and processed similarly to the flume experiments. The corresponding $f_D > -FG_D$ and $f_L > -FG_L$ (Figure 3c) were calculated similarly to the flume experiments. The average travel distance for each transport event was 2 $\pm 0.43$ m (from the point of release to the point of deposition, tape measurements) and the average event duration, after the first second of release, was 3 $\pm 0.6$ seconds (Figure 3). To establish a representative orientation for the reach in relation to the orientation of gravity, the IMU was aligned parallel to the approximate centreline (X-IMU axis, $X^+$ = cross-section averaged flow direction), trasverse (Y-IMU axis) and normal to the bed (Z-IMU axis) directions within the stream. The hydraulic parameters are summarised in Table D1.

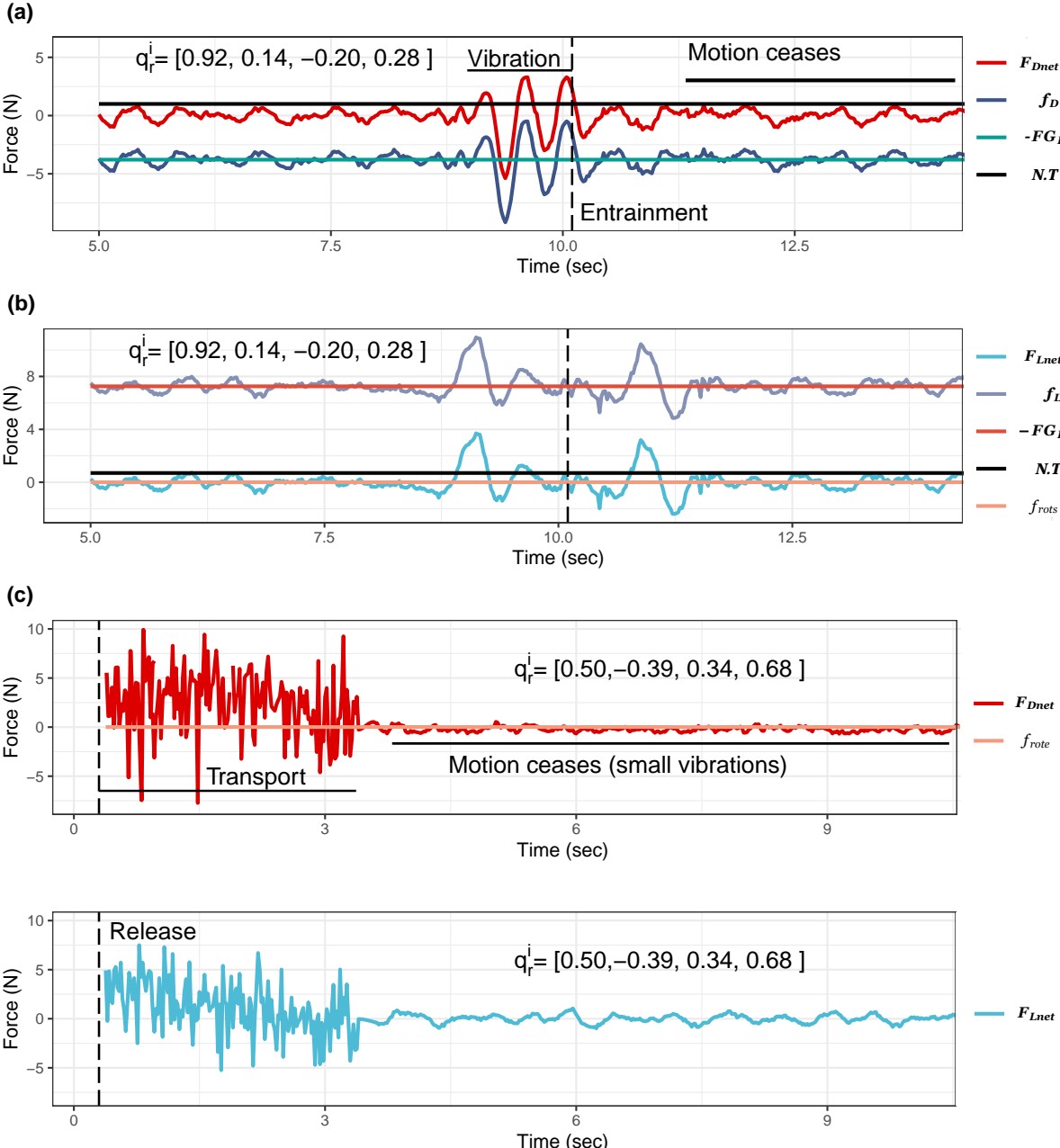

**Figure 3. Example flume and field experiments (a, b)** Calculated drag (a) and lift forces (b) for the same flume experiment using the spherical sensor. The vectors $\boldsymbol{F_{Dnet}}$ and $\boldsymbol{f_D}$ record the magnitudes of the net force and the resultant hydraulics- resistance interaction on the 2D-plane parallel to the flume bed. The vectors $\boldsymbol{F_{Lnet}}$ and $\boldsymbol{f_L}$ record the net force and the hydraulic resistance interaction, respectively, along the normal to the flume bed direction. The gravitational components ($\boldsymbol{FG_D}$ = 3.7 N and $\boldsymbol{FG_L}$ = -7.25 N) were calculated using equation 4. The vertical dashed line (t=10.1 s), shows the exact point of the entrainment of the particle as determined from the video recording. The $N.T$ lines indicates the noise threshold of the accelerometer sensor, the calculated impulses concern sequences of point forces that exceed $N.T$ **(c)** Net drag forces recorded during one field experiment (Erlenbach). The vertical line indicates the time of one second after the release of the sensor (see section 3.3). The calculation of the tangential force magnitude ($f_{rot_s}$ and $f_{rot_e}$ for the sphere and the ellipsoid, respectively) is based on equations C1 and C5.

## 4 Results

The flume experiments demonstrate the differences between the spherical and the ellipsoid particle during incipient motion (Figures 4 and 5). For the sphere, drag and lift impulses over the gravity forces ($\boldsymbol{f_D} > -\boldsymbol{FG_D}$ and $\boldsymbol{f_L} > -\boldsymbol{FG_L}$, equations 13 and 14) occur for similar durations and generate impulses of similar magnitude ($\boldsymbol{f_D} > -\boldsymbol{FG_D}$ impulses median = 0.45 N.s, $\boldsymbol{f_L} > -\boldsymbol{FG_L}$ impulses median= 0.46 N.s). The relationship between the duration of exceedance events and the generated impulse follows an approximately linear trend, although variability is marginally higher for the relationship between drag

impulses *I* and corresponding durations (*t*). For the relationship *I* vs *t*, $R^2 = 0.78$ (p-value $< 6.52$ x $10^{-16}$) for the drag events and 0.89 (p-value $< 4.2$ x $10^{-9}$) for the lift events, Figure 4a).

   The results from the ellipsoid sensor demonstrate a strong influence of the lift forces. Exceedance impulses occur for similar durations and magnitudes, however there is a strong bias of the lift distribution towards the shorter and low impulse events. The drag duration and impulse distributions include more and higher magnitude outliers than the lift distributions ($\boldsymbol{f_D} > -\boldsymbol{FG_D}$

impulses median = 0.08 N.s, $\boldsymbol{f_L} > -\boldsymbol{FG_L}$ impulses median= 0.022 N.s). For the ellipsoid, the *I* vs t relationship has $R^2 = 0.95$ (p-value $< 2.2$ x $10^{-16}$) for the drag events and 0.67 (p-value $< 2.2$ x $10^{-16}$) for the lift events (Figure 4b). For all these threshold exceeding events the sensor was vibrating until entrained as observed from both video and IMU data.

   Using the video recording observations, the impulse thresholds for entrainment were approximated with logistic regression. The probability of 0.5 corresponds to the threshold impulse for which the probability changes from the particle being more

likely to be at rest, to being more likely to be entrained. In this context, with this approximation we calculate a gradational threshold of entrainment (Begin and Schumm, 1979) and not an absolute one. The probability of entrainment as a function of impulse (Figure 5 (a) and (b)) highlights the control of short lift events on the entrainment of the ellipsoid. The impulse threshold for the sphere is close to 0, as all the approximated probabilities exceed 0.5. However, there is significant variability in this calculation (wide 95% confidence intervals) which indicates the wide range of impulses that can lead to entrainment of

the sphere. In contrast, the entrainment of the ellipsoid demonstrates a dependency on lift impulses as the lift threshold is lower (ellipsoid lift impulse threshold = 0.27 $\pm$ 0.03 N.s) and the drag threshold is approximated with less confidence (ellipsoid drag impulse threshold = 0.74 $\pm$ 0.27 N.s).

   Finally, the results from the field experiments (Figure 6) indicate similar statistical behaviour to the laboratory experiments but higher variability. Drag forces are of higher magnitude and duration than the lift forces ( $\boldsymbol{f_D} > -\boldsymbol{FG_D}$ impulses median

= 0.13 N.s, $\boldsymbol{f_L} > -\boldsymbol{FG_L}$ impulses median= 0.08 N.s), but there is an abundance of low magnitude lift impulses that affect strongly the motion of the ellipsoid. In the Erlenbach, the duration of the exceedence events is also a proxy for the generated impulse with the *I* vs t relationship being more linear for the lift events compared to the drag events (*I* vs t $R^2 = 0.66$, p-value $< 2.2$ x $10^{-16}$, for the drag events and 0.88, p-value $< 2.2$ x $10^{-16}$, for the lift events).

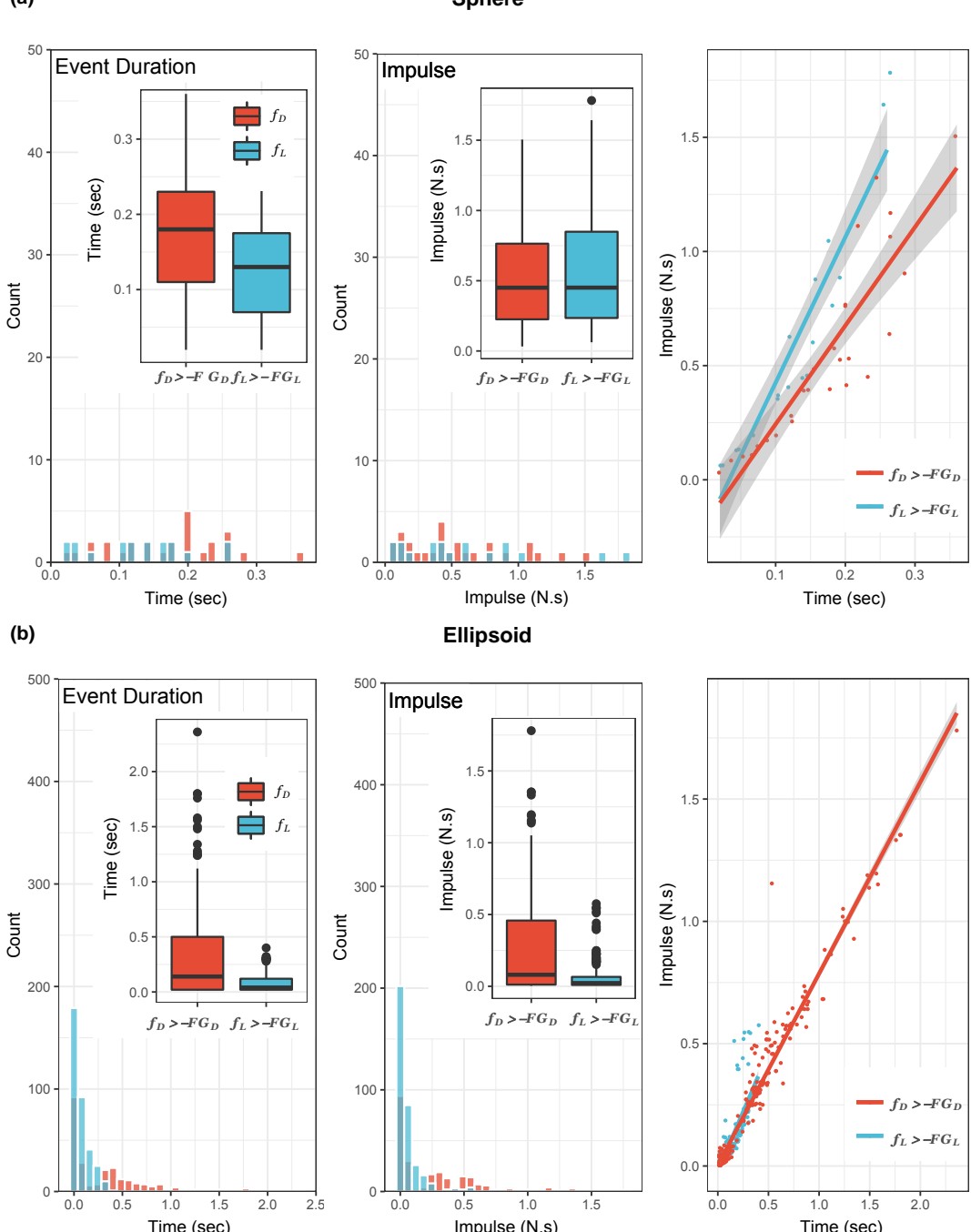

**Figure 4. Inertial impulses and duration of threshold exceedance events for laboratory experiments**. Impulses of all the inertial forces that exceeded the gravity forces. For the spherical particle **(a)** the drag impulse median (median of $f_D > -FG_D$ events) is 0.01 N.s higher than the lift impulse median (median of $f_L > -FG_L$ events). For the ellipsoid **(b)** the equivalent difference is 0.013 N.s. The relationship between the duration ($t$) and the impulse ($I$) during the exceedence events is linear for both the sphere and the ellipsoid (and for both $f_D > -FG_D$ and $f_L > -FG_L$) events. The entrainment of the ellipsoid is more dependent on short and low lift impulses than the sphere, demonstrating the effect of shape on the inertial dynamics.

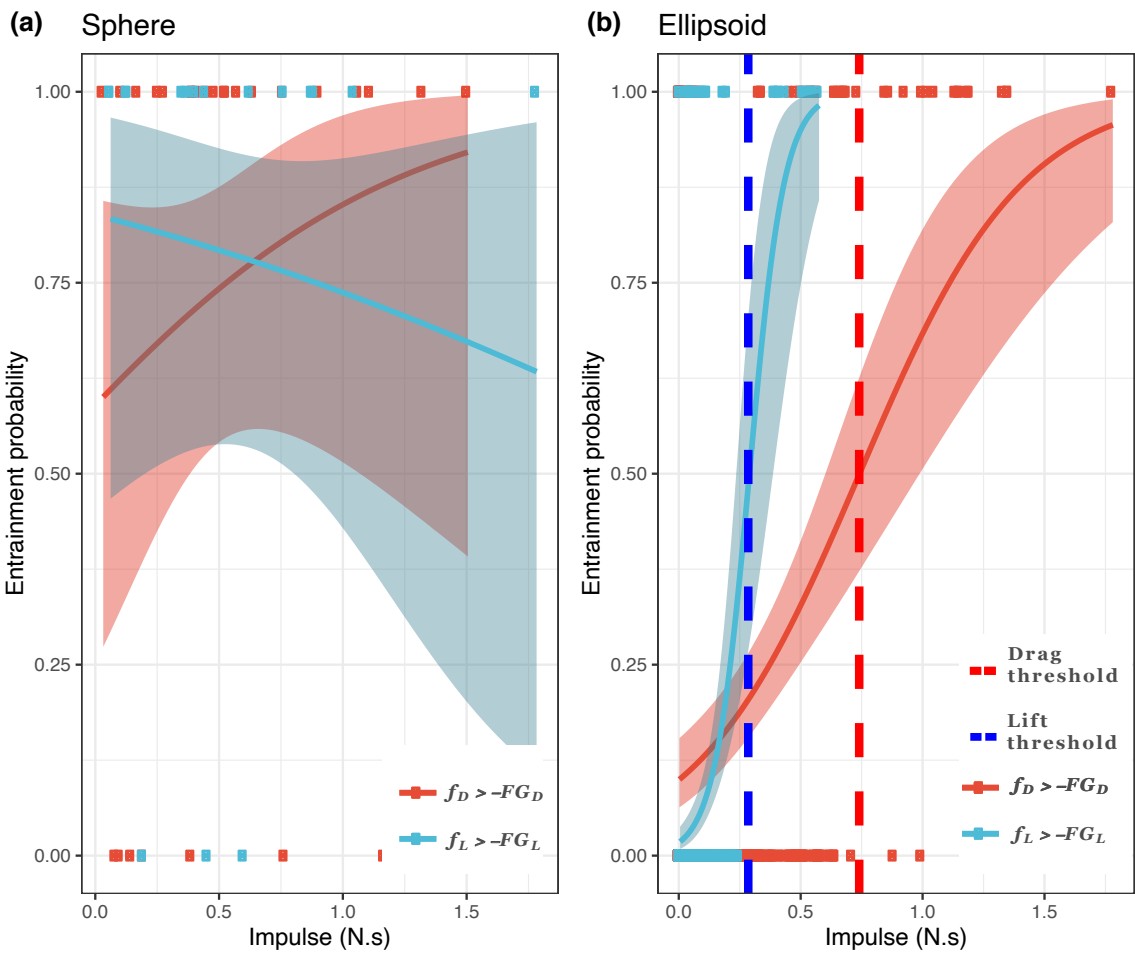

**Figure 5. Probabilistic inertial impulse threshold for laboratory experiments**. Logistic regression of the probability of entrainment for the spherical **(a)** and ellipsoid **(b)** particles. The calculation is based on the combination of video recordings and inertial impulse measurements during drag and lift threshold exceedance ($f_D > -FG_D$ and $f_L > -FG_L$ events). For the sphere there is little statistical difference between the calculated inertial impulses as over 95 % of the values relate to an entrainment event (the probability threshold 0.5 is always exceeded). However, there is significant variability in this calculation (wide 95% confidence intervals) which indicates the wide range of impulses that can lead to entrainment of the sphere. For the ellipsoid, the probabilistic lift inertial impulse threshold relaxes to 0.27 N.s (blue vertical line, (b)) and the drag threshold relaxes to 0.74 N.s (red vertical line, (b)).

# Ellipsoid Erlenbach

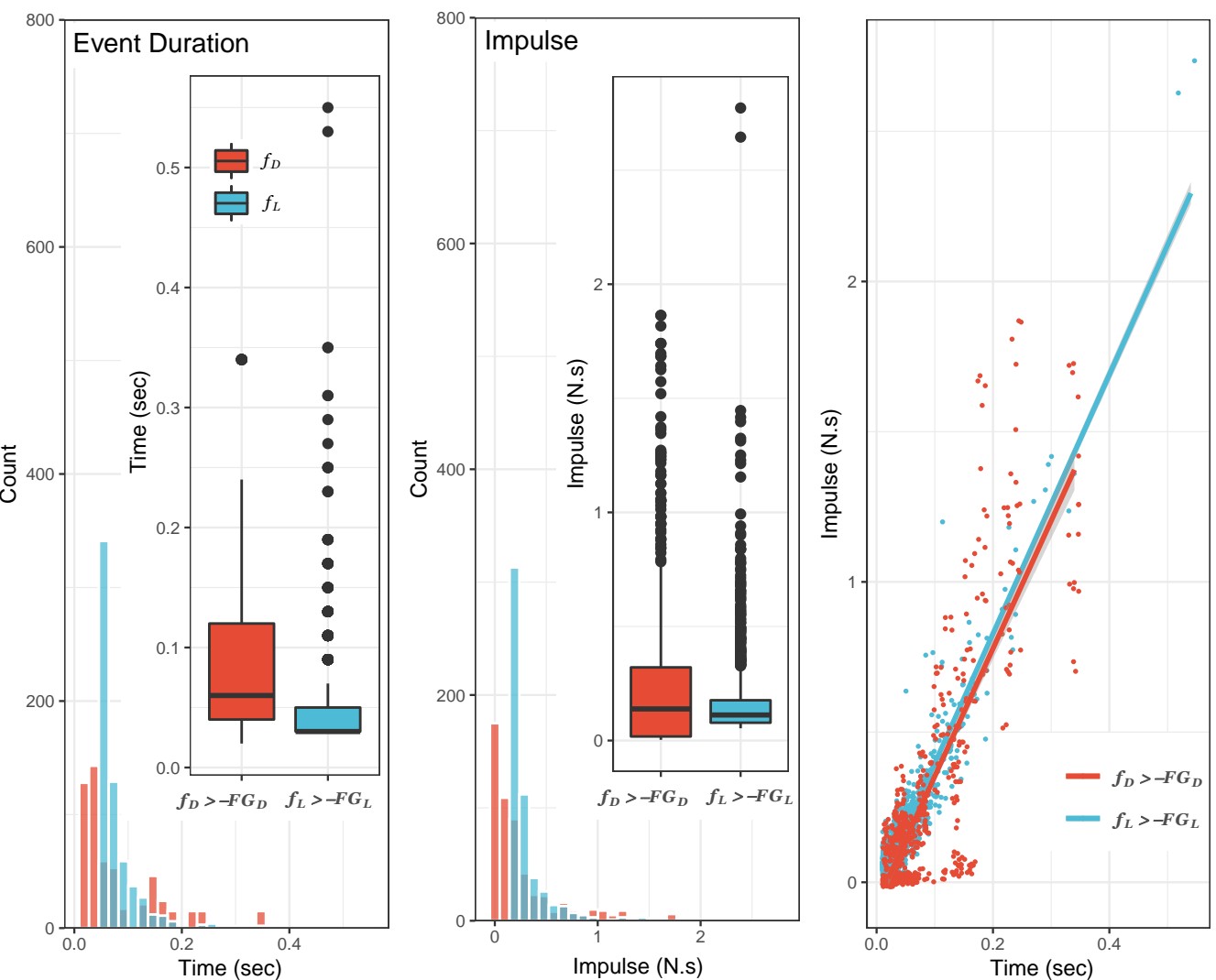

**Figure 6. Inertial Impulses and duration of threshold exceedance events for field experiments**. Impulses of all the inertial forces that exceeded the gravity forces. During short transport events (average travel distance = 2 $\pm 0.43$ m) the drag inertial impulse median (median of $\boldsymbol{f_D} > -\boldsymbol{FG_D}$ events) is 0.05 N.s higher than the lift inertia impulse median (median of $\boldsymbol{f_L} > -\boldsymbol{FG_L}$ events). The relationship between the duration ($t$) and the inertial impulse ($I$) is linear for both for both $\boldsymbol{f_D} > -\boldsymbol{FG_D}$ and $\boldsymbol{f_L} > -\boldsymbol{FG_L}$ events. During *in-situ* transport the drag forces are of comparable magnitude and duration, however, short and low magnitude $\boldsymbol{f_L} > -\boldsymbol{FG_L}$ impulses have a strong influence on the motion of the ellipsoid.

# 5 Discussion

## 5.1 IMU sensors and geomorphological applications

The advantage of using an IMU sensor for capturing grain motion is that the sensor solves a complex force and torque balance and removes any ambiguity in whether or not a test particle is in motion, as motion leads to the explicit thresholds $F_{net}$ and/or $T_{net}$ exceeding 0. Entrainment is captured directly and, assuming correct sensor calibration, robustly. IMUs can be a useful tool for geomorphologists since they offer a realistic prospect for monitoring particle motion during transport without invasive apparatus which is not possible with standard equipment, especially in field applications (e.g. PIT tracers). At the same time, it is important to recognise that that exceedance of the explicit thresholds above does not always produce complete dislodgment of the particle and also does not directly describe the modes of transport in the context that is commonly assumed for sediment hydraulics (e.g. differences in spinning and orbital rotations, section 2.1). For a complete understanding and effective prediction of grain motion both the hydraulic and the particle forces need to be measured, analysed and decoupled from the inertial forces we measure in this study.

Further, there has been a recent rapid increase in use of IMU sensors, but most off-the-shelf IMU sensors are not suitable for the range of forces characterising natural sediment transport, especially if the focus is on particle interaction or impacts (Maniatis et al., 2013). In addition, the physics of IMU sensors are complex and a number of common assumptions about their use do not always hold. For example, while dead-reckoning appears to allow positions to be recovered by double-integration of linear accelerations, uncertainties introduced during the production of IMUs (mostly nm scale imperfections on the alignment of the MEMS) lead to extreme uncertainty in positional estimates. A second issue involves calibrating IMU accelerometers which has often been done using free fall drop experiments. An accelerometer in free fall will measure 0 acceleration despite being subjected to the acceleration of gravity, as gravity in the context of the body frame of the accelerometer is a so-called fictitious force (Appendices A and B). Consequently, the force/impact results of a free fall drop experiment which relies solely on IMU measurements are highly dependent on how quickly the sensor is programmed to enter and wake up from the free fall detection state (Clifford, 2006). It is possible to approximate the height of the free fall using the approximate time of the free fall state. However, the measurement of the impact force needs a very detailed description of both the impact surface and the low-level code that controls all the basic operations of the sensors (on-off routines, logging, storage-handling etc.). This low-level programming is a black box for proprietary off-the-shelf sensors and for users without suitable programming skills. Finally, it is not possible to derive directional information, even for forces, for long mobile periods without complementary corrections (Kok et al., 2017) or without a detailed presentation of the reference frames involved and their initial alignment.

Here, we calibrated and deployed a commercial IMU sensor following standard procedures (Maniatis, 2016), but the precise corrections used are sensor specific and similar procedures should be followed again for any other IMU sensor. The calibration of force measurements is likely to be standardised and simplified in the near future as the use of IMU sensors develops further. Similar standardisation for the direction of forces is potentially further away as it requires using IMU sensors that rely on optical technology and which are currently not manufactured with physical dimensions or within a price range that is accessible for sediment transport studies (De Agostino et al., 2010).

## 5.2 Relationship with previous work and first order statistical generalisation

Two aspects of this study are particularly important to address before we make comparisons with previous studies. The first is that we made inertial measurements from within the sediment particles, which are fundamentally different from measurements of fluid turbulence that are often used for predicting sediment motion. The second is that the flow regimes under which we made measurements, with varying shallow flows, differ from those in many studies of sediment motion. Both of these aspects provide new insights into sediment movement, but they require care in making direct comparisons with studies that have used different approaches and/or hydraulic conditions. In addition, it is useful to note that grain protrusion is not discussed in this work, despite being an important control on grain motion and particularly entrainment (e.g. Dey and Ali, 2018), since the presented laboratory and field experiments only correspond to particles that are highly exposed to the turbulent flow.

This work uses a theoretical framework which has the potential to enhance the mathematical modelling of sediment transport. The Newton-Euler model of section 2, in conjunction with the quaternion transformations of Appendix A, can be read as a 3D and unrestricted Lagrangian - Eulerian model for sediment transport. In our analysis, particle dynamics are transformed from a Lagrangian domain (and the mobile body frame of the particle $b$) to a static Eulerian domain (frame $r$) which is most commonly used for the analysis of turbulent flow. Ballio et al. (2018) analyse the topic in detail and provide a comprehensive 1-D Lagrangian - Eulerian model which also accounts for the intermittency of sediment transport using a binary classification of mobile and non-mobile states. Our presentation can be used to define 3D Lagrangian dynamics, including rotation, in full and then to transform the corresponding kinematic properties to the Eulerian domain for direct comparison with the turbulent forces. We acknowledge that the verification of the 3D Lagrangian - Eulerian model is heavily dependent on the inertial measurements, and particularly the constant tracking of relative orientation between the frames. However, it is possible to predict that future calibration experiments deploying IMUs will be used this way to parametrise simulations.

Previous laboratory studies using fixed force meters attached to grains (Schmeeckle et al., 2007; Cameron et al., 2019) report nominal drag and lift forces of the order of $10^{-1}$ N for 0.008 - 0.025 m diameter particles. The measured force normalised by the submerged weight of the average diameter particle is of order 0.08. In the flume results presented here, the inertial drag and lift forces during entrainment are of order $10^1$ N, with normalised values for the sphere and ellipsoid being 15.6 and 17.1, respectively. Differences in force magnitude are expected since: a) the inertial sensor is freely mobile enabling the inertia of the moving particle to be recorded; and, b) the hydraulic regimes in both Schmeeckle et al. (2007) and Cameron et al. (2019) correspond to flows deeper than the particle diameters, with fully developed boundary layers.

Static vibration sensors were also deployed by Lamb et al. (2017) who attached them to a wide range of test particles ($D =$ between 0.075 and 0.218 m). They reported drag forces of order $10^1$ - $10^2$ N and and lift forces of order $10^1$ N (drag forces from 5.4 to 40.7 when normalised by the submerged weight of the average diameter particle). Also, the hydraulic regime used by Lamb et al. (2017) is comparable to both the laboratory and the field experiments presented here. The two studies yield very similar forces at entrainment, using different measurement methods, so validating the results from our inertial sensor. However, in general it is important to consider the type of sensor (static or restricted vs mobile), the data processing model and the experimental protocol (varied or steady flow) when different force measurements are compared.

Lamb et al. (2017) also observe a predominance of negative lift forces, especially for partially submerged particles, that may have significant morphological implications as potentially explaining, along with turbulence intensities, lower than predicted sediment fluxes in steep mountain streams. In our work, the inertial negative lift forces are measured (Appendix E) but the exceedance events ($f_L > -FG_L$) are only calculated for the positive lift forces ($F_{Lnet} > 0$). The negative lift forces are components of the resultant force which can have a strong hydraulic component, as argued in Lamb et al. (2017), but they can also be a reaction to positive lift forces during the motion of the particle (and especially the motion of the ellipsoid, Figures 3 and E) which requires further investigation.

The laboratory inertial impulse calculations demonstrate that, for unrestricted entrainments, there are observable differences between spherical and ellipsoid particles with the latter being more sensitive to the lift forces at entrainment threshold conditions. Those differences support previous work on the effect of shape on the response of particles in various hydraulic regimes (e.g. Komar and Li, 1986; Demir, 2000) and on the mode of near-bed transport. Measurements of the effect of particle shape can now be made directly using inertial sensors.

The corresponding inertial calculations from the field also demonstrate that the ellipsoid is highly sensitive to lift forces and impulses. We observe higher mean magnitude lift forces on the ellipsoid in the field (2.57 N) than in the laboratory (2.13 N) (Figure E1). The greater negative lift forces during motion suggest that the particle has a reaction to the positive lift forces, specifically those that exceed the threshold of motion and lead to transport. A similar increase in magnitude was observed for the drag components, with the instantaneous forces being up to 10 times the mean (Figure E1). The magnitude and duration of the exceedance events in the field are comparable to the observations in the laboratory experiments (Figures 4 and 6, the relationship between impulse and duration of events is more variable for the field experiments). However, the extreme forces are short lived and and so generate very small impulses close to 0 N.s. Differences in force magnitude and duration can relate to transitions, as described by Shih and Diplas (2018), from hydraulic "impulse controlled" transport, as in our flume experiments, to "force-magnitude controlled" transport corresponding to the dynamics recorded in Erlenbach. However, an important difference between laboratory and field experiments lies in the scales of turbulence (Coleman and Nikora, 2008; Singh et al., 2009), which requires further investigation since detailed flow measurements were not made during the presented experiments (eg. PIV measurements).

Overall, differences of particle inertial dynamics during grain entrainment and translation are important because they can potentially enhance predictions for grain particle travel distances with measurements from the field and particularly for large distances (Hassan et al., 1991, 2013). Measuring those differences is the most direct insight we can have for studying the effect of several morphological controls (eg. grain arrangement, burial depths, sediment sorting) until the high-frequency 3D measurement of tracer positions during transport becomes possible.

Considerable effort has been applied to define distributions for hydraulic impulses during the entrainment of spherical particles and relating them to critical thresholds (Diplas et al., 2008; Valyrakis et al., 2010; Celik et al., 2010; Valyrakis et al., 2010, 2011). In addition, there are recent efforts towards upscaling the effect of hydraulic impulses to fully developed bed-load equations (Shih and Diplas, 2018) and results pointing towards evaluating the morphological impact of different hydraulic

impulse regimes (Phillips et al., 2018). These works highlight the importance of deriving general statistical descriptions for grain inertial impulses.

Our data provide new insights into the roles of drag and lift impulses in entrainment. To begin to generalise these findings and to assess the interactions between lift and drag forces, a bootstrapping method is used here. We approximate the distributions of inertial lift and drag impulses for an ellipsoid particle and from a combination of laboratory and field measurements. This analysis is also the first step towards calculating the combined behaviour of the drag and lift distributions which can lead to the definition of joint distributions that have stronger explanatory and perhaps predictive value.

To combine the results from the laboratory and field ellipsoid experiments the impulsive exceedance events were normalised since the conditions are different for the laboratory and the natural conditions. Normalisation used the mean impulses for all drag and all lift forces, respectively. Also, the mean impulses were calculated separately for the laboratory and field experiments ($\hat{I} = I/\bar{I}exp$). After the normalisation, the laboratory and field results are combined into one dataset of normalised drag ($\hat{I}_{Drag}$) and normalised lift ($\hat{I}_{Lift}$) impulses, which are assumed to be uncorrelated. The latter is justified by the fact that the point lift and drag forces are statistically uncorrected as shown in Appendix E (Figure E2).

The fitting of the representative distributions for $\hat{I}_{Drag}$ and $\hat{I}_{Lift}$, permits bootstrap sampling from those distributions. Figure 7 shows 50000 random $\hat{I}_{Drag}$ and $\hat{I}_{Lift}$ combinations, sampled from the selected gamma and log-normal distributions (for $\hat{I}_{Drag}$ and $\hat{I}_{Lift}$, respectively, Appendix F). After taking into account the normalised drag and lift impulse thresholds as defined in the Results (Figure 5), we conclude that the probability for the exceedance of the lift threshold is approximately 0.02, the probability for the exceedance of the drag threshold is approximately 0.005 and the probability of both thresholds being exceeded simultaneously is approximately 0.0001. The calculation confirms the observation that the transport of the ellipsoid particle is defined by states of zero or very low mobility (97.8% for the calculated combinations corresponds to dynamics that are below the normalised probabilistic impulse threshold for entrainment, Figure 5). 80% of the 1371 threshold exceedance events corresponds to lift threshold exceedances, 19% to drag threshold exceedances and 0.4% to exceedances of both thresholds. The calculation suggests that the majority of the mobility states of the ellipsoid will relate to the action of lift forces. The very small probability for the simultaneous exceedance of both thresholds is another possible effect of the particle's shape as spherical particles will protrude more and are more likely to be equally affected by both drag and lift components (Figure 5a). This type of calculations requires sample sizes that only advanced instrumentation, such as that presented in this work, can deliver. Similar frameworks can be used for meta analysis of existing results and to inform the design of future experiments.

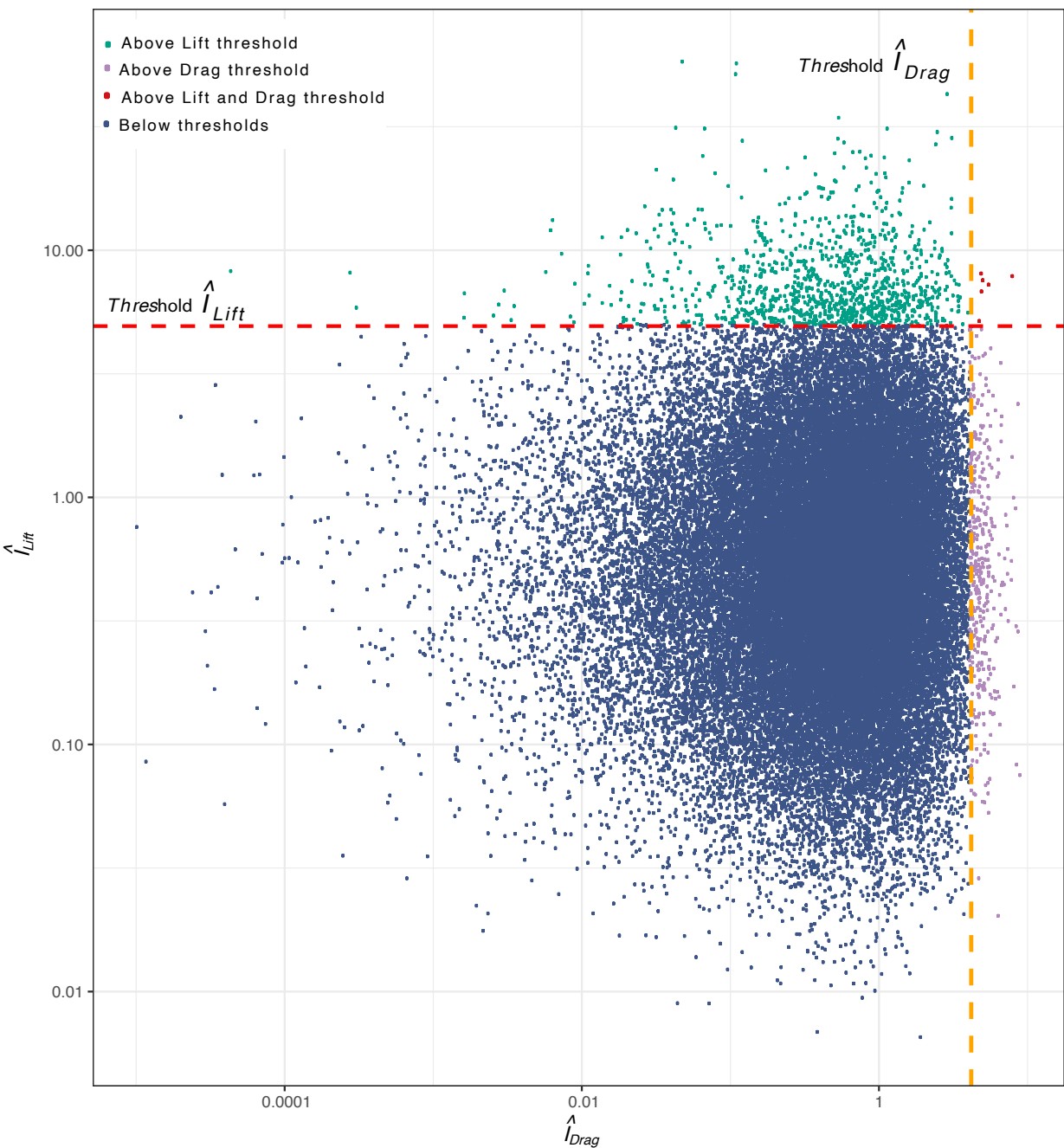

**Figure 7. Bootstrap normalised impulse sampling (lift and drag).** $\hat{I}_{Drag}$ and $\hat{I}_{Lift}$ are fitted with a gamma and a log-normal distribution, respectively (Appendix F2). The normalised drag and lift thresholds (red dashed lines), are calculated using the probabilistic drag and lift inertial impulse thresholds presented in Figure 5 which were divided by the recorded mean inertial drag and lift impulse from the laboratory experiments.

## 6 Conclusions

This work introduces a framework that can be used to derive and interpret IMU measurements in sediment transport studies. The derivation of inertial measurements from mobile sediment grains requires a physical model that links the inertial dynamics with existing force (or moments) balance equations for sediment transport. The types of sensors, and associated smart pebble assemblies currently deployed for the measurements of grain inertial dynamics are not suitable for 2D or 3D tracking of grain position. However, it is possible to measure net forces and impulses if the necessary transformations are applied consistently.

Field and laboratory measurements of inertial lift and drag impulses highlight the different entrainment behaviours of a spherical and an ellipsoidal particle. The lift net force is dominant during the unrestricted entrainment of the ellipsoid while there is no statistical difference between the effects of lift and drag impulses on the entrainment of the sphere. The drag component can be stronger during transport, however short impulses influence the motion of the ellipsoid significantly.

The continuous improvement of the sensor technology along with the better understanding of the physics described by 480 inertial measurements can lead to a unified treatment of the resultant grain dynamics during bed-load transport. These are the dynamics that represent exactly the interaction of hydraulic and sediment forces in different regimes and can enhance the parametrisation of important hydro-morphological controls.

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

*Author contributions.* Georgios Maniatis performed the design and calibration of the sensor, the design and implementation of the experiments, all the physical and statistical calculations and produced the first draft of this paper. Trevor Hoey supervised the laboratory experiments, reviewed several versions of the manuscript and contributed significantly to the interpretation and contextualisation of the results. Rebecca Hodge contributed to the design of the laboratory experiments, reviewed several versions of the manuscript and contributed to the interpretation and contextualisation of the results. Dieter Rickenmann contributed to the design, supervised and assisted with the field experiments, reviewed several versions of the manuscript and contributed to the interpretation and contextualisation of the results. Alexandre Badoux contributed to the design of the field experiments and reviewed several versions of this manuscript.

*Competing interests.* The authors declare no competing interests

*Acknowledgements.* The flume experiments were contacted in the School of Engineering of the University of Glasgow. At the time of the flume experiments Georgios Maniatis (GM) was supported by a University of Glasgow Kelvin Smith Scholarship. The field experiments were supported by an Early Career Researcher award from the British Society for Geomorphology (2018). The authors thank Tim Montgomery (University of Glasgow) and Tobias Nicollier (WSL) for their assistance with the laboratory and field experiments, respectively, three anonymous reviewers who improved significantly this manuscript with their contribution and Jens Turowski who provided feedback on previous versions of this manuscript. Finally, GM thanks Katerina Georgiou for her assistance with the design and production of the figures and the typewriting of the manuscript.

## Appendix A: Frames of reference, rotations and IMU measurements

To discuss the measurements recorded by an IMU, and particularly the measurements from an accelerometer and a gyroscope, it is necessary to introduce three basic frames of reference and select one of the many representations for arbitrary rotations in 3D.

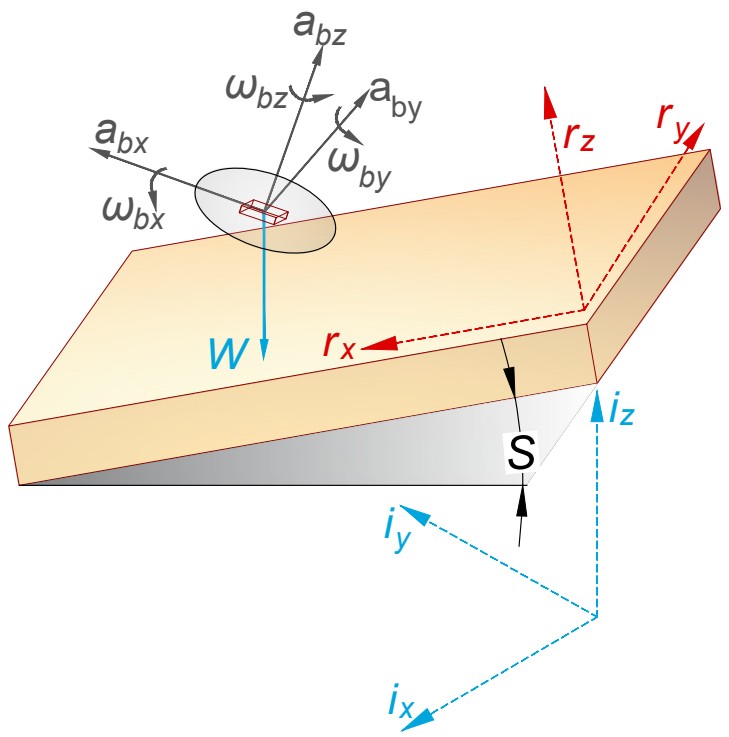

**Figure A1. Frames of reference** $r$ stands for river-bed (flume in this case) reference frame, $b$ for body frame and $i$ for the inertial reference frame.

The following assumptions and simplifications are used throughout this study:

– due to the small scale ($10^{-1}$ to $10^1$m, typically) motion of sediment grains, an Earth frame (one that coincides with

the inertial frame as defined below, but rotates with the Earth) is not defined. Also, the angular velocity of the earth (approximately $7.29 \cdot 10^{-5}$ rad.s$^{-1}$) is ignored.

– for the same reason, the non-gravitational fictitious forces (such as the Coriolis effect) are ignored.

- for the mathematical derivations, ideal IMUs (no error accumulation is considered) and perfectly aligned sensor assemblies are assumed. The errors associated with IMUs and especially with the magnitude of the integration errors are presented in relevant electrical engineering sources (eg. Kok et al., 2017).

We define the body frame $b$ as the coordinate frame of the moving IMU. For an ideal IMU the origin of this frame is located exactly at the center of both the accelerometer and the gyroscope and this center falls precisely on the center of mass of the complete sensor assembly (Maniatis et al., 2013, 2017). Frame $b$ is mobile but fixed in the sensor assembly (fixed axes representation).

The local geographical frame $r$ is the stationary frame within which hydrodynamics are analysed. This is the reference frame used implicitly for all the single grain motion studies (Dey and Ali, 2018). For laboratory experiments, the $r_x$-$r_y$ plane is exactly parallel to the flume bed and the $r_z$ direction is normal to the bed. For the field experiments this alignment will be an approximation due to variations of the local topography. The inertial frame $i$ is a stationary frame. Strap-down IMUs measure acceleration and angular velocity changes in response to this frame and its origin lies at the center of the Earth.

Transforming information between these three reference frames is non-trivial. A widely-used method to represent the change between frame is the application of quaternions (Hamilton, 1844; Diebel, 2006). Quaternions are an extension of complex numbers used in the description of 3D mechanics, particularly 3D rotations. They are considered the most efficient description of unrestricted 3D rotations, as they are free from numerical errors that occur when other representations are used (such as the Gimbal Lock error associated with rotations expressed by Euler Angles, Appendix G2). A typical introduction to quaternions can be found in Valenti et al. (2015) and we follow that primer for a brief introduction to quaternion algebra in Appendix G1.

A unit quaternion $_A^Bq$ defines a rotation from frame $A$ to frame $B$ and successive rotations are represented by quaternion multiplication. For each $_A^Bq$, a Direction Cosine Matrix (DCM) $R(_A^Bq)$ is defined as a function of $_A^Bq$ components (equation G11) which also represents a rotation from frame $A$ to $B$ . If $^Bv$, $^Av$ are observations of the vector $v$ in frames $B$ and $A$ respectively, they are related through the following typical matrix operation:

$$^Bv = R(_A^Bq)^Av \tag{A1}$$

If the frames $A$ and $B$ are relatively static (such as the inertial frame $i$ in relation to the local geographic frame $r$) then both $_A^Bq$ and $R(_A^Bq)$ are explicit. If $B$ is rotating in relation to $A$ (such as the body frame $b$ in relation to the inertial frame $i$), $_A^Bq$ and the corresponding $R(_A^Bq)$ need to be recursively updated. The transition quaternion $\tilde{q}$ between two successive poses is defined by the applied angular velocity as:

$$\tilde{q} = \left[\cos\frac{\|\omega\|\delta t}{2} \quad \sin\frac{\|\omega\|\delta t}{2}\frac{\omega_{bx}}{\|\omega\|} \quad \sin\frac{\|\omega\|\delta t}{2}\frac{\omega_{by}}{\|\omega\|} \quad \sin\frac{\|\omega\|\delta t}{2}\frac{\omega_{bz)}}{\|\omega\|}\right]^T \tag{A2}$$

where $\omega_{bx}, \omega_{by}, \omega_{bz}$ are angular velocities observed along the $b_x, b_y, b_z$ body frame axes respectively by the 3D gyroscope, $\|\omega\| = \|\omega_b\| = \sqrt{\omega_{bx}^2 + \omega_{by}^2 + \omega_{bz}^2}$ is the norm of angular velocities and $\delta t$ the time of rotation, set here equal to the frequency

of the IMU measurements.

equation A2 is part of the direct multiplication method (Whitmore, 2000; Zhao and van Wachem, 2013) and the updated quaternion ${}_A^B q'$ is derived as:

$$
{}_A^B q' = {}_A^B q \bigotimes \tilde{q} \tag{A3}
$$

with the operation $\bigotimes$ denoting quaternion multiplication (equation G4). After each update ${}_A^B q'$ is set as ${}_A^B q$.

Inertial accelerometers measure the proper acceleration $a_b$ applied within the body frame $b$. These accelerations will include gravitational acceleration, a uniform force in the inertial frame $i$. To derive the linear acceleration in the inertial frame $i$ it is necessary to rotate the body frame measurement to the inertial frame and then to subtract gravitational acceleration. For this

rotation the recursively updated $R({}_b^i q')$ DCM is used after calculating the ${}_b^i q'$ through equation A3. The linear acceleration in the local geographical frame $r$ is then given by:

$$
a_r = R({}_i^r q)(R({}_b^i q')a_b - g) \tag{A4}
$$

where $a_b = [a_{bx}\ a_{by}\ a_{bz}]^T$ is the vector of the $b$ frame accelerometer measurements, and $R({}_i^r q)$ the explicit DCM that rotates the accelerations from $i$ to $r$ derived from the quaternion ${}_i^r q$. $a_r$ defines the magnitude and direction of the resultant force or

730 net force.

**A1 Accelerometers as orientation sensors**

In the equation A4, the raw accelerometer measurement is "compensated" for the action of the gravitational field in order to extract a measurement for the resultant force in the $r$ frame. To explain the need for this calculation, it is important to note that accelerometers measure proper acceleration. Proper acceleration is different from coordinate acceleration which is

735 generally defined as the rate of velocity (in a fixed inertial frame). In practice, if an accelerometer is placed on a flat surface will measure an upwards (positive) acceleration equal to 1g (9.81 m.s$^{-2}$). In free fall, an accelerometer will measure 0 acceleration because in the non-inertial frame of the sensor there is neither a static (eg. gravity) nor a dynamic (motion or vibrational) force applied and the accelerometer will only "feel" the terminal velocity of the impact if it lands on a surface. When the accelerometer is subjected to an external force and it begins to move (in relation to a static inertial frame), there is no way

to separate the components of the static and the dynamic forces, unless the relative orientation of the frames is constantly monitored (in the manner we describe in equation A4). At the same time, when the sensor is static, the non-compensated signal (the raw acceleration from the sensor) can provide an estimate for the relative orientation of the sensor's frame ($b$ - $i$ or $b$-$r$

quaternions in relation to the frame that static forces are applied (gravity frame, $i$). Valenti et al. (2015), provide the solution for the calculating the $_i^b q$ quaternion from raw accelerometer measurements which, adapted to the notation used here is:

$$
\quad {}_i^b q = \begin{cases} \left[ \sqrt{\dfrac{a_{bz}+1}{2}} \quad -\dfrac{a_{by}}{\sqrt{2(a_{bz}+1)}} \quad \dfrac{a_{bx}}{\sqrt{2(a_{bz}+1)}} \quad 0 \right]^T & , a_{bz} \geq 0 \\[4ex] \left[ -\dfrac{a_{by}}{\sqrt{2(1-a_{bz})}} \quad \sqrt{\dfrac{1-a_{bz}}{2}} \quad 0 \quad \dfrac{a_{bx}}{\sqrt{2(1-a_{bz})}} \right]^T & , a_{bz} < 0 \end{cases} \tag{A5}
$$

In this work we used equation A5 to estimate the initial alignment of the frames for both the flume and the field experiments. In addition, the YEI sensor implements an onboard calibration routine which can verify the initial alignment using sensor fusion and a series of nonlinear filters (YEI, 2014).

## Appendix B: 3D IMU Measurements

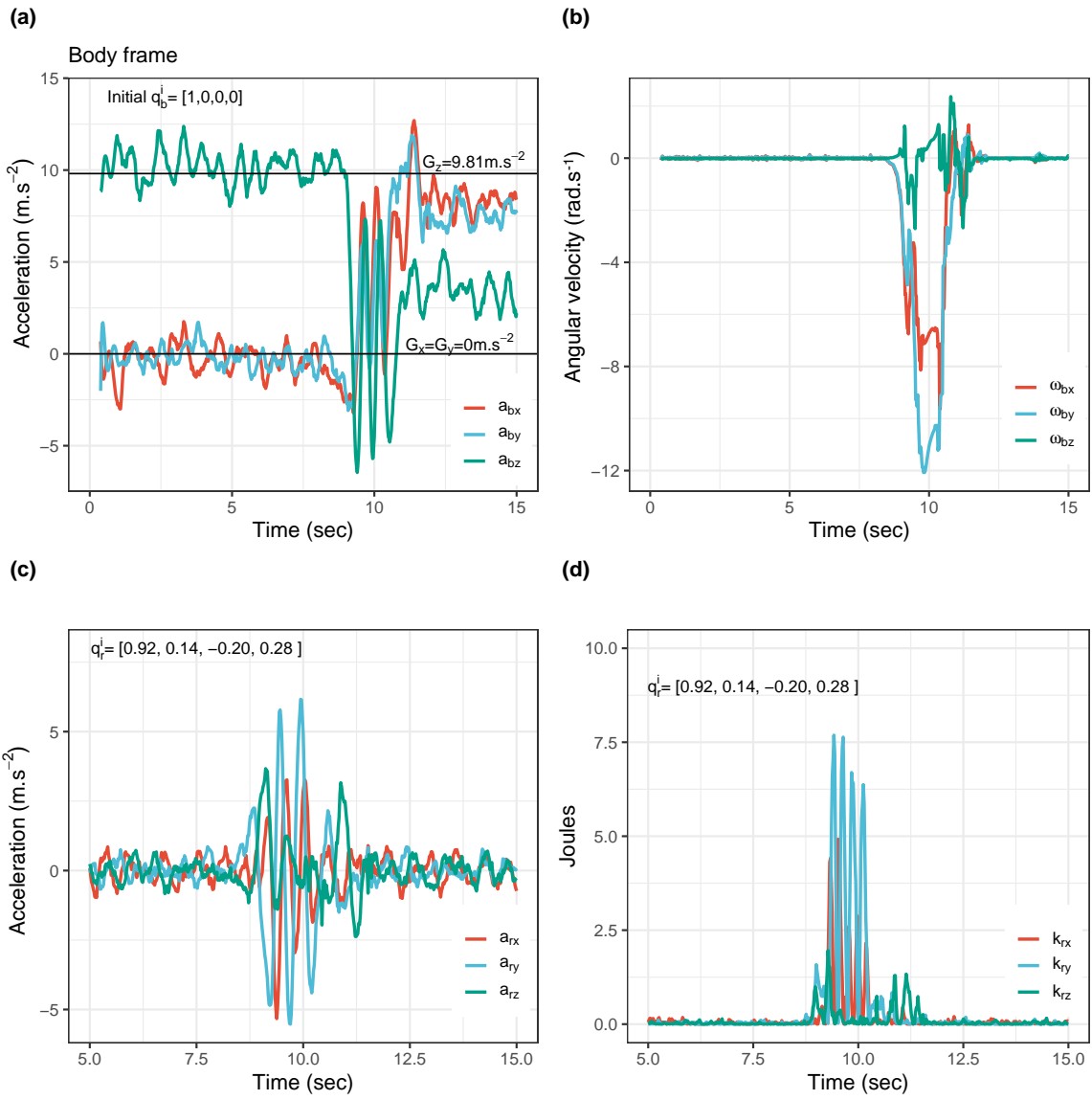

**Figure B1. Example incipient motion IMU data.** Measurements from the incipient motion experiments using the spherical sensor. **(a)** unfiltered and uncompensated inertial acceleration measurements. The sensor is initially aligned to gravity which results in the $z$ axis of the accelerometer measuring a mean value of 9.81 m.sec$^{-2}$. **(b)** angular velocity (rad.sec$^{-1}$) measurements derived from the gyroscope. **(c)** linear acceleration along the three body frame axis. This is is the result of removing gravity from the inertial measurements shown in A and applying a FFT-high pass filter as described in Maniatis, 2016 (Chapter 6). **(d)** shows the kinetic energy calculation after integrating once the signal presented in (c) and applying the formula $K = \frac{1}{2}m\|v_r\|^2 + \frac{1}{2}I_{cm}\|\omega_b\|^2$ as described in section 2.

## Appendix C:  The magnitude of the tangential forces and resultant torques

The rotational component captured by the gyroscope relates to the moments of the forces applied on the surface of the measured particle via equation 2. For a sphere, $f_{rot}$ is defined by rearranging equation 11 as:

$$f_{rots} = \frac{I_{cm}\left\|\frac{d\boldsymbol{\omega_b}}{dt}\right\|}{R} \tag{C1}$$

where R is the radius of the particle. The relationship holds because the moment of inertia ($I_{cm}$) of the sphere is uniform. This implies that the shape of particle does not affect the direction of the resultant force. For the spherical sensor presented in this work the moment of inertia is calculated using the formula for a sphere ($I_{cm} = \frac{5}{2}mR^2$) and is equal to 0.00085 kg.m$^2$.

For the ellipsoid, the same calculation is significantly more complicated. Firstly, the moment of inertia is not uniform. In this work, we implemented a numerical calculation during the design phase of the enclosure (using Solidworks, Système, 2016) for the principle axes of the ellipsoid. The principle axes coincide with the $b_x$, $b_y$ and $b_z$ body frame axes. Those are the equivalent of the axes $a$, $b$ and $c$ of the ellipsoid (where $a = 0.1$ m, $b = 0.07$ m and $c = 0.03$ m), they are fixed in the body frame and aligned with the IMU axes (see Appendix A). The principle components of inertia for the ellipsoid sensor were calculated as: $I_{xx} = 0.00057$ kg.m$^2$, $I_{yy} = 0.00060$ kg.m$^2$ and $I_{zz} = 0.00094$ kg.m$^2$. The non-principle components of inertia were calculated at the order of $10^{-8}$ kg.m$^2$ and they were ignored. The balance of moments for the principle axes is given by the following system of equations (often discussed as Euler equations, O'Reilly, 2008):

$$M_x = I_{xx}\frac{d\omega_{bx}}{dt} - (I_{yy} - I_{zz})\omega_{by}\omega_{bz} \tag{C2}$$

$$M_y = I_{yy}\frac{d\omega_{by}}{dt} - (I_{zz} - I_{xx})\omega_{bz}\omega_{bx} \tag{C3}$$

$$M_z = I_{zz}\frac{d\omega_{bz}}{dt} - (I_{xx} - I_{yy})\omega_{bx}\omega_{bz} \tag{C4}$$

For the of the tangential force, a good approximation (ignoring the secondary moments of inertia) is given by dividing the principle moments by the half length of the corresponding principle axis of the ellipsoid giving:

$$f_{rote} = \sqrt{\left(\frac{2M_x}{a}\right)^2 + \left(\frac{2M_y}{b}\right)^2 + \left(\frac{2M_z}{c}\right)^2} \tag{C5}$$

The equations C1 and C5 were used to calculate the magnitude of the rotational component for the example experiments presented in Figure 3. Figure C1, shows the same calculations at a scale that reveals their fluctuation.

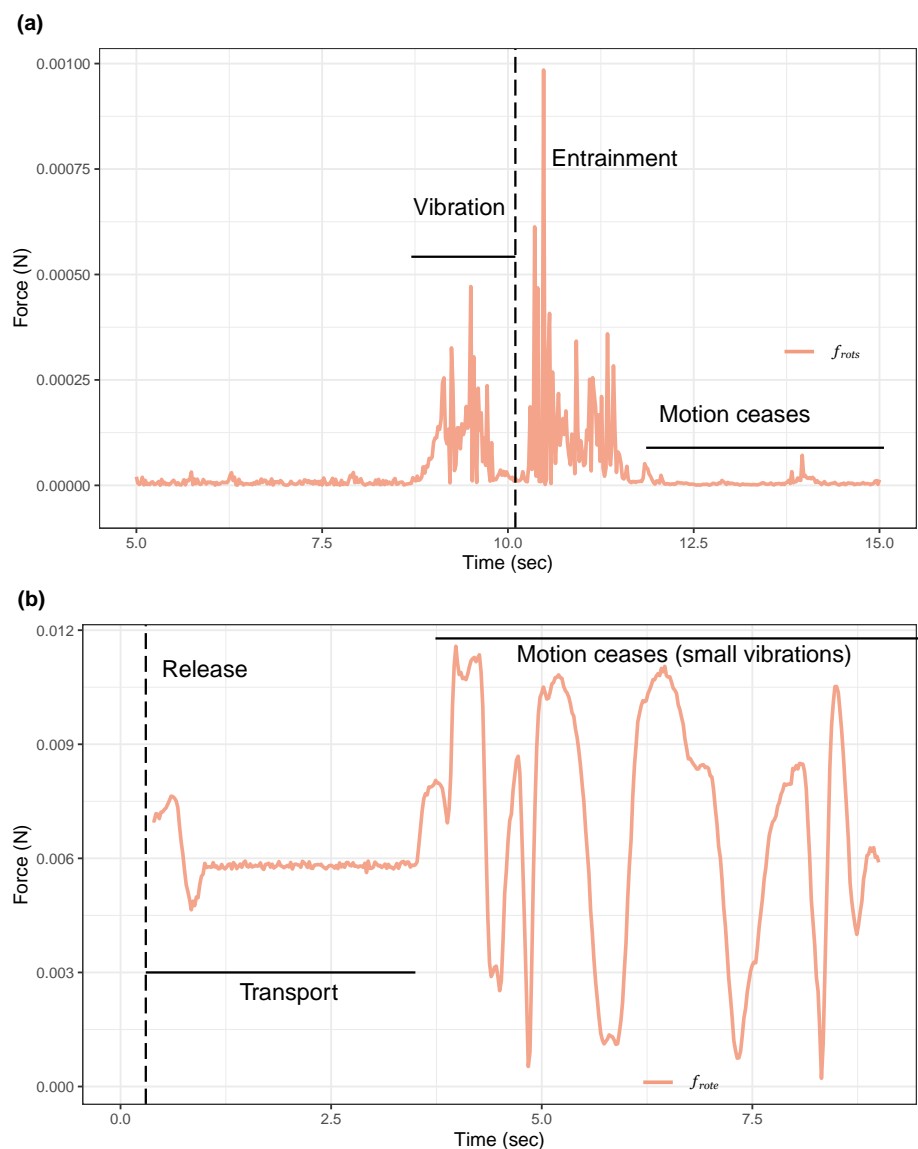

**Figure C1. Force magnitude of the rotational component (a)** Flume experiment (incipient motion) using the spherical sensor (Figure 3a). **(b)** Field experiment (Erlenbach) using the ellipsoid sensor (Figure 3e). The differences between the linear and the rotational componenents are between one and two orders of magnitude. $f_{rot}$ is of the order between $10^{-3}$ and $10^{-2}$ N (sphere in the flume and ellipsoid in Erlenbach respectively) while the linear forces (applied directly on the centre of the mass of the particle) are of the order between $10^{0}$ and $10^{1}$ N (Appendix E)

**Appendix D:  Hydraulic parameters**

**Table D1.** The parameters are estimated as follows: $\rho_p/\rho_f$ is the ratio of an experimental particle density to fluid density ($\rho_f = 1000$ kg.m$^{-3}$); $Q$ is the flow rate at the point of entrainment for the flume experiments and during the transport events in Erlenbach; $H$ is the flow depth (measured from the bottom of the bed to the water surface for Erlenbach and from the top of the hemisphere bed for the flume experiments); $U_b = Q/A$ is the bulk mean velocity ($A$ is the cross sectional area of the flow); $W$ is the channel width and $S_b$ is bed slope. 0.105 (or 0.1) is also the average bedslope of the lowermost natural reach in Erlenbach of about 30 m length upstream of the stream gauging station; $F = U_b/(gd)^{0.5}$ is the Froude number; $R_b = (U_b d)/\nu$ is the bulk Reynolds number (where $\nu$ is the fluid kinematic viscosity at 0°C for Erlenbach and at 25 °C for the flume experiments). For the calculations refering to the ellipsoid, we assume $D$ equal to the particle's b-axis

| Experiment | $\rho_p/\rho_f$ | Particle axis $a$ (m) | Particle axis $b$ (m) | Particle axis $c$ (m) | Protrusion (m) | $Q$ (l/s) | $H$ (m) | $W$ (m) | $S_b$ | $U_b$(m/s) | F | $R_b$ | $\tau_*$ |
|---|---|---|---|---|---|---|---|---|---|---|---|---|---|
| Flume (sphere) | 2.67 | 0.09 | 0.09 | 0.09 | 0.05 | 24.8 | 0.09 | 0.9 | 0.02 | 0.30 | 0.32 | 27555 | 0.01 |
| Flume (ellipsoid) | 2.67 | 0.10 | 0.07 | 0.03 | 0.03 | 45 | 0.12 | 0.9 | 0.02 | 0.41 | 0.38 | 50000 | 0.02 |
| Erlenbach (ellipsoid) | 2.67 | 0.10 | 0.07 | 0.03 | 0.03 | 120 | 0.15 | 3.5 | 0.1 | 0.22 | 0.19 | 34285 | 0.095 |

**Appendix E: Summary statistics for inertial lift and drag forces**

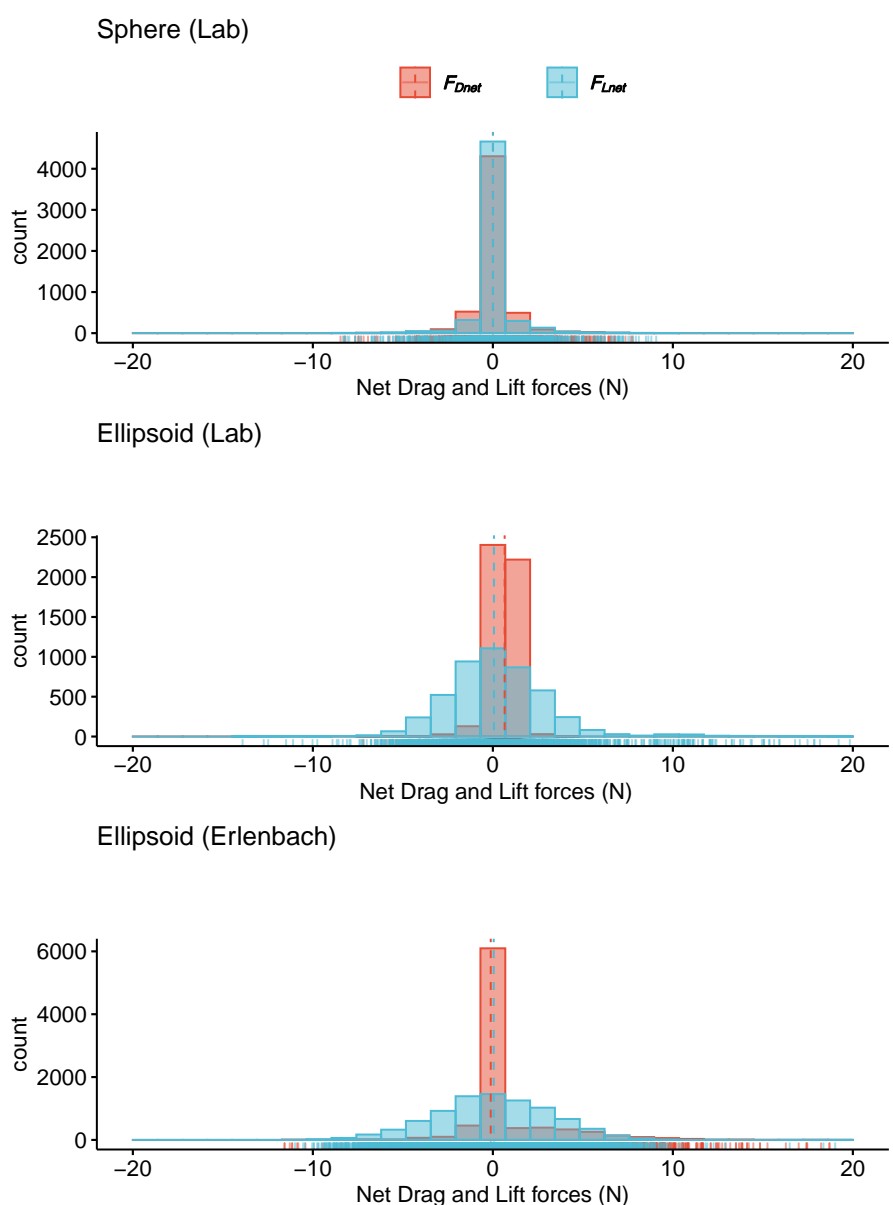

**Figure E1. Histogram of inertial forces from all experiments.** The inertial dynamics show that net lift ($F_{Lnet}$) and drag ($F_{Dnet}$) forces consistently fluctuate around zero. The vertical lines indicate the corresponding medians. The mean force magnitude for the sphere (in the lab) is $\bar{F}_{Dnet} = 0.57$ N and $\bar{F}_{Lnet} = 0.62$ N for the drag and lift directions, respectively. For the ellipsoid in the laboratory experiment the mean drag force magnitude is $\bar{F}_{Dnet} = 0.74$ N and the mean lift force magnitude is $\bar{F}_{Lnet} = 2.13$ N. Finally for the ellipsoid in Erlenbach mean $\bar{F}_{Dnet} = 1.22$ N and $\bar{F}_{Lnet} = 2.57$ N

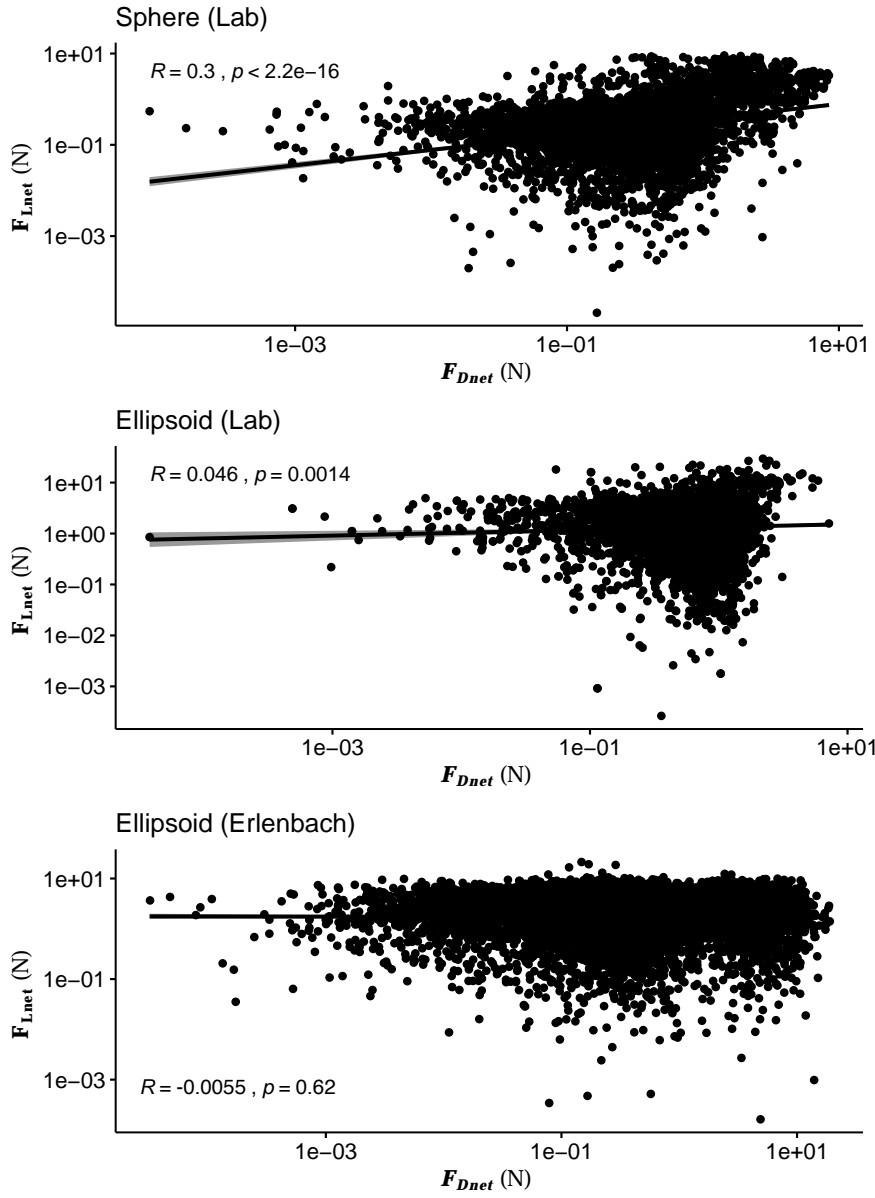

**Figure E2. Lift vs Drag force magnitude correlation (flume experiments).** Regression analysis applied on the magnitude of calculated forces (drag and lift) a moderate correlation for the spherical particle (statistically significant Pearson's R = 0.3) and a weak correlation for the ellipsoid both in the laboratory and the field experiments (statistically significant Pearson's R = 0.046 for the lab measurements, Ellipsoid (Lab), and not significant Pearson's R = -0.00055 for the field ones, Ellipsoid (Erlenbach)). The latter supports the assumption of statistical independence between the two components for the ellipsoid, justifying the randomisation presented in section 5 (Figure 7). Pearson's R is an unbiased metric for this sample size.

## Appendix F: Normalised impulse: selection of representative Drag and Lift distributions

Three types of right tail distributions were considered (Cullen et al., 1999, F2) as good fitting candidates for both $\hat{I}_{Drag}$ and $\hat{I}_{Lift}$: the Weibull, the gamma and the lognormal distributions. Goodness- of- fit analysis (Table F1), shows that $\hat{I}_{Drag}$ is approximated marginally better by a gamma distribution (median shape = 0.529, median scale = 0.5) and the $\hat{I}_{Lift}$ is approximated better by a lognormal distribution (median meanlog = - 0.66, median sdlog = 1.13).

Figure F1(a) shows the Cullen and Frey diagram for the identification of candidate distributions for the normalised Drag Impulses ( $\hat{I}_{Drag}$). Being in the "beta" region, the diagram indicates towards a right tail distribution (beta distributions are restricted between 0 and 1 and there is no physical relationship with the possible values for impulses). The skewness vs kurtosis relationship (blue dots), indicates a right tail distribution as a candidate as well. Figure F1 **(b)-(e)** show the graphical comparison between three candidate distributions (Weibull, gamma, and log-normal). Weibull and gamma distributions outperform the lognormal on the tails of the histogram (Q-Q plot), The median values are also captured better by the Weibull and gamma distributions (P-P plot). Finally, the histogram and CDF diagrams confirm that the log-normal distribution is the least representative of $\hat{I}_{Drag}$. The gamma distribution marginally outperforms the Weibull distribution for $\hat{I}_{Drag}$ both in graphical and goodness of fit comparisons and it is selected for the bootstrap calculation of Figure 7.

Figure F2 (a) shows the Cullen and Frey diagram for the identification of candidate distributions for the normalised lift impulses ( $\hat{I}_{Lift}$). The skewness vs kurtosis relationship (blue dots), indicates a right tail distribution as a candidate. Figure F2 **(b)-(e)** show the graphical comparison between three candidate distributions (Weibull, gamma, and log-normal). The log-normal distribution outperforms the other candidates at the tail (Q-Q plot) and the median regions (P-P plot). Finally, the histogram and CDF diagrams confirm that the log-normal distribution is the best representative of $\hat{I}_{Lift}$ and it is selected for the bootstrap calculation of Figure 7.

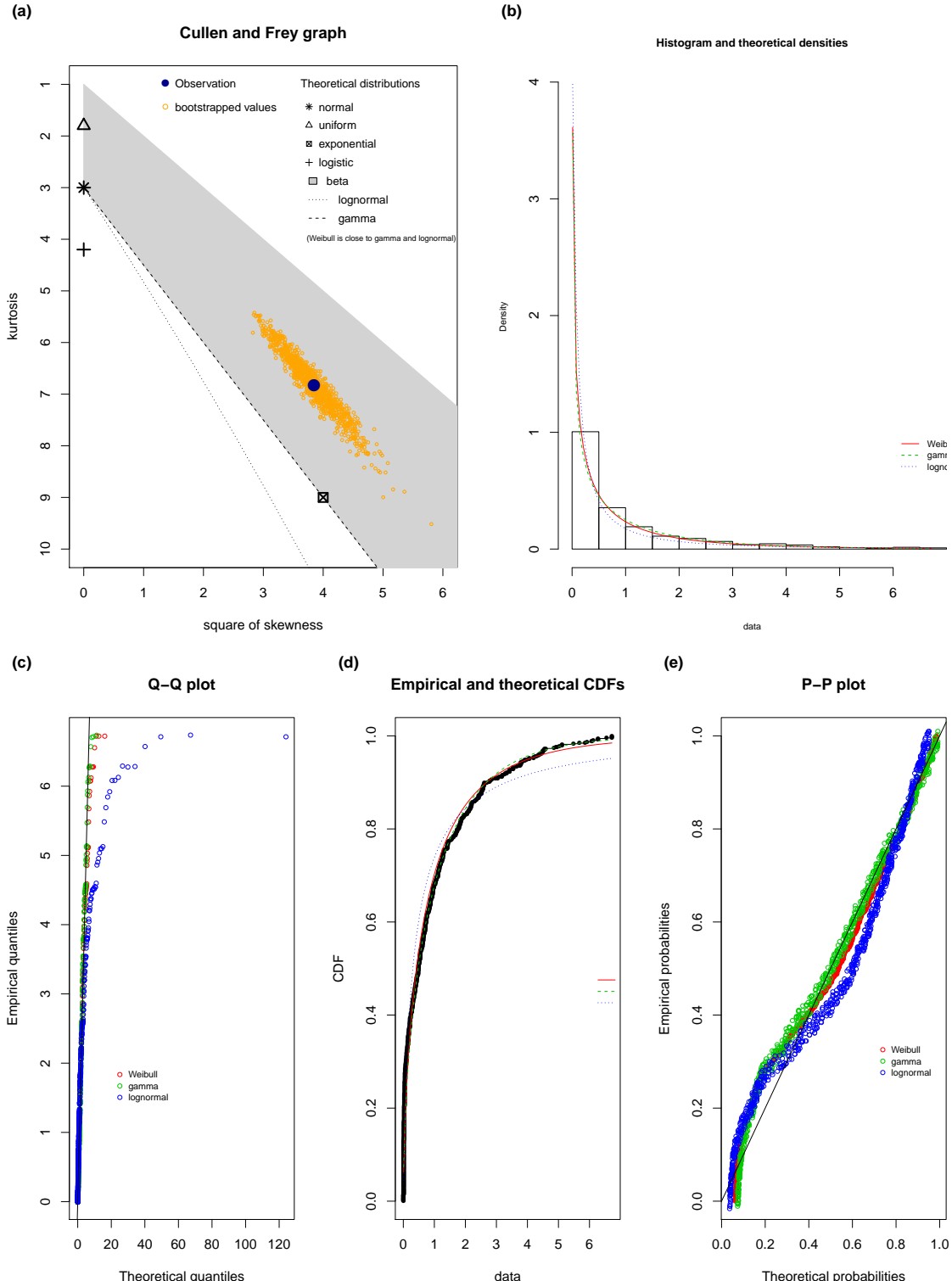

**Figure F1. Choice of distribution for drag impulses**

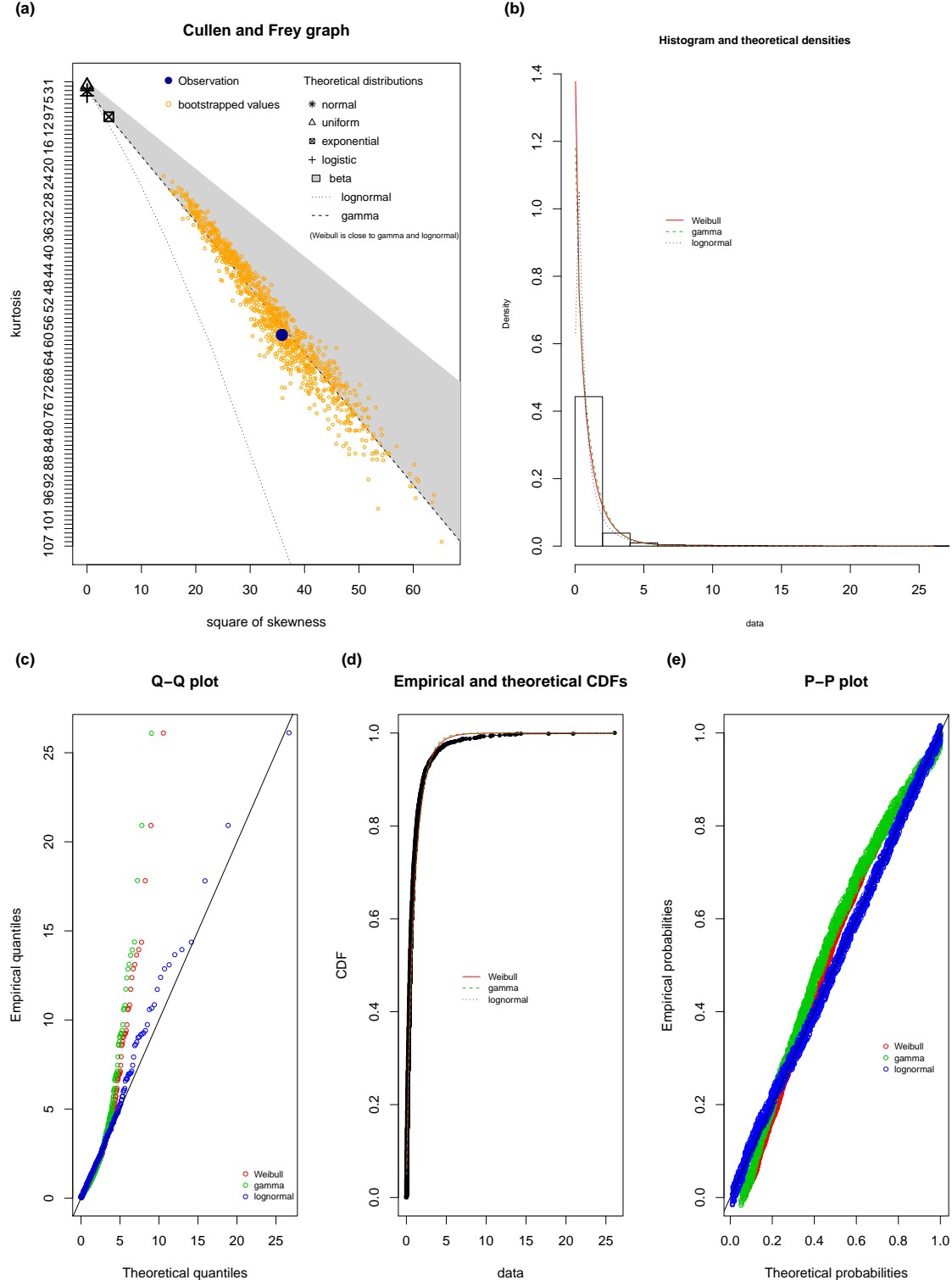

**Figure F2. Choice of distribution for lift impulses**

**Table F1.** Fitted Distribution statistics

**Drag Impulses - Statistics for fitted distributions**

Goodness-of-fit statistics

|  | Weibull | gamma | Lnorm |
|---|---|---|---|
| Kolmogorov-Smirnov statistic | 0.090 | 0.088 | 0.1084 |
| Cramer-von Mises statistic | 1.180 | 0.983 | 2.786 |
| Anderson-Darling statistic | 8.95 | 8.006 | 17.354 |

Goodness-of-fit criteria

| Akaike's Information Criterion | 1311 | 1314 | 1359 |
|---|---|---|---|
| Bayesian Information Criterion | 1321 | 1323 | 1369 |

**Lift Impulses - Statistics for fitted distributions**

Goodness-of-fit statistics

|  | Weibull | gamma | Lnorm |
|---|---|---|---|
| Kolmogorov-Smirnov statistic | 0.069 | 0.090 | 0.019 |
| Cramer-von Mises statistic | 3.571 | 6.057 | 0.093 |
| Anderson-Darling statistic | 26.4 | 34.9 | 0.9 |

Goodness-of-fit criteria

| Akaike's Information Criterion | 3968 | 4042 | 3576 |
|---|---|---|---|
| Bayesian Information Criterion | 3979 | 4054 | 3588 |

**Table F2.** Statistics for selected distributions

**Drag Impulses - Statistics for selected distribution (Gamma)**

Parametric bootstrap medians and 95% percentile CI

|  | Median | 2.5% | 97.5% |
|---|---|---|---|
| shape | 0.529 | 0.489 | 0.577 |
| rate | 0.5 | 0.463 | 0.60 |

**Lift Impulses - Statistics for selected distribution (Lognormal)**

Parametric bootstrap medians and 95% percentile CI

|  | Median | 2.5% | 97.5% |
|---|---|---|---|
| meaning | -0.663 | -0.710 | -0.614 |
| sdlog | 1.132 | 1.096 | 1.165 |

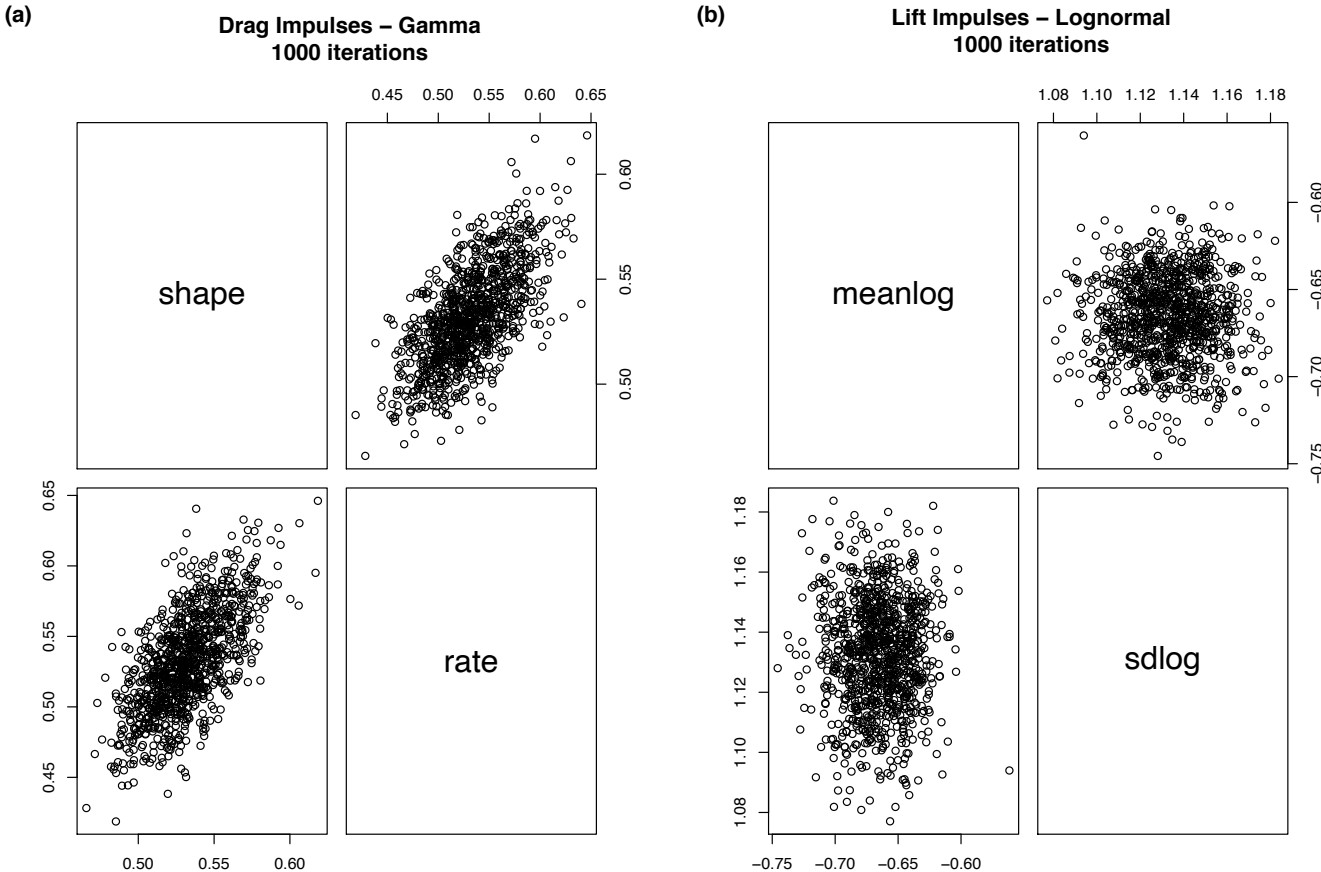

**Figure F3. Bootstrap parameters for selected distributions** The graphs quantify the stability of the selected distributions for drag and lift impulses. For the gamma distribution (drag impulses) 1000 bootstrapped parameters were cross-compared, revealing a range of 0.1 for the shape parameter and 0.2 for the rate parameter **(a)**. For the Lognormal distribution (lift impulses) 1000 bootstrapped parameters were cross-compared, revealing a range of 0.15 for the log-mean parameter and 0.14 for log standard deviation parameter **(b)**. The differences are marginal, indicating good stability of the selected distributions for the scaling of the data.

# Appendix G: Quaternions and Rotations

## G1 Summary Quaternion Algebra

Quaternions can be written in the form:

$$q = q_1 + q_2 i + q_3 j + q_4 k \tag{G1}$$

where $q_1$, $q_2$, $q_3$, $q_4$ are the components of quaternion $q$ (and $i, k, j$ unit imaginary numbers).

The quaternion conjugate is given by:

$$\bar{q} = q_1 - q_2 i - q_3 j - q_4 k \tag{G2}$$

The sum of two quaternions is then:

$$q + w = (q_1 + w_1) + (q_2 + w_2)i + (q_3 + w_3)j + (q_4 + w_4)k \tag{G3}$$

and quaternion multiplication is defined as:

$$
\begin{aligned}
q \otimes w &= (q_1 w_1 - q_2 w_2 - q_3 w_3 - q_4 w_4) + (q_1 w_2 + q_2 w_1 + q_3 w_4 - q_4 w_3)i \\
&\quad + (q_1 w_3 - q_2 w_4 + q_3 w_1 + q_4 w_2)j + (q_1 w_4 + q_2 w_3 - q_3 w_2 + q_4 w_1)k.
\end{aligned} \tag{G4}
$$

The quaternion norm is therefore defined by:

$$n(q) = \sqrt{\bar{q}q} = \sqrt{q_2^2 + q_1^2 + q_3^2 + q_4^2} \tag{G5}$$

With little manipulation, the quaternions can be directly related to four-element vectors.

Quaternions can be interpreted as a scalar plus a vector by writing:

$$q = q_1 + q_2 i + q_3 j + q_4 k = (s, \hat{v}) \tag{G6}$$

where $s = q_1$ and $\hat{v} = q_2 i + q_3 j + q_4 k$. In this notation, quaternion multiplication has the form:

$$
\begin{aligned}
\quad q_1 \bigotimes q_2 \quad &= \quad (s_1, \hat{v}_1) \cdot (s_2, \hat{v}_2) \\
&= (s_1 s_2 - \hat{v}_1 \cdot \hat{v}_2, s_1 \hat{v}_2 + s_2 \hat{v}_1 + \hat{v}_1 \cdot \hat{v}_2)
\end{aligned} \tag{G7}
$$

Finally, the rotation about the unit vector $\hat{n}$ by an angle $\theta$ can be computed using the quaternion:

$$830 \quad q = (s, v) = (\cos(\frac{1}{2}\theta), \hat{n}\sin(\frac{1}{2}\theta)) \tag{G8}$$

The components of this quaternion are called Euler parameters. After rotation, a point $p = (0, p)$ is then given by:

$$p' = qpq^{-1} = qp\bar{q} \tag{G9}$$

since $n(q) = 1$.

A concatenation of two rotations, first $q_1$ and then $q_2$, can be computed using the identity:

$$q_2(q_1 p \bar{q}_1)\bar{q}_2 = (q_2 q_1)p(\bar{q}_1 \bar{q}_2) = (q_2 q_1)p\overline{q_2 q_1} \tag{G10}$$

Finally, the transformation that gives the equivalent DCM for a quaternion $q = q_1 + q_2 i + q_3 j + q_4 k$, is given by:

$$
\begin{aligned}
R(q) \quad = \quad & [q_1^2 + q_2^2 - q_3^2 - q_4^2, 2(q_2q_3 - q_4q_1), 2(q_2q_4 + q_3q_1) \\
& 2(q_2q_3 + q_4q_1), q_1^2 - q_2^2 + q_3^2 - q_4^2, 2(q_3q_4 - q_2q_1) \\
& 2(q_2q_4 - q_3q_1), 2(q_3q_4 + q_2q_1), q_1^2 - q_2^2 - q_3^2 + q_4^2]
\end{aligned}
\qquad \text{(G11)}
$$

## G2    The Gimbal lock (adapted from Maniatis, 2016)

To demonstrate the advantage of quaternions we rotate randomly the static vector of gravity. In an orthogonal Cartesian frame where the origin of the z-axis is the centre of the Earth, gravity is measured as $[G_x, G_y, G_z] = [0, 0, 9.81]$ m.sec$^{-2}$. If we assume a rigid body rotating freely and randomly in this frame we can do the rotation calculations. Avoiding further mathematisation, the series of the calculations is the following:

  – Randomisation of the body frame angular velocities of the rigid body $\omega_x, \omega_y, \omega_z$ in a $[-2\pi, 2\pi]$ range. A frequency of 100 Hz is used.

  – Calculation of successive quaternions using direct multiplication for random angular velocities.

  – Calculate the Direction Cosine Matrix from Euler angles.

  – Rotate the vector of gravity in the body frame of the rigid body using both of the Direction Cosine Matrix using common matrix vector multiplication.

The vector expressed in the body frame is shown in Figure G1. The results are different and Gimbal lock (an inconsistent axis change when the second rotation approaches $\pm\pi/2$) occurs after the $450^{th}$ iteration which corresponds to 8 sec in simulation time.

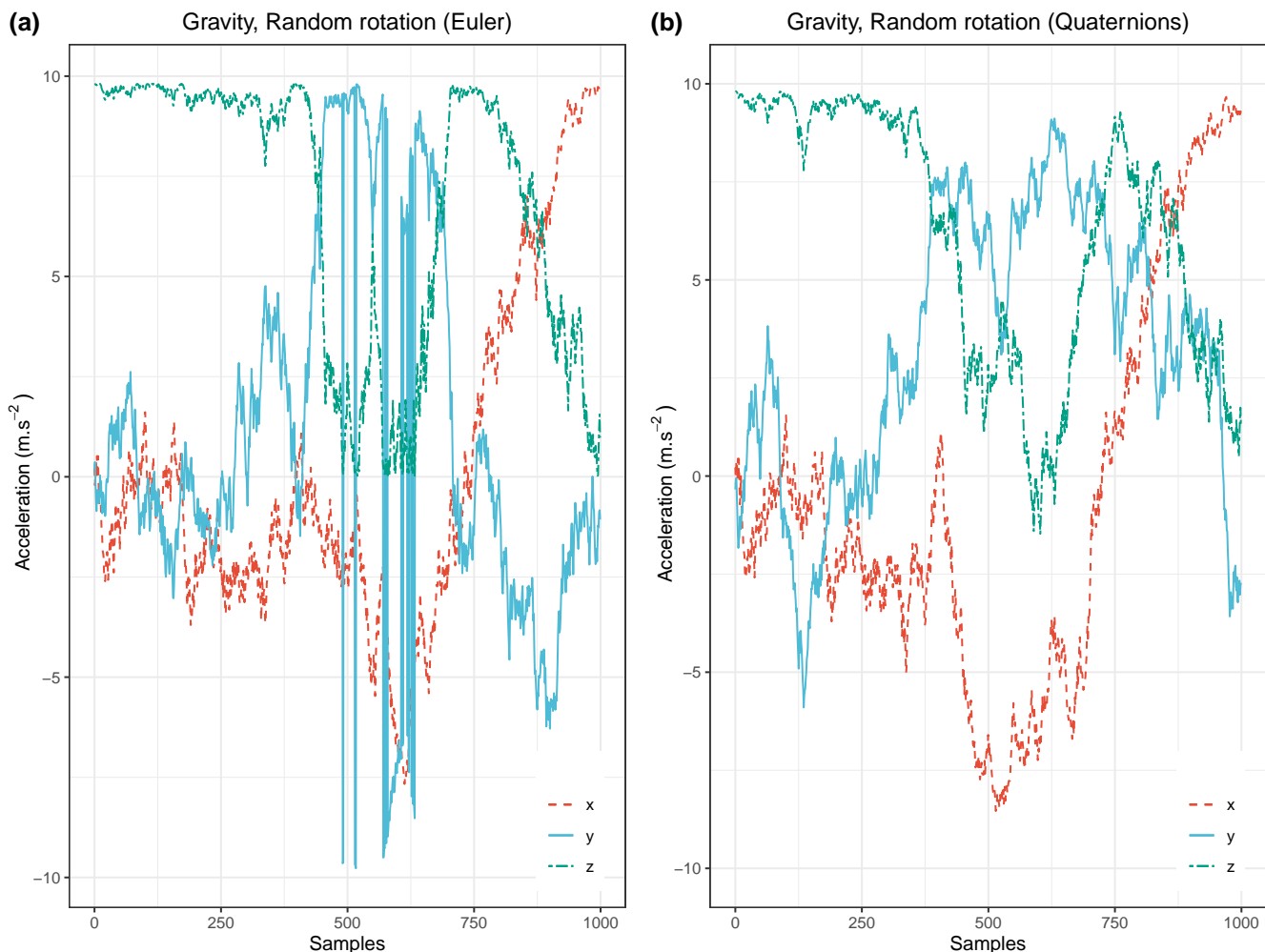

**Figure G1. Random rotation of the static vector of Gravity.** $[g_x, g_y, g_z] = [0, 0, 9.81]$ m.s$^{-2}$ in the gravity frame of reference as expressed in the body frame of randomly rotating rigid body ($dt = 0.01$ sec). **(a)** demonstrates the rotation calculations with the usage of Euler angles. Gimbal lock occurs after the $400$ iterations. **(b)** shows the same rotation series calculated via quaternions. No Gimbal lock occurs and the result is easy to interpret as it is based on the use of the measured body frame angular velocities.