# Peer review of "Inertial drag and lift forces for coarse grains on rough alluvial beds measured using in-grain accelerometers"

_Earth Surface Dynamics, 2020_

## Referee Comment (RC1) · Anonymous Referee #1 · 11 May 2020

The manuscript "Inertial drag and lift forces for coarse grains on rough alluvial bed" presents an interesting method for calculating forces acting on grains in motion in mountain rivers. I think the topic and approach is fascinating as has potential to greatly improve our understanding of transport and flow. My suggestions for improvement mainly focus on (1) explaining more of the conceptual physics and assumptions, (2) simplifying some of the writing, (3) explaining a little more how the flow conditions are perhaps not typical of gravel transport in mountain rivers, and (4) changing figure 1 to illustrate the actual experimental setup of the experiments.

I appreciate the physics framework that the paper is written in, but as a non-physicist I think it would help to have a little more explanation of some of the underlying conceptual ideas (translation into geologese?). Here is an example from the very beginning: To

me, the title clearly claims that it explores drag and lift forces for particles ON the bed. After puzzling through it and trying to think through the physics, I don't think this is correct. The methods proposed only apply to forces acting on a particle when it is in active transport, not when it is on the bed. Maybe to a physicist, having "inertial" in the title would make it obvious that the grain isn't actually resting on the bed, but that wasn't the case for me. I am still not sure what the "inertial" part is supposed to tell me. I don't think this is just a matter of semantics; it's a matter of communicating clearly to the geoscience community. Many papers have been written (and that the authors appropriately cite) using force sensors that measure drag and lift forces for grains on the bed. This paper is doing something different, which is unique and fascinating-to measure forces on grains while they are in motion–but those differences should be made clear. I suggest adding a paragraph or whatever describing forces on particles from the flow when the particle is stable on the bed, when it is moving but still has its mass partially supported by contact with the bed, and when it is fully entrained by the flow, and what the accelerometer-based calculations actually measure. Is the way to describe it that the measurements are not forces acting on the particle, but the net force that the particle responds to, from the combination of fluid and solid contacts along the particle boundary?

I think that simplifying the language in many places could also make the manuscript easier to read and understand. As part of this I suggest the authors take the time to go through it again and get rid of some superfluous words (such as "proven" in the second line of the abstract, maybe "regime" in the first line, and other places throughout. "has historically been proven" could also be replaced by "is").

I think the manuscript would be more clear if the authors define drag and lift forces. It helped my understanding to look up physics definitions: drag is simply force parallel to the direction of fluid motion, and lift is force perpendicular to the direction of fluid motion. I did not realize that drag and lift were defined so simply (thanks, I learned something new). I suppose these are sort of defined on line 192, but this is not nearly obvious

enough; I did not realize that these were definitions. In particular for lift, I thought it implied differential pressure on the top or the bottom of particles from different velocities (i.e. Bernoulli), and I was confused because it seemed like the authors could not know this since they have no measurements of fluid velocities or pressures. I realize I was wrong, but my point is that drag and lift are terms that bring preconceived ideas about the flow.

I couldn't quite figure out from the equations whether the water surface slope is accounted for; i.e. are the lift forces the authors calculate actually perpendicular to the mean flow direction, or are they instead parallel to the vector of gravitational acceleration? Figure 1 indicates that they are perpendicular/parallel to mean flow, not horizontal or vertical. Is a small angle approximation required, or embedded in the equations?

Another point that I think needs to be discussed, and limitations explicitly pointed out, is that the experimental conditions of the flume experiments are not representative of typical gravel transport Table B1 says that the flow depth was 0.1 m, which is essentially the same as the sphere diameter of 0.09 m. Grains the same size as flow depths are relevant to boulder transport, for example. I can accept the conditions as being informative even if not typical for gravel-bed systems, but it does mean that many of the scientific results, like the relative importance of lift and drag forces, may not be more broadly applicable. The Shields stress is very low for typical thresholds of motion (0.013), which my guess is related to the high protrusion and fact that the grain blocks basically all of the flow depth. Another factor in the flume experiments is probably that the hemispheres mounted on the flume bed only spanned a length of 0.5 m, which means that the boundary layer velocity profile was not anywhere close to developed. Somewhere, the paper should say where within this distance the test particle was placed (i.e., what was the distance between the upstream edge of hemispheres and the starting position of the test particle?). Similarly it is unclear whether the result of lift and drag becoming uncorrelated from the lower flow (flume) to the higher flow (field) conditions represents the difference in flow depth relative to grain size or the difference

in flow intensity. Also would be helpful for Table B2 to have a calculation of tau* for comparison.

I am a little perplexed by the measured particle protrusion of 0.045 m, i.e. half of the grain diameter, of both the experimental sphere and also the hemispheres—this means that the test particle was basically resting on the flume bed (or very close to it) in addition to resting on four surrounding hemispheres, right? And it also means that the hemispheres were not spaced as closely as possible, but spread out in order to allow the test grain to be down low, right?

Figure 1 is a conceptual diagram that does not match the actual experimental design. This should be changed so that Figure 1 is drawn to match the experimental conditions. The experimental grain should be on a bed of hemispheres, not full spheres, the spheres should cover a length of 0.5 m (so I guess 5 hemispheres?), and the spacing and placement of the test grains should be appropriate so that it reflects the actual grain protrusion. Table b1 says the protrusion is half of the particle diameter, which means the particle was essentially resting on the flume bed? I also suggest indicating the approximate water surface on the figure, since it is basically at the top of the test particle.

Table b1: was the protrusion the same for the ellipsoid?

Line 25: suggest simplifying wording of "multi-variate two-phase flow defined by a range of interacting complex subprocesses. . ."

Line 34: change "it's" to its

Line 85: I'm not sure its useful for the authors' to give their opinion that not being able to measure position has significantly limited the IMU use. That may be true but there isn't any way to know if that is the main reason, and does it matter? The current paper doesn't solve this problem, it presents a new way to use the devices for another problem. Also, I think the tone of this paragraph is unnecessarily dismissive of previous

work done using similar devices. I don't know what "best considered to be preliminary" is trying to say, other than to belittle this work. The authors come across as arrogant. To me, these works show that there is a benefit and potential of pulling different kinds of information from instrumented particles.

Stylistically, I suggest combining lines 114-128 into one paragraph; I think it is better than having four very short paragraphs.

I confess that I did not check and verify the equations.

Lines 199, 202: I think the critical drag and lift force equations should have their own equation numbers, even though they are simple equations. They are important for understanding the analyses, and it was annoying to have to go back and hunt for in-line equations.

215: I think this point should be made more prominently, and explained, earlier in the paper: that the fluid is not being measured. When I got to this point I became confused; I was still assuming that the paper would compare fluid measurements and forces on the grains, and it did not make sense that the fluid part was ignored.

226: Why 50 hz? More to the point, is there a physical argument that this sampling rate is sufficiently fast to capture the forces acting on the grain? In any case, the authors need to explain why this sampling rate was chosen. Would a slower or faster rate also work?

233: 0.5 m long is very short relative to lengths needed to develop boundary layers (i.e. velocity profiles). How much distance was there between the test particle and the upstream end of the hemispheres? I presume that the upstream flume surface was planar other than the 1.5 mm sand? It seems to me that the possible effects of this this should be addressed or at least acknowledged somewhere.

245: Give a little more explanation than just "section 4" for where these numbers came from. (in the caption to fig2 it says it used equations 9 and 10).

252: Give citations and a more complete explanation of why entrainment is defined by this displacement.

265 or around: I realize the slope is in table b2 but somewhere here say the key numbers, especially slope of 0.1 and flow depth of 0.15 m. Also describe something about the pocket or pockets the grain was placed in—the text says on a step, but explain what that means. Resting on 3 or 4 grains of similar size?

Figure 2: in panel e, I think the Flcr and Fdcr lines are flipped. In panel a, the part labeled "vibration" really represents grain transport and rotation, right? See also line 291. To me vibration implies a grain wobbling or rocking back and forth in place. How would you define grain vibration as similar or different from grain transport? Is grain vibration just being defined as just transport over a distance less than one diameter? A grain rotating as it is transported?

It seems a little surprising to me that at the end of the paper, the analysis and argument is made that lift forces are more important to entrainment than drag forces, but in Figure 2 panel A and D the drag forces clearly exceed their respective thresholds much more than do the lift forces. I don't know if it is just these examples that are shown or that I'm interpreting something wrong.

300: I'm not sure what "scale difference" means. It seems to me that the main difference is that in the flume experiment the particle was resting stably prior to entrainment, and this entrainment was analyzed. In the field case, the particle was just always and fully transported, and so the data just represent particle transport over a rough bed without any data on the particles going from resting to mobile. Also the hydraulic conditions were different.

303: Explain how you know that the distributions are heavy tailed. I presume this is coming from the Weibull, gamma, lognormal fits presented later in figures D1 and D2? If you're going to say heavy tailed in the results section you have to explain it there. You could just remove this mention at this location I think without losing any understanding.

Figure 3b, right panel: the blue line (lift) looks like an odd or poor fit to the data, because there is a whole cluster of blue data points well above the line. Are there a lot of blue points hidden by the red points? Plot the data in some way that data points are not hidden, such as smaller symbols or open symbols.

335: reword "the on the"

350: I realize that this is not the subject of this paper, and I am not sure the authors have enough data to really figure this out, but it would be interesting to know how well cumulative impulse from a given hop scales with transport distance of that hop.

360: "Extended analysis" is not a very informative section title; suggest changing to something different.

368: "of of"

A final point that I think the authors should acknowledge is that these calculations are untested. They do not know how correct these force measurements actually are. I think that is fine as long as it is stated; I would suggest saying that future work should explore and try to validate the accuracy of these measurements in some way.

---

## Short Comment (SC1) · 15 May 2020

Having designed and conducted similar experiments and performed analysis of instrumented particles' data, I have read this paper with interest. It is interesting to see an attempt for field implementation even though not a comprehensive one. There are number of comments that follow below which may be useful for the authors to consider: • The resisting forces (FDcr and FLcr) are not fixed nor are equal to the initial resisting force (which can vary significantly) while the particle is transported (as has even been shown for the case of incipient entrainment) [1]. • Lines 165-200: the authors attempted calculating the drag and lift forces as derived from the total force acting on the particle which is in turn estimated from the particle's accelerometer. Particle's acceleration can result from combinations of drag and lift forces [2] therefore the authors

claim that the drag and lift forces (or respective impulses) can be calculated from the accelerometer's readings, is not valid. • Lines 210-215: The authors' claim that impulses can be calculated directly from particle's motion (sensor's readings) is not valid, as according to Valyrakis et al. [3] and Celik et al. [4] the flow impulses (or energetic flow events) impart momentum (or energy respectively) for a particle's motion at a certain efficiency (depending on the characteristics of the flow structure driving the particle's motion). Thus, the impulses the authors refer to are not flow impulses according to the theories being cited [5,6,2]. It would be interesting to have flow hydrodynamic measurements so as to enable comparison of the inertial impulses the authors estimate with flow impulses. • Details around the flow conditions in the controlled flume experiments are missing. In particular: o The flow seems to be non-uniform because of the locally raised bed where the particle rests and also the presence of a smooth bed upstream this section combined with the short length of the raised bed render the flow not fully developed. o The flow depth and the range of flow conditions tested are not mentioned; this is even more important if the flow depth is of the same order of the particle's size, as in this case the particle may interact with the free water surface and the mechanics of entrainment are different from what the traditional hydraulic literature on incipient motion is discussing. o The authors do not measure any flow hydrodynamics that could be linked to the sensor's metrics they present. Bed shear stress which is based on the bed surface slope is mentioned but it is not commented on how bed slope value was obtained (measured or estimated and how). o For the ellipsoid there is a strong effect of the orientation of the initial placement on the dominance of the forces and the resulting mode of entrainment. More emphasis on this dependency could be discussed in this works. • For the field work there is no comprehensive description of the flow and bed surface characteristics over which the particle is being transported.

1. Valyrakis et al. "Incipient rolling of coarse particles in water flows: a dynamical perspective," in Proc. Riverflow, Braunschweig, Germany, June 2010, pp.769-776. 2. Valyrakis et al., "Role of instantaneous force magnitude and duration on particle entrainment," JGR: Earth Surface, vol. 115, no. F2, pp. 1-18, Apr. 2010. 3. Valyrakis et

al., "Entrainment of coarse particles in turbulent flows: An energy approach," JGR, vol. 118, no. 1, pp. 42-5, Jan. 2013. 4. Celik et al.," Instantaneous pressure measurements on a spherical grain under threshold flow conditions," J. Fluid Mech., vol. 741, pp. 60-97, Feb. 2014. 5. Diplas, P. et al., "The role of impulse on the initiation of particle movement under turbulent flow conditions," Science, vol. 322, no. 5902, pp. 717-720, Oct. 2008. 6. Celik et al., "Impulse and particle dislodgement under turbulent flow conditions," Phys. Fluids, vol. 22, pp. 1-13, Apr. 2010.

---

## Short Comment (SC2) · 15 May 2020

I want to thank the commenter for his contribution. I just wish this came a bit earlier in the discussion timeline so we can have a truly interactive discussion. If he doesn't get the chance to reply to my comment before the end of the discussion period, I commit to take into account his comments and include them in the review process.

My general response is that the forces captured by the accelerometer (which are not described in the literature of hydraulics and need special treatment, O'Reilly, 2008) are not the forces (or impulses) described in the works cited by the author to support his arguments. They relate but they are not the same (Lines 212-216 of the paper under review). This is why we discuss specifically the difference between inertial impulses

(as derived by the accelerometer) and hydraulic impulses (as defined in the works quantifying the response of the particle to the flow). Our work relates more to particle force measurements ( e.g. Schmeeckle et al., 2007; Lamb et al., 2017) rather than the quantification of flow turbulence. Overall, the concept of impulse is general (force over time), so the definition of forces becomes crucial for the interpretations of relevant results.

For the above reasons, many of the comments below are not valid. We define the forces in a different context and from a different frame of reference. I decided to answer those comments quickly because they imply significant misunderstandings regarding the physics of accelerometers (and inertial sensors in general). However, this discussion is definitely not over. This is a quite technical issue.

More specifically Comment 1: The resisting forces (FDcr and FLcr) are not fixed nor are equal to the initial resisting force (which can vary significantly) while the particle is transported (as has even been shown for the case of incipient entrainment) [1].

Response: That is true, the forces acting on the particle are neither fixed nor equal to the initial position. However, after calculating successive orientations (using the angular velocities) they can be transformed to a static frame (frame r in the paper) and this is what we present (sections 2 and 3 of the paper).

Comment 2: Lines 210-215: The authors' claim that impulses can be calculated directly from particle's motion (sensor's readings) is not valid, as according to Valyrakis et al. [3] and Celik et al. [4] the flow impulses (or energetic flow events) impart momentum (or energy respectively) for a particle's motion at a certain efficiency (depending on the characteristics of the flow structure driving the particle's motion). Thus, the impulses the authors refer to are not flow impulses according to the theories being cited [5,6,2].

Response: The commenter's conclusion here is correct. This paper doesn't quantify the same impulses to the works he cites (Lines 210 to 216 of the paper).

Comment 3: It would be interesting to have flow hydrodynamic measurements so as to enable comparison of the inertial impulses the authors estimate with flow impulses

Response: Here it is clear to the commenter that inertial impulses are different to the flow impulses. This contradicts significantly his comments above. However, it is true that this link is important. I am looking forward to reading the commenter's contribution from the experiments he conducts.

Comment 4: Details around the flow conditions in the controlled flume experiments are missing. In particular: The flow seems to be non-uniform because of the locally raised bed where the particle rests and also the presence of a smooth bed upstream this section combined with the short length of the raised bed render the flow not fully developed.

Response: There are hydraulic measurements presented in the appendices. We didn't have the capacity for detailed flow measurements, but that there are no physics to suggest that this affects the accelerometer model we present and the measurements for the conditions we captured. Here it is useful to look at the comments from Reviewer 1 who mentions explicitly that the conditions are closer to boulder motion rather than gravel. This is a very useful observation which doesn't affect the calculations but their interpretation. And I agree that more work is needed on that front and repetition of the experiments under varied conditions.

Comment 5: The flow depth and the range of flow conditions tested are not mentioned; this is even more important if the flow depth is of the same order of the particle's size, as in this case the particle may interact with the free water surface and the mechanics of entrainment are different from what the traditional hydraulic literature on incipient motion is discussing.

Response: This is not true. Firstly, we didn't' test a range of flow conditions in the lab, we repeated one experiment 12 times (the hydraulics are presented in the appendices). Secondly, the mechanics of entrainment we present are exactly the same to

the literature (Lines 169-174 in the paper). They are just linked to an accelerometer model and rotated to a different frame of reference in order to make the accelerometer measurement comprehensive.

Comment 6: The authors do not measure any flow hydrodynamics that could be linked to the sensor's metrics they present. Bed shear stress which is based on the bed surface slope is mentioned but it is not commented on how bed slope value was obtained (measured or estimated and how)

Response: The slope was measured in the flume and estimated in the field (and that is documented in the paper). But I will insist that this has nothing to do with the validity of the definitions and the measurements as the commenter argues from the start of this commentary.

Comment 7: For the ellipsoid there is a strong effect of the orientation of the initial placement on the dominance of the forces and the resulting mode of entrainment. More emphasis on this dependency could be discussed in this works.

Response: I apologise for the repetition: There will be a big effect on the numbers derived under different orientations, but the same (or a similar) model should be applied. And the model we present accounts for the orientation specifically since we can measure it directly (quaternions).

Comment 8: For the field work there is no comprehensive description of the flow and bed surface characteristics over which the particle is being transported.

Response: It is not easy (or even possible sometimes) to take detailed flow measurements in shallow streams. For the stream we conducted the experiments (Erlenbach) there are numerous references in the literature where the commenter can find a lot of details about the topography and the bathymetry. We just placed the sensor on a plain bedrock and the conditions were typical of a riffle and pool setting. We also remove the first second from the measurements to minimise the effect of the local topogra-

phy, we are interested into the forces during transport. For the purpose of this paper (demonstrating the calculation of inertial impulses) the slope and the shear stress should suffice for an understanding of the hydraulic forcing.

References

O'Reilly, O. M.: Intermediate dynamics for engineers: a unified treatment of Newton-Euler and Lagrangian mechanics, AMC, 10, 12, 2008.

Schmeeckle, M. W., Nelson, J. M., and Shreve, R. L.: Forces on stationary particles in near-bed turbulent flows, Journal of Geophysical Research: Earth Surface (2003–2012), 112, https://doi.org/10.1029/2006JF000536, 2007.

Lamb, M. P., Brun, F., and Fuller, B. M.: Direct measurements of lift and drag on shallowly submerged cobbles in steep streams: Implications for flow resistance and sediment transport, Water Resources Research, 53, 7607–7629, 2017.

---

## Short Comment (SC3) · 15 May 2020

Many thanks Maniatis for the response. Apologies for late reply due to being busy with other involvements. Please find my reply below to your response marked as reply.

More specifically Comment 1: The resisting forces (FDcr and FLcr) are not fixed nor are equal to the initial resisting force (which can vary significantly even for small changes in particle orientation) while the particle is transported (as has even been shown for the case of incipient entrainment) [1].

Response: That is true, the forces acting on the particle are neither fixed nor equal to the initial position. However, after calculating successive orientations (using the angular velocities) they can be transformed to a static frame (frame r in the paper) and

this is what we present (sections 2 and 3 of the paper).

Reply: Even though the authors discuss in section 2 and 3 the frame conversion and its application (only for the acceleration data), they have not discussed its application in changing the critical force based on the particle's ordination as they claim to be doing above. At the same time, as is clear from their text and figure 2, FDcr and FLcr are kept fixed and unchanged, regardless of the particle's orientation, which invalidates their above claim (simply, the critical forces are not shown to be transformed into frame r as the authors claim above - this is also clearly shown in figure 2).

Comment 2: Lines 210-215: The authors' claim that impulses can be calculated directly from particle's motion (sensor's readings) is not valid, as according to Valyrakis et al. [3] and Celik et al. [4] the flow impulses (or energetic flow events) impart momentum (or energy respectively) for a particle's motion at a certain efficiency (depending on the characteristics of the flow structure driving the particle's motion). Thus, the impulses the authors refer to are not flow impulses according to the theories being cited [5,6,2]. Response: The commenter's conclusion here is correct. This paper doesn't quantify the same impulses to the works he cites (Lines 210 to 216 of the paper).

Comment 3: It would be interesting to have flow hydrodynamic measurements so as to enable comparison of the inertial impulses the authors estimate with flow impulses.

Response: Here it is clear to the commenter that inertial impulses are different to the flow impulses. This contradicts significantly his comments above. However, it is true that this link is important. I am looking forward to reading the commenter's contribution from the experiments he conducts.

Reply to comments 2/3 above: These comments are not contradictory rather they are intended to promote clarity for the presentation of the author's intended contribution. The authors here have significant fallacy in both their understanding of the framework they are presenting and their calculations which is at the crux of their analysis - which is best demonstrated in reference to figure 2-a: The FLr and FDr derived from the

accelerometer's data refer to the total forces acting on the particle so the thresholding with and assumed (fixed) critical force is meaningless because the resultant force is the vector sum of the driving hydrodynamic forces (which are here unknown) and critical (resistance) forces which are also unknown (and both are wildly fluctuating) during transport. Simply, this type of thresholding has a questionable value (or relevance) to flow induced transport processes of solids. Even if (this is just a gross mistake and) the author intends to remove the thresholding, the physical relevance of the inertial impulses within the context of sediment transport or incipient entrainment is completely missing and would need be discussed.

Comment 4: Details around the flow conditions in the controlled flume experiments are missing. In particular: The flow seems to be non-uniform because of the locally raised bed where the particle rests and also the presence of a smooth bed upstream this section combined with the short length of the raised bed, render the flow not fully developed.

Response: There are hydraulic measurements presented in the appendices. We didn't have the capacity for detailed flow measurements, but that there are no physics to suggest that this affects the accelerometer model we present and the measurements for the conditions we captured. Here it is useful to look at the comments from Reviewer 1 who mentions explicitly that the conditions are closer to boulder motion rather than gravel. This is a very useful observation which doesn't affect the calculations but their interpretation. And I agree that more work is needed on that front and repetition of the experiments under varied conditions.

Reply: line 254: are the experiments shown herein the same or different to Maniatis 2017? The test bed around which the particle is positioned is not described in any detail: this is crucially important as it interrelates to the particle's transport once entrainment has initiated. For example, if the raised bed has limited length (<2m), to which the author refer to as the minimum transport distance, the entrainment processes described herein are more relevant to a particle falling from the raised bed

rather than being transported over plain bedrock, as described in the manuscript. Also, the presence of rough or smooth bed upstream of the significantly raised microtopography would involve the generation of statistically different flow structures compared to those acting on the particle for its transport, which renders these experiments not relevant to the body of work found in the traditional turbulence induced particle incipient entrainment literature, commonly referenced in this manuscript.

Comment 5: The flow depth and the range of flow conditions tested are not mentioned; this is even more important if the flow depth is of the same order of the particle's size, as in this case the particle may interact with the free water surface and the mechanics of entrainment are different from what the traditional hydraulic literature on incipient motion is discussing.

Response: This is not true. Firstly, we didn't' test a range of flow conditions in the lab, we repeated one experiment 12 times (the hydraulics are presented in the appendices). Secondly, the mechanics of entrainment we present are exactly the same to the literature (Lines 169-174 in the paper). They are just linked to an accelerometer model and rotated to a different frame of reference in order to make the accelerometer measurement comprehensive.

Reply: still the author doesn't for some reason offer the flow depth at the critical flow conditions. (Just to clarify that the comment offered above, inquiries about the range of flows assessed at the lab, which indeed have been tested, as the authors comments-via implementing a rising hydrograph- so it is not clear why the author disagrees). Also the authors in their manuscript describe 10 out of 12 measurements mentioned above, what were the reasons to discard two of the measurements? Are the authors showing the (eg aggregate) results for 10, 12 or just one of the experiments? Again, the experiments described in this manuscript are not relevant to the typical incipient motion literature, as the author agrees with the previous reviewer's comment: these are more relevant to boulder transport processes, rather turbulence induced transport of coarse particles (as a reader might wrongly infer by just reading the article's title).

Comment 6: The authors do not measure any flow hydrodynamics that could be linked to the sensor's metrics they present. Bed shear stress which is based on the bed surface slope is mentioned but it is not commented on how bed slope value was obtained (measured or estimated and how) Response: The slope was measured in the flume and estimated in the field (and that is documented in the paper). But I will insist that this has nothing to do with the validity of the definitions and the measurements as the commenter argues from the start of this commentary.

Reply: Could the author detail how the slope of the flume was measured? (if the experiments were conducted at the 0.9 m wide flume of the University of Glasgow, which I am also using, the maximum bed surface of the flume, which I have measured, cannot reach the mentioned slope of 0.02 as claimed (!) (line 230).

Comment 7: For the ellipsoid there is a strong effect of the orientation of the initial placement on the dominance of the forces and the resulting mode of entrainment. More emphasis on this dependency could be discussed in this works.

Response: I apologise for the repetition: There will be a big effect on the numbers derived under different orientations, but the same (or a similar) model should be applied. And the model we present accounts for the orientation specifically since we can measure it directly (quaternions).

Reply: the authors have misunderstood my commentary. I am not discussing whether their method can be applied to different orienations (which could be done using quaternions or Eulerian angles.. etc), they simply do not discuss the dependency of the initial orientation of ellipsoid on the resisting forces (and subsequently the derived impulses). It will also be of interest to elaborate if the application in the field is intended with a given initial orientation or any will do and why? Also, the authors choose to use a very expensive accelerometer of up to 400g, while only recording at a frequency of 50Hz, can the authors discuss the utility of such a design choice (why not use a smaller range for the accelerometer which has less cost?). Also, what is the advantage of recording

at such a high acceleration range over such a relatively low frequency (one would expect the high range of acceleration recordings -if indeed needed- to be matched by a high frequency of recordings too (eg 500 Hz or more)?

Comment 8: For the field work there is no comprehensive description of the flow and bed surface characteristics over which the particle is being transported.

Response: It is not easy (or even possible sometimes) to take detailed flow measurements in shallow streams. For the stream we conducted the experiments (Erlenbach) there are numerous references in the literature where the commenter can find a lot of details about the topography and the bathymetry. We just placed the sensor on a plain bedrock and the conditions were typical of a riffle and pool setting. We also remove the first second from the measurements to minimise the effect of the local topography, we are interested into the forces during transport. For the purpose of this paper (demonstrating the calculation of inertial impulses) the slope and the shear stress should suffice for an understanding of the hydraulic forcing.

Reply: it is appreciated this is not easy to do, but the resisting forces during incipient entrainment as well as transport are significantly dependant on the initial resting microtopogrpahy and bed topography over which the particle will be advected. Also, FDcr and FLcr in the beginning and during transport will depend on these features, which make them important to be detailed. For example, was the particle initially positioned at a plane bedrock or a riffle-pool topography (which part specifically)? Is the transport taking place at the flat bedrock (or pool?) or over the drop from the riffle? In this case, what is the relevance of the inertial impulses to the plane of the bedrock slope and bed shear stresses which the authors claim above to be relevant to the transport processes?

New comments:

It is not clear if the field test the authors discuss truly refer to incipient motion conditions or full transport processes; for example for how long was the particle immobile under

the free flow, before being entrained? Is the framework the authors intend discussing refer to incipient entrainment (described in the lab) or transport processes (apparently relevant to the field work). Usually experiments are used for helping calibrate conditions in the field: what is the practical utility of the authors' lab experiments? What is learnt from this sensor in a quantifiable manner? And how does it physically relate to the processes and past literature of sediment transport? Given that the resistance (highly dependent on the resting microtopograpy) in the lab experiments and the field tests is different (or not measured exactly in the field), then what is the value of normalising the inertial impulses from the field with the impulses from the lab? Also, for the lab experiments: was the particle stopped because the flow was not able to push it further or it is stopped because it reached near or at the tailgate? To that goal, can you comment on the distance of the test section from the tailgate (ie how much bigger than 2 plus meters is it?).

---

## Referee Comment (RC2) · Anonymous Referee #2 · 19 May 2020

The manuscript by Maniatis is based on a study that uses two custom-built smart particles (one spherical and one elliptical in shape) that contain an accelerometer and gyroscope. The manuscript focuses on (1) presenting a method for putting the accelerations extracted in the particle's frame of reference to one from the perspective of a fixed observer and (2) presenting the lift and drag forces measured while the particle was in motion in a laboratory and field experiment.

The paper's strengths are the presentation of the mapping method between the two frames of reference and the introduction of the idea of inertial drag and lift impulses. The inertial drag and lift impulse are similar to the impulse idea put forward by Diplas et al. 2008 (Impulse = integral of force with respect to time applied to a particle), but is based on particle forces inferred from particle motion rather than particle forces inferred

from fluid measurements.

I believe the paper presents (1) useful methodology information for those using accelerometer data in particles, and (2) useful data pertaining to particle motion. At the same time, I have a few questions for the authors and some suggestions on the presentation of the work that I think should be addressed.

**Scientific questions/concerns/suggestions**

1. My primary suggestion, or concern, is that the authors make it explicitly clear that all of their force measurements (and impulse calculations) require that the particle be moving (if I am understanding things correctly). It would also be helpful if they expanded their discussion on the benefits and limitations of such measurements. Many of my presentation suggestions below reflect this desire for it to be clear that the forces measured are only those extracted from the particle accelerations.

2. It seems that you have the highest number of entrainment events for an inertial impulse of zero. Is this because the primary motivating impulse came at a time before that which could be measured by the particle?

3. Along these lines, do the potential travel paths of the particle dictate the forces measured? That is, are the $F_L$ and $F_D$ measurements sort of pre-determined by the orientation of the particle relative to others in the bed?

4. In equation 5, where is the contact force with the bed? Is it tied into the gravitational force terms? The critical drag and lift forces will depend on the submerged weight of the particle and the orientation of the grain within a pocket through the contact reaction force. The orientation of this force will also influence where it is more likely to have lift or drag dominate. How does it factor into equations 9 and 10? It should, shouldn't it?

5. Along with the preceding question, how is this contact force for accounted for throughout the range of a particle's motion as it moves and interacts with the bed?

6. Please provide a figure showing the experimental setup with flow depth and bed arrangement. I would think that $\tau_B$ has little meaning in terms of mobilization with your particular laboratory setup.

7. Terminology in Figure 2 and elsewhere. When do vibrations turn into motion? Does entrainment start when the particle reaches a distanced traveled of 1 diameter, or does entrainment start when the particle starts to move and then continues on a path that leads to it moving 1 particle diameter? Also, what is a non-entrainment event with a measured inertial impulse? Does that correspond to a case where the particle started to move out of the pocket but then fell back back down before reaching the apex?

8. In the discussion, I'd suggest non-dimensionalizing the force (maybe using submerged specific weight) when making the comparisons to other work. Also, how do your inertial forces compare to standard drag estimates using velocity, particle size, and a drag coefficient? What types of relative fluid/particle velocities are needed?

**Presentation questions/concerns/suggestions**

The paper is reasonably well written, but I do have some suggestions below that I think would help to improve the presentation of the work.

Abstract:

- L.1 Delete "been"

- L.4 Replace "on sediment" with "on a particle during"

- L.5-7 The sentence "Today, twenty years.... for the issue" is not needed in an abstract. Suggest deleting it

- L.9 Change "grains on" to "grain moving on"

- L.11 Change "resulting to the" to "resulting in a"

Introduction:

- overall I think you can shorten down the introduction

- L.38-41 I think you can remove the sentence that starts with "The term Lagrangian..." Most people know the difference between a Lagrangian and an Eulerian frame of reference

- L.44 Delete "(the exact definition of turbulence impulse)"

- L.72 Change opening sentence to: MEMS-IMU sensors ideally measure forces at the center of mass....

- L.74 change to "acting on the grain as it moves"

- L.83 suggest changing to, "...and electrical engineering. Modeling of IMU error..."

- L.91&92 suggest changing resolve and resolution to "map" and "mapping" or "rotate" and "rotation"

Frames of ref

- L.123 Suggest, "... frames is non-trivial. A widely-used method..."
- L.124 Suggest changing "to apply" to "the application of"

- L.129-end of the section. I'm torn here. I could see all of this being better suited for the appendix if the focus of the paper is on the data from the particles. However, if you want the paper to be about the mapping between the frames of reference then you should keep it here.

- L.157-158 Delete the beginning of the sentence that spans the text "As sediment" to "linear acceleration". Just start with $a_r$.

Inertial measurements

- L.211 change to $T_i$

- L.216 suggest changing "that mobilizes the particle" to "acting on the particle once it starts to move"

Lab and field experiments

- L.238 0.028 l/s is a discharge, not a rate of increase in discharge

- L.240 change "recorded" to "video of"

- Figure 4. If I'm correct, I think you reference figure 4 before referencing figure 3. It should be the other way around

- L.254 suggesting changing to, "...inertial impulses for cases were the grain started to move."

Discussion

L. 360 - I'd suggest making the extended analysis part of the discussion. Use subsection for the different components of the discussion such as comparison with other work and L.368 - delete one of the "of"s

---

## Referee Comment (RC3) · Anonymous Referee #3 · 21 May 2020

This manuscript is intended to quantify the dynamics of entrained sediment particles by using advanced inertial force monitoring systems. I believe this work has good contributions to this long-standing problem and it is worth being published. Overall, the materials are properly organized and relevant concepts explained concisely. I see no significant problems for its publication, but there are several minor suggestions for the authors to consider in their revision. Below, I listed my comments and suggestions as I read through the manuscript.

L50-54: Regarding the Lagrangian approach, Ballio et al. (2018, "Lagrangian and Eulerian description of bed load transport") provide a unified framework to describe sediment particle motion under different frames of reference and quantification methods. The authors may want to refer to this work and highlight their particular contributions to

the topic.

L58: The issue of over-prediction of transport rates was also extendedly discussed by Bunte et al. (2004, "Measurement of coarse gravel and cobble transport using portable bedload traps") and Singh et al. (2009, "Experimental evidence for statistical scaling and intermittency in sediment transport rates"), and other related papers. The reference can be updated with these contributions.

Sec. 2 (Frames of reference, rotations and IMU measurements) provides details of mathematical expressions for the conversion/transformation among different frames of reference. But Fig. 1 needs some more clarity on the specifics of these frames, e.g. (x,y,z) (rx,ry,rz),(ix,iy,iz).

Based on their formulation, Eq. (9) only concerns the threshold condition in the tractive mode towards downstream, and Eq. (10) the upward movement. Yet, when approaching the strictly critical condition, particle rolling, which has even lower resistance, can be the most predominant entrainment mode. The authors can consider adding an extra formula describing the rolling threshold for providing a more complete framework.

L234. Has the 1.5 mm uniform sand glued to surface or also movable?

Fig.2 (b) describes the change of drag forces during the noted five entrainment events. The pattern shown here, however, somewhat contradicts the impulse model mentioned later in the manuscript (e.g. L278-279). If all these five events follow exactly the impulse criterion, the events of higher magnitude should persist for a shorter duration, and vice versa, to maintain at the same impulse level. In other words, these events, very likely, represent different degrees of particle mobility and not the most extreme cases of entrainment (minimal critical impulse). I think this aspect is worth mentioning in the discussion.

Sec. 5.0.2. The calculation of entrainment probability is not clear to me. Is it described as the ratio of entrainment duration to the total observation time, or the ratio of entrainment events to the total exceedance events observed? I assume it is the latter definition. Adding an equation will help to clarify this point.

L300. The scaling effects are usually attributed to the intermittency of particle motion dictated by turbulent flow structures. Yet, this sentence seems to suggest the role of physical scale between the laboratory flume and field stream. A clarification will be helpful (see discussion in Singh et al., 2009).

Fig. 4(a) can be improved by using different shading colors for FD and FL, respectively.

L310-317. The differences in magnitude of critical drag and lift forces between this work and literature data are attributed to the particle sizes, or corresponding mass, used differently. To resolve this issue, a dimensionless quantity, e.g. FD/particle weight, FL/particle weight, can be considered for both the present work and previous reports.

L325-332. The description of the negative lift forces can be more precise in relating to the threshold of motion conditions. Specifically, when the negative lift force appears, the probability of particle lifting reduces, and also, the resistance to the tractive movement increases via the enhanced inter-surface friction.

L353-357. The description of the entrainment mode of rolling can be placed in the earlier section (see the previous comment) to avoid confusion.

---

## Author Comment (AC1) · 13 Jun 2020

Comment: The manuscript "Inertial drag and lift forces for coarse grains on rough alluvial bed" presents an interesting method for calculating forces acting on grains in motion in mountain rivers. I think the topic and approach is fascinating as has potential to greatly improve our understanding of transport and flow. My suggestions for improvement mainly focus on (1) explaining more of the conceptual physics and assumptions, (2) simplifying some of the writing, (3) explaining a little more how the flow conditions are perhaps not typical of gravel transport in mountain rivers, and (4) changing figure 1 to illustrate the actual experimental setup of the experiments.

Reply: We sincerely thank the author for the thorough review and the supportive com-

ments. We set out to address all the suggestions in a revised version. Hopefully this will become clear in our following responses.

Comment: I appreciate the physics framework that the paper is written in, but as a non-physicist I think it would help to have a little more explanation of some of the underlying conceptual ideas (translation into geologese?). Here is an example from the very beginning: To me, the title clearly claims that it explores drag and lift forces for particles ON the bed. After puzzling through it and trying to think through the physics, I don't think this is correct. The methods proposed only apply to forces acting on a particle when it is in active transport, not when it is on the bed. Maybe to a physicist, having "inertial" in the title would make it obvious that the grain isn't actually resting on the bed, but that wasn't the case for me. I am still not sure what the "inertial" part is supposed to tell me. I don't think this is just a matter of semantics; it's a matter of communicating clearly to the geoscience community. Many papers have been written (and that the authors appropriately cite) using force sensors that measure drag and lift forces for grains on the bed. This paper is doing something different, which is unique and fascinating-to measure forces on grains while they are in motion–but those differences should be made clear.

Reply: We appreciate the value of explaining the concepts using language that will be familiar to the reader. Some of the underlying concepts we present will be more familiar to people with backgrounds in hydraulics or physics, but we consciously target the audience of this journal (broadly the audience working on geomorphological and geological problems). In response to this comment we have made several revisions to the manuscript to improve its clarity.

The term "inertial" in the context we present primarily means that the reference frame we use to analyse particle forces is static, i.e. that the frame of reference doesn't move with the particle. This is not a common approach in sediment transport but is necessary to enable us to interpret the accelerometer measurements: these are meaningless if untransformed. In addition, the term "inertial" means that, in the reference frame we

use, there are no so-called fictitious forces. This observation is very straightforward for the hydraulic forces (and the forces from the surrounding particles) acting on a target particle but needs to be specifically stated for gravity because gravity is a fictitious force for a free moving accelerometer (Appendix Figure A1a).

Finally, it is very important to clarify that inertial sensors measure forces only when the particle moves. The advantage of this is that the motion of the particle is captured explicitly (the sensor solves a very complicated force balance for us). The disadvantage of this is that there is no way to "decouple" the acting forces without complementary measurements (no information about the hydraulic drag component, for example).

All the above, which overlap with comments and suggestions from the other two Reviewers, are addressed in the revised manuscript.

Comment: I suggest adding a paragraph or whatever describing forces on particles from the flow when the particle is stable on the bed, when it is moving but still has its mass partially supported by contact with the bed, and when it is fully entrained by the flow, and what the accelerometer-based calculations actually measure.

Reply: This paragraph will be added and will include the description of the rolling threshold of motion as suggested by Reviewer 3 (we will revise the structure of section 3 of the paper). As we say above, it is not possible to describe all the conditions in a quantitative manner (this would require detailed hydraulics measurements which we don't have), but we can clarify further which are the force components captured by the sensor.

Comment: Is the way to describe it that the measurements are not forces acting on the particle, but the net force that the particle responds to, from the combination of fluid and solid contacts along the particle boundary?

Reply: This is exactly the case; the only difference is that we use the term "resultant" for the force.

Comment: I think that simplifying the language in many places could also make the manuscript easier to read and understand. As part of this I suggest the authors take the time to go through it again and get rid of some superfluous words (such as "proven" in the second line of the abstract, maybe "regime" in the first line, and other places throughout. "has historically been proven" could also be replaced by "is").

Reply: We accept this comment, it is true that parts of the paper can be simplified, and we will address that in a revised version

Comment: I think the manuscript would be more clear if the authors define drag and lift forces. It helped my understanding to look up physics definitions: drag is simply force parallel to the direction of fluid motion, and lift is force perpendicular to the direction of fluid motion. I did not realize that drag and lift were defined so simply (thanks, I learned something new). I suppose these are sort of defined on line 192, but this is not nearly obvious enough; I did not realize that these were definitions. In particular for lift, I thought it implied differential pressure on the top or the bottom of particles from different velocities (i.e. Bernoulli), and I was confused because it seemed like the authors could not know this since they have no measurements of fluid velocities or pressures. I realize I was wrong, but my point is that drag and lift are terms that bring preconceived ideas about the flow.

Reply: The fact that the Reviewer had to go through that enquiry makes the clearer definition of forces necessary. We will address that in section 3 of a revised manuscript.

Comment: I couldn't quite figure out from the equations whether the water surface slope is accounted for; i.e. are the lift forces the authors calculate actually perpendicular to the mean flow direction, or are they instead parallel to the vector of gravitational acceleration? Figure 1 indicates that they are perpendicular/parallel to mean flow, not horizontal or vertical. Is a small angle approximation required, or embedded in the equations?

Reply: The resultant lift force (FL) is perpendicular to the mean flow direction. We will

address that by revising significantly Figure 1 in order to include a clearer definition of the frames of reference and more details about the flume experiments (e.g. critical depth) as suggested in both the following comments here and the comments from the other two reviewers.

Comment: Another point that I think needs to be discussed, and limitations explicitly pointed out, is that the experimental conditions of the flume experiments are not representative of typical gravel transport Table B1 says that the flow depth was 0.1 m, which is essentially the same as the sphere diameter of 0.09 m. Grains the same size as flow depths are relevant to boulder transport, for example. I can accept the conditions as being informative even if not typical for gravel-bed systems, but it does mean that many of the scientific results, like the relative importance of lift and drag forces, may not be more broadly applicable.

Reply: We want to thank the reviewer for highlighting this. It is crucial to clarify that all the experiments in the thesis of Maniatis 2016 were designed to address transport at grain diameter – flow depth ratios close to 1 (D/H approximately equal to 1) for particles sitting on top of the surrounding bed. This is common for the transport of large grains in mountain streams [Schneider et al., 2014, Lamb et al., 2017] and the results should only be interpreted in this context. In Erlenbach, the D/H ration was also approximately 1 if a sphere of equivalent volume to the ellipsoid is assumed but the ellipsoid c-axis/H ratio was close to 0.3. Although the Newton Euler model that we present is general, we have no intention to claim that the results can be generalised in different transport conditions. We will clarify that in the introduction and in all the parts of the paper we can.

Comment: The Shields stress is very low for typical thresholds of motion (0.013), which my guess is related to the high protrusion and fact that the grain blocks basically all of the flow depth. Another factor in the flume experiments is probably that the hemispheres mounted on the flume bed only spanned a length of 0.5 m, which means that the boundary layer velocity profile was not anywhere close to developed. Somewhere,

the paper should say where within this distance the test particle was placed (i.e., what was the distance between the upstream edge of hemispheres and the starting position of the test particle?).

Reply: We agree with all those comments and our response above regarding grain protrusion and flow depth applies here. We will describe in more detail the flume experiments (adapting parts of Maniatis 2016) and we will revise Figure 1 in order to include a sketch of the flume, the position of the bed of plastic hemispheres, the position of the test particle and the critical depth. The flume bed upstream of the test section was covered by gravel leading to rapid development of uniform flow for several metres upstream of this section. There was a roughness transition between the gravel bed and the test section, leading to flow acceleration. Although the flow was non-uniform, which is very representative of natural conditions, the measurement of forces by the sensor can be tested in these conditions.

Comment: Similarly, it is unclear whether the result of lift and drag becoming uncorrelated from the lower flow (flume) to the higher flow (field) conditions represents the difference in flow depth relative to grain size or the difference in flow intensity. Also, would be helpful for Table B2 to have a calculation of tau* for comparison. Reply: For the first part of the question, we observe a generally weak correlation between the resultant drag and lift forces. The highest correlation was observed for the spherical particle in the flume (R=0.44). For the ellipsoid the correlation is consistently week (from R=0.17 in the lab to R=0.018 in the Erlenbach during transport). To answer why the correlation becomes even weaker in the field we would need a series of incipient motion experiments in the field and that was not possible. For the conditions we captured, the sensor was constantly in tractive motion after released. However, in the discussion we refer to the work of Shih and Diplas, 2018, where we see that the differences in turbulence will have a very strong effect on the applied forces. For the second part of the question, we will provide tau* and remove the metrics of Table 1 that rely on $\tau$b which probably complicate the comparisons (highlighted also by Reviewer 2).

Tables B1 and B2 will be combined.

Comment: I am a little perplexed by the measured particle protrusion of 0.045 m, i.e. half of the grain diameter, of both the experimental sphere and also the hemispheres. This means that the test particle was basically resting on the flume bed (or very close to it) in addition to resting on four surrounding hemispheres, right? And it also means that the hemispheres were not spaced as closely as possible, but spread out in order to allow the test grain to be down low, right? Figure 1 is a conceptual diagram that does not match the actual experimental design. This should be changed so that Figure 1 is drawn to match the experimental conditions. The experimental grain should be on a bed of hemispheres, not full spheres, the spheres should cover a length of 0.5 m (so I guess 5 hemispheres?), and the spacing and placement of the test grains should be appropriate so that it reflects the actual grain protrusion. Table b1 says the protrusion is half of the particle diameter, which means the particle was essentially resting on the flume bed? I also suggest indicating the approximate water surface on the figure, since it is basically at the top of the test particle. Table b1: was the protrusion the same for the ellipsoid?

Reply: Although the hemispheres where glued in positions that allowed for a 0.045 m protrusion of the sphere, the sand layer made the protrusion higher (by partially filling the gaps between the hemispheres) and non-uniform around the test particle ($\approx$ 0.050 m). The ellipsoid was only supported by the hemisphere bed and was fully exposed to flow. We will provide a clear sketch of the sensor position in Figure 1 using relevant material from Maniatis 2016 (draft figure attached).

Line 25: suggest simplifying wording of "multi-variate two-phase flow defined by a range of interacting complex subprocesses. . ." Line 34: change "it's" to its Reply: We accept the comments, we will correct and re-phrase.

Comment: Line 85: I'm not sure its useful for the authors to give their opinion that not being able to measure position has significantly limited the IMU use. That may be true

but there isn't any way to know if that is the main reason, and does it matter? The current paper doesn't solve this problem, it presents a new way to use the devices for another problem. Also, I think the tone of this paragraph is unnecessarily dismissive of previous work done using similar devices. I don't know what "best considered to be preliminary" is trying to say, other than to belittle this work. The authors come across as arrogant. To me, these works show that there is a benefit and potential of pulling different kinds of information from instrumented particles.

Reply: We have no intention of being dismissive of previous work. The comments about position and the wider use of IMUs is based on several discussions we had with colleagues from all over the world regarding this technology. Many previous studies are pioneering in developing the approach, but they rely on several assumptions about inertial sensors that are often not consistent. One of those is the prospect of deriving the position of the particle from acceleration measurements. This is not possible with current technology due to error accumulation during the integration of accelerations. Another common assumption is that calibration of accelerometers can be achieved under free fall. This is not possible because accelerometers cannot measure free fall (because all the gravity components are constant) and such result relies on the programming of the sensor which for many end users is a black box. Finally, it is often assumed that it is possible to derive directional information (direction of rotation for example) without analysing the frames of reference explicitly. We show the theory behind such analysis and hence why it is required. Many of the published results from early implementation of IMU sensors are no-longer considered reliable by the community, and those that are robust need to be read and interpreted carefully after consideration of the above limitations. We will rephrase our discussion of previous work, but we strongly believe that an open discussion needs to begin around this technology in order to make sure that we actually realise its potential.

Comment: Stylistically, I suggest combining lines 114-128 into one paragraph; I think it is better than having four very short paragraphs. Reply: We accept this comment, we

ESurfD

will combine the paragraphs.

Comment: Lines 199, 202: I think the critical drag and lift force equations should have their own equation numbers, even though they are simple equations. They are important for understanding the analyses, and it was annoying to have to go back and hunt for in- line equations.

Reply: We will give present separate equations and explain the difference between the rolling threshold defined by the torques of equation 6 in revised section 3. Comment: 215: I think this point should be made more prominently, and explained, earlier in the paper: that the fluid is not being measured. When I got to this point I became confused; I was still assuming that the paper would compare fluid measurements and forces on the grains, and it did not make sense that the fluid part was ignored.

Reply: We will address the comment early in the paper and make sure that the reader understands that we don't measure hydraulic forces. We will also change the notation in order to clarify that the hydraulic components (FDr and FLr at the moment) are not measured and that the hydraulic threshold conditions are estimated from the arrangement of the slope (hence also not measured).

Comment: 226: Why 50 hz? More to the point, is there a physical argument that this sampling rate is sufficiently fast to capture the forces acting on the grain? In any case, the authors need to explain why this sampling rate was chosen. Would a slower or faster rate also work?

Reply: This frequency was chosen after considering displacements of 15 particle diameters per second (Drake et al., 1988) which is adequate for capturing the dynamics of the particles if collisions and strong interactions with the bed are excluded (different type and higher frequency piezoelectric sensors are more suitable for the full measurement of impacts).

Comment: 233: 0.5 m long is very short relative to lengths needed to develop boundary

layers (i.e. velocity profiles). How much distance was there between the test particle and the upstream end of the hemispheres? I presume that the upstream flume surface was planar other than the 1.5 mm sand? It seems to me that the possible effects of this this should be addressed or at least acknowledged somewhere.

Reply:The test position was 4.5 m downstream from the flume entrance (see previous comment on flow development). We will clarify that in a revised description of the experimental conditions.

Comment: 245: Give a little more explanation than just "section 4" for where these numbers came from. (in the caption to fig2 it says it used equations 9 and 10).

Reply: We accept this comment and will address it a revised version.

Comment: 252: Give citations and a more complete explanation of why entrainment is defined by this displacement.

Reply: This is the point where the particle has been fully dislodged from its initial position. It distinguishes full entrainment from sub grain diameter vibrations that do not fully dislodge the particle (but are also captured by the sensor).

Comment: 265 or around: I realize the slope is in table b2 but somewhere here say the key numbers, especially slope of 0.1 and flow depth of 0.15 m. Also describe something about the pocket or pockets the grain was placed. In the text says on a step, but explain what that means. Resting on 3 or 4 grains of similar size?

Reply: The difference between the field and flume data in that the flume records the forces when the grain starts to move, but the field only records forces when the grain is already moving. The ellipsoid was released over plain bedrock and the first second of the measurement was removed to eliminate bias from the transport time series (so the effect of the initial arrangement on the measurements from the field should be minimal). We will add the slope and the depth during the experiments in this sentence/paragraph.

Comment: Figure 2: in panel e, I think the Flcr and Fdcr lines are flipped. In panel a,
the part labeled "vibration" really represents grain transport and rotation, right? See also line 291. To me vibration implies a grain wobbling or rocking back and forth in place. How would you define grain vibration as similar or different from grain transport? Is grain vibration just being defined as just transport over a distance less than one diameter? A grain rotating as it is transported?

Reply: The FLcr and FDcr are defined from the orientation of gravitational forces over the specific slope. Without knowing the local bed slope around the grain, we cannot predict which component will have a higher magnitude until the orientation of the slope is defined (we do that using the initial quaternion as stated in each panel). It is however useful to note that these are the norms of the force components and FDcr is defined over an x-y plain, which means that the 3D version of this diagram could indicate different threshold conditions. Grain vibration is defined as sub diameter motion that doesn't result in entrainment, e.g. the grain rocking within its pocket. This was defined visually for the flume experiments and used to calculate the critical impulse probability.

Comment: It seems a little surprising to me that at the end of the paper, the analysis and argument is made that lift forces are more important to entrainment than drag forces, but in Figure 2 panel A and D the drag forces clearly exceed their respective thresholds much more than do the lift forces. I don't know if it is just these examples that are shown or that I'm interpreting something wrong.

Reply: Figure 2 has examples from both the sphere and the ellipsoid in both flume and field conditions. When we collated the force data from the ellipsoid (Figure C1, Appendix) it became clear that the effect of the lift component is more pronounced for the ellipsoid than for the sphere (for both the laboratory and the field experiments).

Comment: I'm not sure what "scale difference" means. It seems to me that the main difference is that in the flume experiment the particle was resting stably prior to entrainment, and this entrainment was analysed. In the field case, the particle was just always and fully transported, and so the data just represent particle transport over a

rough bed without any data on the particles going from resting to mobile. Also the hydraulic conditions were different.

Reply: This is a point also raised by Reviewer 3. Our intention was to point out that in the field we should have different turbulence scaling than in the lab, and that this could partly explain the behaviour of the ellipsoid (and specifically the dependence on the lift component which is even more pronounced in the field). We understand that it needs further clarification and we will do that in a revised version.

Comment: 03: Explain how you know that the distributions are heavy tailed. I presume this is coming from the Weibull, gamma, lognormal fits presented later in figures D1 and D2? If you're going to say heavy tailed in the results section you have to explain it there. You could just remove this mention at this location I think without losing any understanding.

Reply: This is purely observational from the histograms of Figure 5. The fitting in the appendices concerns the calculations presented in Figure 6. We will re-phrase to "right -skewed" and clarify that in a revised version.

Comment: Figure 3b, right panel: the blue line (lift) looks like an odd or poor fit to the data, because there is a whole cluster of blue data points well above the line. Are there a lot of blue points hidden by the red points? Plot the data in some way that data points are not hidden, such as smaller symbols or open symbols.

Reply: We will replot Figures 3 and 5 in order to show the hidden points in the scatter graphs too.

Comment: 335: reword "the on the"

Reply: We will do

Comment: I realize that this is not the subject of this paper, and I am not sure the authors have enough data to really figure this out, but it would be interesting to know how well cumulative impulse from a given hop scales with transport distance of that

**ESurfD**
hop.

Reply: This is very interesting but to calculate hop distances using our measurements we would need to integrate the accelerometer measurement and that is very unreliable at the moment. We plan experiments where position is calculated externally which could address this point.

Comment: "Extended analysis" is not a very informative section title; suggest changing to something different.

Reply: This section is going to be embedded in the discussion as suggested here and in the comments from the other Reviewers. 368: "of of"

Reply: Will delete.

Comment: A final point that I think the authors should acknowledge is that these calculations are untested. They do not know how correct these force measurements actually are. I think that is fine as long as it is stated; I would suggest saying that future work should explore and try to validate the accuracy of these measurements in some way.

Reply: We performed a calibration of the sensor (Maniatis 2016). This included a number of shaking table tests and rolling drop experiments. The main issue is that such a calibration is very sensor specific and not necessarily applicable to any other sensor (in terms of the actual corrections). We acknowledge that this technology develops very rapidly, and we may be able to derive much more accurate (in terms of force magnitude) results in the future. We are a bit more conservative about the derivation of the direction of forces which requires a completely different sensor set up (which is also significantly more expensive). This is the reason why we insist that similar experiments need to be repeated, in a number of settings and after unbiased calibration.

References Lamb, M. P., Brun, F., and Fuller, B. M.: Direct measurements of lift and drag on shallowly submerged cobbles in steep streams: Implications for flow resistance and sediment transport, Water Resources Research, 53, 7607–7629, 2017. Schneider,
J. M., Turowski, J. M., Rickenmann, D., Hegglin, R., Arrigo, S., Mao, L., and Kirchner, J. W.: Scaling relationships between bed load volumes, transport distances, and stream power in steep mountain channels, Journal of Geophysical Research: Earth Surface, 119, 533–549, 2014. Shih, W. and Diplas, P.: A unified approach to bed load transport description over a wide range of flow conditions via the use of conditional data treatment, Water Resources Research, 54, 3490–3509, 2018. Drake, T. G., Shreve, R. L., Dietrich, W. E., Whiting, P. J., and Leopold, L. B.: Bedload transport of fine gravel observed by motion-picture photography, Journal of Fluid Mechanics, 192, 193–217, 1988. Maniatis, G.: Eulerian-Lagrangian definition of coarse bed-load transport: theory and verification with low-cost inertial measurement units, Ph.D. thesis, University of Glasgow, 2016. Ancey, C., F. Bigillon, P. Frey, J. Lanier, and R. Ducret (2002), Saltating motion of a bead in a rapid water stream, Physical review E, 66(3), 036,306.

Plan View of Test Position

Test Position

—0.500m—

—0.900m—

$H_{cr}$ = 0.12m

$b_z$

$b_y$

$b_x$

$r_z$

$r_y$

$r_x$

$H_{cr}$ = 0.09m

$b_z$

$b_y$

$b_x$

= 0.047m

$i_z$

$i_y$

Test Position

Flow

$r_z$

$r_y$

$r_x$

$D_{50}$= 0.015

S = 0.02

—4.5m—
—7m—

**Fig. 1.** Flume_arrangment_draft

---

## Author Comment (AC2) · 13 Jun 2020

Comment: The manuscript by Maniatis is based on a study that uses two custom-built smart particles (one spherical and one elliptical in shape) that contain an accelerometer and gyroscope. The manuscript focuses on (1) presenting a method for putting the accelerations extracted in the particle's frame of reference to one from the perspective of a fixed observer and (2) presenting the lift and drag forces measured while the particle was in motion in a laboratory and field experiment. The paper's strengths are the presentation of the mapping method between the two frames of reference and the introduction of the idea of inertial drag and lift impulses. The inertial drag and lift impulse are similar to the impulse idea put forward by Diplas et al. 2008 (Impulse = integral of force with respect to time applied to a particle) but is based on particle

forces inferred from particle motion rather than particle forces inferred from fluid measurements. I believe the paper presents (1) useful methodology information for those using accelerometer data in particles, and (2) useful data pertaining to particle motion. At the same time, I have a few questions for the authors and some suggestions on the presentation of the work that I think should be addressed.

Reply: We sincerely thank the Reviewer for the thorough review and the positive comments.

Comment: 1. My primary suggestion, or concern, is that the authors make it explicitly clear that all of their force measurements (and impulse calculations) require that the particle be moving (if I am understanding things correctly). It would also be helpful if they expanded their discussion on the benefits and limitations of such measurements. Many of my presentation suggestions below reflect this desire for it to be clear that the forces measured are only those extracted from the particle accelerations.

Reply: We realise that this is a point we need to highlight in the Introduction, Discussion and the section we introduce the Newton – Euler model. It is crucial for the understanding of the measurements to clarify that they only concern a state where the resultant forces on and/or the torques around the centre of mass of the particle exceed zero. We will refer to that explicitly in a revised section 3 of the paper where we will: a) slightly revise the notation in order to make sure that it is completely clear what we measure and what we don't (e.g. FLr is the hydraulic lift force, which is analysed in the r frame, but we cannot decouple it from the FL signal) and b) introduce the rotation threshold (which also refers to a subsequent comment from the Reviewer here). In the discussion we will also summarise the pros and cons of such a method. The main advantage is that the sensor "solves" a very complicated force balance for us, hence the threshold of motion is explicit (sum of torques and forces exceeding 0). The main disadvantage is that in order to predict the motion, complimentary measurements may be required (e.g. detailed flow measurements). However, it is worth noting that the pre-entrainment vibrations of the particle (where the particle moves without being

completely dislodged) are captured by the sensor which can be informative (depending on the size of the particle).

Comment: 2. It seems that you have the highest number of entrainment events for an inertial impulse of zero. Is this because the primary motivating impulse came at a time before that which could be measured by the particle?

Reply: This is difficult to answer with certainty. The zero impulses are generated from forces that reached the threshold value and did not exceed it for more than 1/50th of a second (the sensor's sampling frequency). Also, not every exceedance dislodges the particle and we define the entrainment point at the start of continuous 1 particle of motion. Particle dislodgement is still stochastic for the inertial measurements (Figure 4). There is a very high chance that the motivating hydraulic impulse that begins the mobilisation of the particle comes before the particle starts to vibrate but we have no way to verify that with the existing dataset.

Comment: 3. Along these lines, do the potential travel paths of the particle dictate the forces measured? That is, are the FL and FD measurements sort of pre-determined by the orientation of the particle relative to others in the bed?

Reply: FL and FD would be different in a different bed topography. We are going to highlight this even further in the revised manuscript because we need to clarify that, although the method we present is general, the results are highly dependent on the experimental settings. At the same time, it is difficult to distinguish the most dominant control on the resistance of the particle (the orientation of the particle, the surrounding particle forces or the inertia of the particle). The orientation of the particle and the pivoting angle will dictate the mode of entrainment because the boundary condition will be different. At the same time, it is fair to assume that the trajectory of large particles will depend a lot on their moment of inertia (Icm in the paper) which describes the distribution of mass within the body frame of the particle and affects the direction of the resultant force. This is an exceptionally interesting question that will require a different

data set in order to be answered.

Comment: 4. In equation 5, where is the contact force with the bed? Is it tied into the gravitational force terms? The critical drag and lift forces will depend on the submerged weight of the particle and the orientation of the grain within a pocket through the contact reaction force. The orientation of this force will also influence where it is more likely to have lift or drag dominate. How does it factor into equations 9 and 10? It should, shouldn't it? And 5. Along with the preceding question, how is this contact force for accounted for throughout the range of a particle's motion as it moves and interacts with the bed?

Reply: We want to thank the Reviewer for those observations, we realise that this needs further explanation. The short answer is that the threshold that will depend on the orientation of the particle is the rolling threshold of motion and we explicitly discuss the linear threshold of motion (Reviewer 3 refers to that as the threshold of tractive motion along the downstream and upward directions). In this context gravity keeps the particle on the bed and (since it is significantly exposed to the flow) it defines the threshold of the linear motion because of the opposite an equal reaction of the bed to the particle. We will provide a more directional description of the force balance (with signs) explaining this point.

Also, in the more general Newton Euler model (presented in the current section 3), the surface resistance forces (such as friction) are "hidden" in the FR term of equation 6. Because we resolve the force and torque balance on the centre of the mass of the particle (the sensor does it), all the forces applied on the surface of the particle have to be accounted for as torques. And, strictly speaking, they will only alter the direction of the motion through a rotational component (generating an angular acceleration and a torque that is normal to the direction of rotation). The integral of that angular acceleration is the angular velocity measured by the gyroscope in the body frame.

This rotational component is a spinning component (defined around the centre of the

mass of the particle) and not an orbital component (defined around the centre of mass of a supporting particle as it is common in the literature of sediment hydraulics). It is also negligible when a particle is exposed (in terms of the magnitude of the turning moment applied around the centre of mass of the particle) because it depends on the moment of inertia which is generally a very small number even for relatively large particle diameters (for our spherical particle it is uniform and equal to 0.00085 kg. m2).

Overall, particle orientation will control the mode of entrainment, but we see here that for complete dislodgment (of a quite coarse exposed particle) the gravitational forces need to be exceeded. And that generally depends on the mean orientation of the bed. To address comments (4) and (5) of the Reviewer, we will:

a) Revise sections 3 and 4 in order to include a formula for the rotation threshold (also suggested by Reviewer 3 and described in previous comments here) and explain how it relates with the rolling mode of entrainment.

b) Demonstrate how much smaller the rotational component is compared to the linear component by adding the norm of torques in Figure 1 (for both the sphere and the ellipsoid) and

c) Add an appendix showing the same time series (derived from the norm of angular velocities) at a scale that is more visible.

Comment: 6. Please provide a figure showing the experimental setup with flow depth and bed arrangement. I would think that $\tau B$ has little meaning in terms of mobilization with your particular laboratory setup.

Reply: We are going to revise extensively Figure 1 to address that and will describe the flume setup specifically. We will also remove $\tau b$ based calculations completely; they are not directly relevant, and they complicate any possible comparisons between the flume and the field experiments.

Comment: 7. Terminology in Figure 2 and elsewhere. When do vibrations turn into

motion? Does entrainment start when the particle reaches a distanced travelled of 1 diameter, or does entrainment start when the particle starts to move and then continues on a path that leads to it moving 1 particle diameter? Also, what is a non-entrainment event with a measured inertial impulse? Does that correspond to a case where the particle started to move out of the pocket but then fell back down before reaching the apex?

Reply: We define entrainment at the beginning of a vibration leading 1 particle diameter motion. A non-entrainment event with a measured inertial impulse corresponds to a case where the particle vibrates within the pocket (either parallel to the x or the y direction of the flow) but it is not fully entrained (dislodged into a trajectory of a minimum of one diameter). This explanation will be part of a revised section 5.0.1.

Comment: 8. In the discussion, I'd suggest non-dimensionalizing the force (maybe using sub- merged specific weight) when making the comparisons to other work. Also, how do your inertial forces compare to standard drag estimates using velocity, particle size, and a drag coefficient? What types of relative fluid/particle velocities are needed?

Reply: We are happy to non-dimensionalise the force with the submerged weight of the particle and we agree that it will enhance the comparisons significantly. The comparison of inertial forces with standard drag estimates is more difficult because most of the literature is devoted to fully developed flow which is not the case for our experimental setting (small bed of hemispheres and grain diameter/ depth ratios close to 1). We will refer however specifically to that difference in the discussion.

Presentation questions/concerns/suggestions Comment: The paper is reasonably well written, but I do have some suggestions below that I think would help to improve the presentation of the work. Abstract: L.1 Delete "been" L.4 Replace "on sediment" with "on a particle during" L.5-7 The sentence "Today, twenty years.... for the issue" is not needed in an abstract. Suggest deleting it L.9 Change "grains on" to "grain moving on" L.11 Change "resulting to the" to "resulting in a"

Reply: We accept of all those comments and will revise accordingly.

Comment: Introduction: overall I think you can shorten down the introduction L.38-41 I think you can remove the sentence that starts with "The term Lagrangian..." Most people know the difference between a Lagrangian and an Eulerian frame of reference

Reply: There is a quite delicate balance in terms of what type of terminology is required for such a paper (in this journal). We are in principle happy to make the introduction more concise, but don't want to alienate a broader audience (see comments from Reviewer 1).

Comment: L.44 Delete "(the exact definition of turbulence impulse)" L.72 Change opening sentence to: MEMS-IMU sensors ideally measure forces at the center of mass.... L.74 change to "acting on the grain as it moves" L.83 suggest changing to, "...and electrical engineering. Modeling of IMU error..." L.91&92 suggest changing resolve and resolution to "map" and "mapping" or "rotate" and "rotation" Frames of ref L.123 Suggest, "... frames is non-trivial. A widely-used method..." L.124 Suggest changing "to apply" to "the application of"

Reply: We accept those comments and revise accordingly

Comment: L.129-end of the section. I'm torn here. I could see all of this being better suited for the appendix if the focus of the paper is on the data from the particles. However, if you want the paper to be about the mapping between the frames of reference then you should keep it here.

Reply: We mainly focus on the method; however, we will probably need to move this to an appendix as we need to make room for an expanded explanation on the Newton Euler model.

Comment: L.157-158 Delete the beginning of the sentence that spans the text "As sediment" to "linear acceleration". Just start with ar.

Inertial measurements L.211 change to Ti L.216 suggest changing "that mobilizes

the particle" to "acting on the particle once it starts to move" Lab and field experiments L.238 0.028 l/s is a discharge, not a rate of increase in discharge L.240 change "recorded" to "video of" Figure 4. If I'm correct, I think you reference figure 4 before referencing figure 3. It should be the other way around L.254 suggesting changing to, "...inertial impulses for cases were the grain started to move." Discussion L. 360 - I'd suggest making the extended analysis part of the discussion. Use subsection for the different components of the discussion such as comparison with other work and L.368 - delete one of the "of"s

We accept all those comments and will revise accordingly.

References Maniatis, G.: Eulerian-Lagrangian definition of coarse bed-load transport: theory and verification with low-cost inertial measurement units, Ph.D. thesis, University of Glasgow, 2016.

---

## Author Comment (AC3) · 13 Jun 2020

Comment: This manuscript is intended to quantify the dynamics of entrained sediment particles by using advanced inertial force monitoring systems. I believe this work has good contributions to this long-standing problem and it is worth being published. Overall, the materials are properly organized and relevant concepts explained concisely. I see no significant problems for its publication, but there are several minor suggestions for the authors to consider in their revision. Below, I listed my comments and suggestions as I read through the manuscript.

Reply: We want to thank the reviewer for the thorough review and the supportive comments. We set out to address all the suggestions as will hopefully become clear from

our following responses.

Comment: L50-54: Regarding the Lagrangian approach, Ballio et al. (2018, "Lagrangian and Eulerian description of bed load transport") provide a unified framework to describe sediment particle motion under different frames of reference and quantification methods. The authors may want to refer to this work and highlight their particular contributions to the topic.

Reply: This is an important aspect of our work (see also Maniatis 2016 thesis). Ballio et al. (2018) defined this problem for 1D motions. We also find the consideration of the intermittency of sediment motion from the setup of their model to be very useful. For the scale of particles and motions discussed by Ballio et al. (2018) their approach is the best available to extract Lagrangian metrics from the Eulerian domain (and vice versa). Our approach is directly relevant to this work, but it is also heavily based on the type of measurement we derive. We monitor constantly the Eulerian-Lagrangian orientation changes which gives us the chance to formalise the Lagrangian Eulerian transformations in 3D (with the quaternion multiplication). There are obvious benefits from this approach because it can be applied to all the kinematics (e.g. in theory the trajectory of the particles is fully resolved and the ensemble of trajectories, the Lagrangian, can reconstruct the sediment flux) and there is no reason for even the minimal considerations described in Ballio et al. 2018 (e.g. our operational window is in theory infinite and fixed). But we need to repeat that the measurements are not perfect and the size of particles we can use at the moment to verify that is quite large. We will include a paragraph in the discussion in order to reflect on all the above.

Comment: L58: The issue of over-prediction of transport rates was also extendedly discussed by Bunte et al. (2004, "Measurement of coarse gravel and cobble transport using portable bedload traps") and Singh et al. (2009, "Experimental evidence for statistical scaling and intermittency in sediment transport rates"), and other related papers. The reference can be updated with these contributions.

Reply: We accept the comment. We wanted to focus specifically on the role of particle inertia with that sentence, but referring to those articles will make the presentation more complete.

Comment: Sec. 2 (Frames of reference, rotations and IMU measurements) provides details of mathematical expressions for the conversion/transformation among different frames of reference. But Fig. 1 needs some more clarity on the specifics of these frames, e.g. (x,y,z) (rx,ry,rz),(ix,iy,iz).

This is requested by all three Reviewers. Figure 1 will be significantly revised to reflect better both the frames of reference and the experimental design.

Comment: Based on their formulation, Eq. (9) only concerns the threshold condition in the tractive mode towards downstream, and Eq. (10) the upward movement. Yet, when approaching the strictly critical condition, particle rolling, which has even lower resistance, can be the most predominant entrainment mode. The authors can consider adding an extra formula describing the rolling threshold for providing a more complete framework.

Reply: We accept the comment and we will revise accordingly. As we describe in our reply to Reviewer 2, the rolling condition in the Newton Euler model we present is defined by the sum of torques around the centre of the mass of the particle exceeding 0 (Equation 6). This sum is measured by differentiating the angular velocities derived by the gyroscope. At the same time, we know that this rotational component is negligible (in terms of force magnitude applied on the centre of mass of the particle) because it needs to be multiplied by the moment of inertia (Icm in the paper). Icm is a generally small number even for relatively large particles (for our sphere it is 0.00085 kg. m2). Finally, it is important to clarify that this rotational component is a spinning component (defined around the centre of the mass of the particle) and not an orbital component (defined around the centre of mass of a supporting particle as it is common in the literature of sediment hydraulics). Overall to address this comment we will: a) Revise

sections 3 and 4 in order to include a formula for the rolling threshold

b) Demonstrate how much smaller the rotational component is compared to the linear component by adding the norm of tangential force in Figure 1 (for both the sphere and the ellipsoid) and

c) Add an appendix showing the same time series (derived the norm of angular velocities) at a scale that is more visible.

Comment: L234. Has the 1.5 mm uniform sand glued to surface or also movable?

Reply: No, the bed was washed before the experiments and the sand was static. Will add that in the experimental description.

Comment: Fig.2 (b) describes the change of drag forces during the noted five entrainment events. The pattern shown here, however, somewhat contradicts the impulse model mentioned later in the manuscript (e.g. L278-279). If all these five events follow exactly the impulse criterion, the events of higher magnitude should persist for a shorter duration, and vice versa, to maintain at the same impulse level. In other words, these events, very likely, represent different degrees of particle mobility and not the most extreme cases of entrainment (minimal critical impulse). I think this aspect is worth mentioning in the discussion.

Reply: The exceedance of the threshold doesn't lead always lead to entrainment. In Figure 2b, there are five events where the linear force threshold was exceeded, but entrainment (1 particle diameter dislodgement) took place only during one of them. We need to clarify that even further in the manuscript and this point has also been raised by both Reviewers 1 and 2. In this context the Reviewer is absolutely correct, the exceedances represent both pre-entrainment and entrainment vibrations and the critical impulse can only be approximated statistically as shown in Figure 4.

Comment: Sec. 5.0.2. The calculation of entrainment probability is not clear to me. Is it described as the ratio of entrainment duration to the total observation time, or the

ratio of entrainment events to the total exceedance events observed? I assume it is the latter definition. Adding an equation will help to clarify this point.

Reply: The latter is the case (entrainment / exceedance events) but instead of using the ratio we perform a logistic regression (as also used in Maniatis et al., 2017).

Comment: L300. The scaling effects are usually attributed to the intermittency of particle motion dictated by turbulent flow structures. Yet, this sentence seems to suggest the role of physical scale between the laboratory flume and field stream. A clarification will be helpful (see discussion in Singh et al., 2009).

Reply: We intended to refer to the differences in turbulence (non-fully developed flow in the lab, developed (but still shallow) flow in the field). We will adapt material from Maniatis (2016) to explain that the flume experiments were specifically designed to address the case of particle diameter/ depth ration close to 1 and that the flow was not fully developed and clarify that we refer here to differences in coherent structures between the flume and the field.

Comment: Fig. 4(a) can be improved by using different shading colors for FD and FL, respectively.

Reply: We accept this comment and revise accordingly. Comment: L310-317. The differences in magnitude of critical drag and lift forces between this work and literature data are attributed to the particle sizes, or corresponding mass, used differently. To resolve this issue, a dimensionless quantity, e.g. FD/particle weight, FL/particle weight, can be considered for both the present work and previous reports.

Reply: This point is also raised by Reviewer 2. We are happy to de-dimensionalise using the submerged weight.

Comment: L325-332. The description of the negative lift forces can be more precise in relating to the threshold of motion conditions. Specifically, when the negative lift force appears, the probability of particle lifting reduces, and also, the resistance to the

**ESurfD**

Interactive
comment

tractive movement increases via the enhanced intersurface friction.

Reply: We thank the reviewer for this observation, we are very happy to include it in the discussion despite the fact that we don't account for the surface forces in the thresholds we present in the results.

Comment: L353-357. The description of the entrainment mode of rolling can be placed in the earlier section (see the previous comment) to avoid confusion.

Reply: The comment is accepted and is going to be resolved in a previous section as suggested.

References Maniatis, G.: Eulerian-Lagrangian definition of coarse bed-load transport: theory and verification with low-cost inertial measurement units, Ph.D. thesis, University of Glasgow, 2016. Maniatis, G., Hoey, T. B., Hassan, M. A., Sventek, J., Hodge, R., Drysdale, T., and Valyrakis, M.: Calculating the explicit probability ofentrainment based on inertial acceleration measurements, Journal of Hydraulic Engineering, 143, 2017.

---

## Author Comment (AC4) · 13 Jun 2020

Comment: Even though the authors discuss in section 2 and 3 the frame conversion and its application (only for the acceleration data), they have not discussed its application in changing the critical force based on the particle's ordination as they claim to be doing above. At the same time, as is clear from their text and figure 2, FDcr and FLcr are kept fixed and unchanged, regardless of the particle's orientation, which invalidates their above claim (simply, the critical forces are not shown to be transformed into frame r as the authors claim above - this is also clearly shown in figure 2).

Reply: - As previously stated, we analyse all the forces in a fixed frame the axes of which coincide with the approximate parallel (x-y) and normal (z) direction of the flume

or the riverbed.

- The critical condition for the linear motion of the particle is given by the sum of the resultant forces acting on the centre of the particle being equal to ( $\Sigma$F=0). For a particle highly exposed to the flow, this condition is satisfied by the hydrodynamic forces (FDr, FLr in the paper) being equal to the gravity forces (equation 9 and 10, see also responses to Reviewers 2 and 3 for differences with the rolling threshold and why it is not discussed in this work)

- Gravity has a fixed direction perpendicular to the stream bed. When gravity is analysed in a fixed frame (like the frame r in the presented work), its components are unvarying. Hence the magnitudes of the components of the fluctuating hydraulic forces when projected in the same static frame (again frame r in this work) should match the magnitudes of the gravitational components to satisfy the critical condition. The rotation of the gravitational component to the r frame is given by equation 7 in our paper.

- The components of gravity are not fixed when a mobile frame of reference is used (e.g. Valyrakis et al., 2010). For a frame of reference that is fixed to the centre of the particle (and rotates with it, the body frame in the paper under review), the effect of gravity along the direction of the resultant force will vary. This is not because the weight of the particle changes, but because the reference system moves and the resisting components of gravity depend on the orientation of the particle (the pivoting angle ($\theta$Î£) in Valyrakis et al., 2010).

- we monitor the orientation of the particle constantly and project the components of the resultant force (resolved into FD and FL) to the static r frame. Local microtopography will affect the resultant force, but we have no way to decouple that effect from the other applied forces. At the same time, we know that the critical condition ($\Sigma$F>=0), does not characterise sediment transport and the resultant force has to be sustained for a certain period over to dislodge the particle and/ or sustain its motion. We propose that the resultant force will need to be sustained at least above the critical level for the

hydrodynamic forcing (FDcr, FLcr) which is overall the critical condition for the motion of the particle ($\Sigma F=0$) and we calculate the impulses above that level.

Comment These comments are not contradictory rather they are intended to promote clarity for the presentation of the author's intended contribution. The authors here have significant fallacy in both their understanding of the framework they are presenting and their calculations which is at the crux of their analysis - which is best demonstrated in reference to figure 2-a: The FLr and FDr derived from the accelerometer's data refer to the total forces acting on the particle so the thresholding with and assumed (fixed) critical force is meaningless because the resultant force is the vector sum of the driving hydrodynamic forces (which are here unknown) and critical (resistance) forces which are also unknown (and both are wildly fluctuating) during transport. Simply, this type of thresholding has a questionable value (or relevance) to flow induced transport processes of solids. Even if (this is just a gross mistake and) the author intends to remove the thresholding, the physical relevance of the inertial impulses within the context of sediment transport or incipient entrainment is completely missing and would need be discussed.

Reply:These points are mainly addressed in the reply to the three Reviewers and the first reply here. In addition, it is necessary to note that FLr and FDr are not derived from the accelerometer data and they are not the total forces (denoted FD and FL in the manuscript under review).

Comment: Are the experiments shown herein the same or different to Maniatis 2017? The test bed around which the particle is positioned is not described in any detail: this is crucially important as it interrelates to the particle's transport once entrainment has initiated. For example, if the raised bed has limited length (<2m), to which the author refer to as the minimum transport distance, the entrainment processes described herein are more relevant to a particle falling from the raised bed rather than being transported over plain bedrock, as described in the manuscript. Also, the presence of rough or smooth bed upstream of the significantly raised microtopography would involve the generation of statistically different flow structures compared to those acting on the particle for its transport, which renders these experiments not relevant to the body of work found in the traditional turbulence induced particle incipient entrainment literature, commonly referenced in this manuscript.

Reply: All three Reviewers have asked for more details regarding the flume setting (which was designed to address the case of D/depth close to 1). We will address this in a revised version. As for the characterisation of the flow; we do not measure the flow, we only measure resultant forces. Those forces are characteristic of the setting we tested, and more experiments need to be done in order to generalise to other settings. The intended contribution of this paper is introducing a framework that can be used for this application.

Comment: Still the author doesn't for some reason offer the flow depth at the critical flow conditions. (Just to clarify that the comment offered above, inquiries about the range of flows assessed at the lab, which indeed have been tested, as the authors comments- via implementing a rising hydrograph- so it is not clear why the author disagrees). Also, the authors in their manuscript describe 10 out of 12 measurements mentioned above, what were the reasons to discard two of the measurements? Are the authors showing the (eg aggregate) results for 10, 12 or just one of the experiments? Again, the experiments described in this manuscript are not relevant to the typical incipient motion literature, as the author agrees with the previous reviewer's comment: these are more relevant to boulder transport processes, rather turbulence induced transport of coarse particles (as a reader might wrongly infer by just reading the article's title).

Reply: The flow depth at mean critical discharge was 0.10 m and we are going to provide the exact critical depths for the sphere and the ellipsoid in the revised manuscript. 12 experiments were presented in Maniatis' PhD, 10 of which are used in the manuscript. The initial orientation was not measured to sufficient accuracy in the first two runs, and so they are not used in this paper. This does not affect the calculations in his thesis but could affect the results in the manuscript. No aggregate results are shown in this manuscript.

Comment: Could the author detail how the slope of the flume was measured? (if the experiments were conducted at the 0.9 m wide flume of the University of Glasgow, which I am also using, the maximum bed surface of the flume, which I have measured, cannot reach the mentioned slope of 0.02 as claimed (!) (line 230).

Reply: The slope was directly measured using a surveying level, and the measurements were repeated and validated by other users. Note that the flume has been moved to a new location since these experiments were conducted.

Comment: the authors have misunderstood my commentary. I am not discussing whether their method can be applied to different orientations (which could be done using quaternions or Eulerian angles. etc), they simply do not discuss the dependency of the initial orientation of ellipsoid on the resisting forces (and subsequently the derived impulses). It will also be of interest to elaborate if the application in the field is intended with a given initial orientation or any will do and why? Also, the authors choose to use a very expensive accelerometer of up to 400g, while only recording at a frequency of 50Hz, can the authors discuss the utility of such a design choice (why not use a smaller range for the accelerometer which has less cost?). Also, what is the advantage of recording at such a high acceleration range over such a relatively low frequency (one would ex- pect the high range of acceleration recordings -if indeed needed- to be matched by a high frequency of recordings too (eg 500 Hz or more)?

Reply: The orientation of the ellipsoid affects the application of the surface forces and the rolling mode of transport, but we don't discuss this here (see comments from Reviewers 2 and 3 and our responses). Moreover, we measure the tangential forces (responsible for the turning moments around the centre mass of the particle for the fixed axes representation we use here) to be negligible in comparison to the linear forces (and we demonstrate that in a revised manuscript). For the second issue, high range

does not automatically justify a high sampling rate. We chose the 50Hz frequency after back calculating from previous grain displacement measurements [Drake et al., 1988]. Higher frequencies are more suitable for capturing accurately particle-particle interactions and impacts which is not discussed in this work. High range is necessary because if smaller range sensors are used in natural conditions they are going to be saturated by particle interactions and impacts (even if the focus is not on those) and the data will be either lost or unusable.

Comment: It is appreciated this is not easy to do, but the resisting forces during incipient entrainment as well as transport are significantly dependant on the initial resting micro- topography and bed topography over which the particle will be advected. Also, FDcr and FLcr in the beginning and during transport will depend on these features, which make them important to be detailed. For example, was the particle initially positioned at a plane bedrock or a riffle-pool topography (which part specifically)? Is the transport taking place at the flat bedrock (or pool?) or over the drop from the riffle? In this case, what is the relevance of the inertial impulses to the plane of the bedrock slope and bed shear stresses which the authors claim above to be relevant to the transport processes?

Reply: All those questions are asking about the dependency of critical forces on microtopography, which is covered in previous replies here. In the field, grain translation was initiated over plain bedrock.

Comment: It is not clear if the field test the authors discuss truly refer to incipient motion conditions or full transport processes; for example, for how long was the particle immobile under the free flow, before being entrained?

Reply: We refer to both incipient motion and full transport. The manuscript clearly states that the particle was released on an area of exposed bedrock and was immediately transported in Erlenbach.

Comment: Is the framework the authors intend discussing refer to incipient entrainment

(described in the lab) or transport processes (apparently relevant to the field work).

Reply: The framework is applicable to both incipient entrainment and transport.

Comment: Given that the resistance (highly dependent on the resting microtopography) in the lab experiments and the field tests is different (or not measured exactly in the field), then what is the value of normalising the inertial impulses from the field with the impulses from the lab?

Reply: For the relationship between resistance and microtopography; refer to replies above and responses to Reviewers. The impulses from the field are not normalised by the lab impulses; they are normalised by the mean Impulse in the field. We will clarify the wording in the revised paper.

Comment: Also, for the lab experiments: was the particle stopped because the flow was not able to push it further or it is stopped because it reached near or at the tailgate? To that goal, can you comment on the distance of the test section from the tailgate (ie how much bigger than 2 plus meters is it?).

Reply: There were some experiments where the sensor moved up to end of the flume and some experiments (for example, as in Figure 2 where the sensor stopped within the section with hemispherical roughness elements). In none of the experiments did the sensors hit the tailgate. We will summarise additional details of the experiments (from Maniatis 2016) in the revised manuscript (covering also the requests from the 1st reviewer).

Comment: Usually experiments are used for helping calibrate conditions in the field: what is the practical utility of the authors' lab experiments? What is learnt from this sensor in a quantifiable manner? And how does it physically relate to the processes and past literature of sediment transport?

Reply: It was not a goal of the study to use the flume experiments to "calibrate" field conditions. Rather, the flume and field experiments help to illustrate the particle behaviour under different "boundary" conditions. The other elements of the comment are addressed in the discussion and in the revised version of the manuscript.

References Maniatis, G.: Eulerian-Lagrangian definition of coarse bed-load transport: theory and verification with low-cost inertial measurement units, Ph.D. thesis, University of Glasgow, 2016.

Valyrakis et al. "Incipient rolling of coarse particles in water flows: a dynamical perspective," in Proc. Riverflow, Braunschweig, Germany, June 2010, pp.769-776.

Drake, T. G., Shreve, R. L., Dietrich, W. E., Whiting, P. J., and Leopold, L. B.: Bed-load transport of fine gravel observed by motion-picture photography, Journal of Fluid Mechanics, 192, 193–217, 1988.

---

## Author Response (AR1)

**Authors' response *esurf-2020-20**

Inertial drag and lift forces for coarse grains on rough alluvial beds
Authors: Georgios Maniatis, Trevor Hoey, Rebecca Hodge, Dieter Rickenmann, and Alexandre Badoux

This document is based on the responses to the comments of the interactive discussion. In addition, there are further clarifications and both the notation and the numbering of the lines here follow the revised manuscript. There were three points highlighted by the reviewers which we focused our revisions on.

- In the previous manuscript there was a general agreement that, although the physics of the IMU measurements were introduced, there was not enough clarity about what an IMU sensor measures and how it is different from other force sensors. This was also the basis for a lot of reasonable questions about how we use the IMU measurements to describe sediment motion. To address this, we restructured all the methodology sections, simplified the notation and introduced a lot of non-technical descriptions in the text (moving all the transformation mathematics to Appendices). In addition, we clarified and simplified the presentation of the drag and lift forces on the tested particles as we describe below.

- The comments from Reviewer 1 lead us to introduce our experimental setup fully. In the previous manuscript we tried to hit a balance introducing the sensors and describing how they can be used (guiding the reader to the thesis of Maniatis 2016 for details). The revised manuscript is a self contained document with all the details about the experimental settings discussed and clarified.

- Reviewers 2 and 3 suggested the introduction of the rotational component of sediment motion in the description. We extended the methodology to introduce the necessary derivations for rotation and we added a new appendix which shows the scale difference between the rotational and the linear (translational) components of transport (Appendix C). We also discuss the differences between the rotation of a sphere and ellipsoid (Appendix C) which makes the description of the Newton - Euler model presented in Section 2 more comprehensive.

**Additional Revisions, Clarifications and Simplifications**

In addition to the comments from the reviewers, we realised that Figure 2 (Figure 3 in the revised manuscript) was misleading. To explain the issue we need to introduce briefly the force balance we use in this work. We describe the net force that mobilises the particle as;

$$\vec{F_{net}} = \vec{f} + \vec{F_{Gr}} \tag{1}$$

$\vec{F_{net}}$ is the net force ($\Sigma F$ in the previous manuscript). $\vec{f}$ is the interaction between hydraulic forces (turbulence) and particle forces (support forces and friction) that is not measured directly by an IMU and cannot be decoupled using IMU measurements. $\vec{F_{Gr}}$ defines the force of gravity.

In the previous manuscript $\vec{f}$ was denoted as $F_r$ and the components of $F_{Gr}$ were denoted as $F_{cr}$ depending on how they are calculated (along the drag or lift direction). There was no clear distinction between $F_{net}$ and $F_r$ which could be very missleading.

For the impulse calculations we account for the forces above the threshold of motion thus for when the norm $F_{net} > 0$. However, the previous version of Figure 3 was suggesting that we consider the impulse of the net force above gravity $F_{net} > F_{Gr}$ which is not correct and has no physical meaning. What we did consider are the impulses of interaction $f$ above gravity both in a scalar form (drag plane/direction) and in a vector form (lift direction). This has a physical meaning because at the threshold of motion ($F_{net} = 0$) the interaction $f$ balances gravity $F_{Gr}$.

It is possible to understand that we calculate the $f > F_{Gr}$ impulses throughout the previous manuscript because in the corresponding section (Lines 195-207 in the previous manuscript) we show a clear distinction between the components of the $f$ interaction ($F_r$ hydraulic force in the previous manuscript) and the net acceleration components ($a_r$). However, Figure 2 could really confuse the reader.

To clarify and simplify the presentation we only consider the downstream direction for the drag force in the revised manuscript (while in the previous manuscript we were considering the added forces on the x-y plane parallel to the bed). This has two significant (and we consider positive) effects on the clarity of the manuscript. Firstly, when the reader interprets the force impulse calculations it is easy to relate the drag and the lift forces because now they are both vectors (instead of the version of the previous manuscript where we calculated a vector for the lift direction and a norm for the drag x-y plane). Secondly, we avoid any bias on the statistics and the interpretation of the calculated impulses. Because the particle moves freely the recorded forces have both positive and negative components along each x and y directions. This can complicate the interpretation of impulses as they do not all represent a drag (downstream) motion. For this reason, in the revised version, we only consider the drag component (along the x downstream direction) and we calculate the impulses

over the drag positive direction (similarly to how we calculate the impulses over the positive lift direction). This resulted into the following changes on the calculations:

- All the statistics and the graphs of the Results (section 4) that refer to drag impulses are now for the 1D drag force and for the positive direction.

- The fitting of the empirical distribution in the Discussion (section 5.2 in the revised manuscript) for the drag normalised impulses (Figure 7) is different since the drag impulses as defined in the new manuscript are approximated (marginally) better by a gamma distribution.

- In this context, all the fitting and correlation statistics in appendices E and F are now calculated for the 1D drag force as defined in the revised manuscript.

- Finally, the statistics of the 1D- drag force do not support the interpretation of a clear difference between the impulses recorded in the lab and the field for ellipsoid. However, there is an observable increase of the magnitude of the point forces (from the lab to the field). As a result we adapted all the interpretation and discussion points to reflect the drag components as they are defined in the revised manuscript.

It is important to note that the lift calculations are exactly the same with the previous version (where already calculated in a vector one directional form). In addition, this does not affect the introduction of the IMU model and measurements (Appendix A in the revised manuscript). Here, we provide the previous and the revised version of Figure 3, highlighting the main difference and the reason we decided to do this revision.

[Figure]

**Previous (misleading) version of Figure 3 in the revised manuscript (Figure 2 in the previous one)** The exceedance of the drag force over threshold is depicted as the net force exceeding the threshold of gravity (orange window). This thresholding has no physical meaning and it is a presentation error. By focusing the graphs b,c and d above the green line for the drag force we confuse the reader. The generated impulse we are interested into, and we calculated both in the previous and the revised manuscript, is the impulse of the net force exceeding 0 (green window). This is the equivalent of the f interaction exceeding gravity (see new version of Figure 3, below and Comment 3.7)

[Figure]

**New Figure 3 in the revised manuscript (Figure 2 in the previous one)** The windows (green and purple) show the exceedance used for calculating the impulses. This can be described either as the net force exceeding 0 ($F_{net} > 0$, green windows) or as the f interaction exceeding gravity ($f > F_G$, purple windows). This is the thresholding used in the previous version as well and this is why the impulse results are exactly the same for the lift force (which is defined as one directional vector in both versions) . The drag force is now an 1D vector and not a force magnitude scalar as was introduced in the previous manuscript. Also, in the new manuscript we discuss briefly the noise threshold of the sensor denoted by the black line N.T

Hereafter, for each comment, we have first highlighted the issue, then we provided an answer, and finally we described how the manuscript was adjusted.

**REVIEWER # 1**

**COMMENT # 1.1**

The manuscript "Inertial drag and lift forces for coarse grains on rough alluvial bed" presents an interesting method for calculating forces acting on grains in motion in mountain rivers. I think the topic and approach is fascinating as has potential to greatly improve our understanding of transport and flow. My suggestions for improvement mainly focus on (1) explaining more of the conceptual physics and assumptions, (2) simplifying some of the writing, (3) explaining a little more how the flow conditions are perhaps not typical of gravel transport in mountain rivers, and (4) changing figure 1 to illustrate the actual experimental setup of the experiments.

We sincerely thank the author for the thorough review and the supportive comments. We set out to address all the suggestions in the revised version. Hopefully this will become clear in our following responses.

**COMMENT # 1.2**

I appreciate the physics framework that the paper is written in, but as a non-physicist I think it would help to have a little more explanation of some of the underlying conceptual ideas (translation into geologese?). Here is an example from the very beginning: To me, the title clearly claims that it explores drag and lift forces for particles ON the bed. After puzzling through it and trying to think through the physics, I don't think this is correct. The methods proposed only apply to forces acting on a particle when it is in active transport, not when it is on the bed. Maybe to a physicist, having "inertial" in the title would make it obvious that the grain isn't actually resting on the bed, but that wasn't the case for me. I am still not sure what the "inertial" part is supposed to tell me. I don't think this is just a matter of semantics; it's a matter of communicating clearly to the geoscience community. Many papers have been written (and that the authors appropriately cite) using force sensors that measure drag and lift forces for grains on the bed. This paper is doing something different, which is unique and fascinating-to measure forces on grains while they are in motion–but those differences should be made clear.

We appreciate the value of explaining the concepts using language that will be familiar to the reader. Some of the underlying concepts we present will be more familiar to people with backgrounds in hydraulics or physics, but we consciously target

the audience of this journal (broadly the audience working on geomorphological and geological problems).

The term "inertial" in the context we present primarily means that the reference frame we use to analyse particle forces is static, i.e. that the frame of reference doesn't move with the particle. This is not a common approach in sediment transport but is necessary to enable us to interpret the accelerometer measurements: these are meaningless if untransformed. In addition, the term "inertial" means that, in the reference frame we use, there are no so-called fictitious forces. This observation is very straightforward for the hydraulic forces (and the forces from the surrounding particles) acting on a target particle but needs to be specifically stated for gravity because gravity is a fictitious force for a free moving accelerometer (Appendix A).

Finally, it is very important to clarify that inertial sensors measure forces only when the particle moves. The advantage of this is that the motion of the particle is captured explicitly (the sensor solves a very complicated force balance for us). The disadvantage of this is that there is no way to "decouple" the acting forces without complementary measurements (no information about the hydraulic drag component, for example).

To address this comment (which also overlaps with comments from the other two reviewers) we:

- Restructured completely the methodology sections of the manuscript. Instead of introducing the necessary transformations for the derivation of the accelerometer measurements in the text, we discuss conceptually the measurements and their relationship to the forces applied on the particle (section 2). However, we still provide all the technical background for the necessary transformations in appendices (Appendices A, B and C)

- Simplified the notation and we avoid terms that can lead to confusion. A good example of this was the term "critical force" which we used in the previous manuscript to define the gravity threshold but readers with a background in hydraulics relate to a combination of forces (friction, reaction etc.). This is now discussed and clarified (throughout section 2).

- Explained the difference between accelerometer measurements and other force sensors (such as load cells, Lines 110 -120)

- Introduced a new section in the discussion on the application of IMU sensors in geomorphology (section 5.1)

COMMENT # 1.3

I suggest adding a paragraph or whatever describing forces on particles from the flow when the particle is stable on the bed, when it is moving but still has its mass partially supported by contact with the bed, and when it is fully entrained by the flow, and what the accelerometer-based calculations actually measure.

In section 2 we clarify further which are the force components captured by the sensor and which are the forces that cannot be decoupled from an IMU. We also describe the rotational component. In the future we hope to perform similar experiments in conjunction with detailed flow measurements and enhance the description of incipient motion.

The threshold of motion (as it is captured by an accelerometer) is discussed specifically in the newly revised section 2.1

COMMENT # 1.4

Is the way to describe it that the measurements are not forces acting on the particle, but the net force that the particle responds to, from the combination of fluid and solid contacts along the particle boundary?

This is exactly the case; the only difference is that we were using the term "resultant" in the previous manuscript. In the revised version we used the terms net force and net torque to simplify the terminology.

COMMENT # 1.5

I think that simplifying the language in many places could also make the manuscript easier to read and understand. As part of this I suggest the authors take the time to go through it again and get rid of some superfluous words (such as "proven" in the second line of the abstract, maybe "regime" in the first line, and other places throughout. "has historically been proven" could also be replaced by "is").

We tried our best to address this comment. We have reduced the text and simplified the language throughout the manuscript.

The attached annotated manuscript shows clearly the extend of language revisions. Some distinct examples are:

- Abstract

- Lines 30-45

- Paragraph starting with line 70

- Section 2 throughout

- Appendix A (lines 737 to 745, in-text in the previous manuscript.)

COMMENT # 1.6

I think the manuscript would be more clear if the authors define drag and lift forces. It helped my understanding to look up physics definitions: drag is simply force parallel to the direction of fluid motion, and lift is force perpendicular to the direction of fluid motion. I did not realize that drag and lift were defined so simply (thanks, I learned something new). I suppose these are sort of defined on line 192, but this is not nearly obvious enough; I did not realize that these were definitions. In particular for lift, I thought it implied differential pressure on the top or the bottom of particles from different velocities (i.e. Bernoulli), and I was confused because it seemed like the authors could not know this since they have no measurements of fluid velocities or pressures. I realize I was wrong, but my point is that drag and lift are terms that bring preconceived ideas about the flow.

The fact that the Reviewer had to go through that enquiry makes the clearer definition of forces necessary. For the revised manuscript we only consider the force along the drag direction $r_x$ ($F_{Dnet}$ in the revised manuscript) and the force along the lift direction $r_z$ normal to the bed ($F_{Lnet}$). Quoting from the revised manuscript (section 2, Lines 160-166)

"we can use the linear accelerations [*the transformed accelerometer measurements*] of equation (1) to separate the components of the net force ($F_{net}$) along the drag and lift directions as:
$$F_{Dnet} = m \, \vec{a_{rx}}$$
and
$$\vec{F_{Lnet}} = m \, \vec{a_{rz}}$$
"

COMMENT # 1.7

I couldn't quite figure out from the equations whether the water surface slope is accounted for; i.e. are the lift forces the authors calculate actually perpendicular to the mean flow direction, or are they instead parallel to the vector of gravitational acceleration? Figure 1 indicates that they are perpendicular/parallel to mean flow, not horizontal or vertical. Is a small angle approximation required, or embedded in the equations?

The resultant lift force ($F_{Lnet}$) is perpendicular to the mean flow direction ($r_z$). The confusion was based on the fact that in the previous manuscript we didn't make clear enough that the spherical and the ellipsoid particle had different sensor alignments (but the lift force was always transformed in the static $r$ frame). We now make this clear with a revised figure for the experimental setup (the figure is scaled to fit in this document, Figure 2 in the revised manuscript):

[Figure]

**Laboratory setting and initial aligment** The diagram shows the arrangement of the bed of hemispheres and the test position (4.5 m downstream the entrance of the flume). Upstream of the hemispheres bed, the flume was filled with densely backed gravel ($D_{50}$ = 0.015 m) to allow for the development of the flow. $r$ stands for river-bed (flume in this case) reference frame, $b$ for body frame and $i$ for the inertial reference frame ($_i^r q$ = [0.92, 0.14, -20, 028]). $S$ is the slope of the channel (0.02). The sketch also depicts the initial aligment for each device which ($b - i$ for the sphere and $b - r$ for the ellipsoid).

COMMENT # 1.8

Another point that I think needs to be discussed, and limitations explicitly pointed out, is that the experimental conditions of the flume experiments are not representative of typical gravel transport Table B1 says that the flow depth was 0.1 m, which is essentially the same as the sphere diameter of 0.09 m. Grains the same size as flow depths are relevant to boulder transport, for example. I can accept the conditions as being informative even if not typical for gravel-bed systems, but it does mean that many of the scientific results, like the relative importance of lift and drag forces, may not be more broadly applicable.

The Shields stress is very low for typical thresholds of motion (0.013), which my guess is related to the high protrusion and fact that the grain blocks basically all of the flow depth. Another factor in the flume experiments is probably that the hemispheres mounted on the flume bed only spanned a length of 0.5 m, which means that the boundary layer velocity profile was not anywhere close to developed. Somewhere, the paper should say where within this distance the test particle was placed (i.e., what was the distance between the upstream edge of hemispheres and the starting position of the test particle?).

We answer two comments here because they relate. In the previous manuscript we did not provide enough detail for the experimental regime, guiding the reader to the thesis of Maniatis 2016. For example, we only reported the mean critical discharge for all the experiments (approximately $30 \, \mathrm{l.s^{-1}}$) which is not complete because the ellipsoid and the sphere have different thresholds. In the new manuscript we describe in detail both the experimental setting and the measured or calculated metrics. We also state clearly that the experiment is only relevant to large grains and high exposure to the flow (Figure in Comment 1.7). Here are two quotes from the revised manuscript that describe the initial position and the hydraulics for the flume experiments (section 3.1):

"The hemispheres were glued to form a 0.5 m (L) x 0.9 m (W) section and the whole section was placed at the point which allowed for the test particle to be at 4.5 m from the upstream boundary of the flume. A thin layer of 1.5 mm uniform sand was glued to both the hemispheres section and the upstream flume surface. The section upstream of the hemispheres was filled with very densely packed non-uniform, rounded gravel ($D_{50}$ = 0.015 m) enabling the development of turbulent flow. No sediment transport occurred from the upstream gravel section during the entrainment experiments. The hemispheres were glued in positions that produced a 0.045 m protrusion of the sphere above the top of the hemispheres, although the sand layer made the protrusion higher, by partially filling the gaps between the hemispheres, and non uniform around the test particle ($\approx$ 0.050 m). The ellipsoid was only supported by the hemispheres and was fully exposed to flow (Figure 2 ."

"The critical discharge for the sphere was 24.8 $\pm$1.8 $\mathrm{l.s^{-1}}$ which corresponds to a critical depth of 0.095 $\pm$ 0.015 m, measured from the top of the supporting hemispheres. The critical discharge for the ellipsoid was 45.2 $\pm$ 2.2$\mathrm{l.s^{-1}}$ which corresponds to a critical depth 0.12 $\pm$ 0.02 m ($\tau_*$ = 0.01 and 0.02 respectively, $\tau^* = \frac{H\,S}{(\rho_p - \rho_f)\,D}$, Appendix D) (...) The flume experiments have particle diameter to flow depth ratio close to 1 (d/H $\approx$ 1) despite the fact that the particles were fully submerged at the critical depth for all the experiments (Figure 2). The tested conditions are relevant to the entrainment of coarse particles in steep mountain streams but they should not be

directly generalised to other bedload transport regimes despite the generality of the Newton-Euler model presented in section 2."

COMMENT # 1.9

Similarly, it is unclear whether the result of lift and drag becoming uncorrelated from the lower flow (flume) to the higher flow (field) conditions represents the difference in flow depth relative to grain size or the difference in flow intensity. Also, would be helpful for Table B2 to have a calculation of tau* for comparison.

For the first part of the question, we observe a generally weak correlation between the net drag and lift forces. The highest correlation was observed for the spherical particle in the flume (R=0.3). For the ellipsoid the correlation is consistently weak (from R=0.046 in the lab to R=-0.0055 in the Erlenbach during transport). The correlation statistics are given in Appendix E. To answer why the correlation becomes even weaker in the field we would need a series of incipient motion experiments in the field and that was not possible. For the conditions we captured, the sensor was constantly in tractive motion after released. However, in the discussion we refer to the work of Shih and Diplas, 2018 (and other works) and state that the differences in turbulence will have a very strong effect on the applied forces. For the second part of the question, we now provide $\tau^*$ and have removed the metrics of Table 1 that rely on $\tau_b$ which complicated the comparisons (highlighted also by Reviewer 2). Tables B1 and B2 were also combined (Appendix D).

Lines 420-425 of the revised manuscript: "...Differences in force magnitude and duration can relate to transitions, as described by Shih and Diplas (2018), from hydraulic "impulse controlled" transport, as in our flume experiments, to "force-magnitude controlled" transport corresponding to the dynamics recorded in Erlenbach. However, an important difference between laboratory and field experiments lies in the scales of turbulence (Coleman and Nikora, 2008; Singh et al., 2009), which requires further investigation since detailed flow measurements were not made during the presented experiments (eg. PIV measurements)."

COMMENT # 1.10

I am a little perplexed by the measured particle protrusion of 0.045 m, i.e. half of the grain diameter, of both the experimental sphere and also the hemispheres. This means that the test particle was basically resting on the flume bed (or very close to it) in addition to resting on four surrounding hemispheres, right? And it also means that the hemispheres were not spaced as closely as possible, but spread out in order to allow the test grain to be down low, right? Figure 1 is a conceptual diagram that does

not match the actual experimental design. This should be changed so that Figure 1 is drawn to match the experimental conditions. The experimental grain should be on a bed of hemispheres, not full spheres, the spheres should cover a length of 0.5 m (so I guess 5 hemispheres?), and the spacing and placement of the test grains should be appropriate so that it reflects the actual grain protrusion. Table b1 says the protrusion is half of the particle diameter, which means the particle was essentially resting on the flume bed? I also suggest indicating the approximate water surface on the figure, since it is basically at the top of the test particle.

Table b1: was the protrusion the same for the ellipsoid?

We reply to both comments in the responses for Comment 1.7 and Comment 1.8 and the revised section 3 of the paper.

COMMENT # 1.11

Line 85: I'm not sure its useful for the authors to give their opinion that not being able to measure position has significantly limited the IMU use. That may be true but there isn't any way to know if that is the main reason, and does it matter? The current paper doesn't solve this problem, it presents a new way to use the devices for another problem. Also, I think the tone of this paragraph is unnecessarily dismissive of previous work done using similar devices. I don't know what "best considered to be preliminary" is trying to say, other than to belittle this work. The authors come across as arrogant. To me, these works show that there is a benefit and potential of pulling different kinds of information from instrumented particles.

As we stated in the online discussion, we had no intention to dismiss previous work. We just believe that there is still a lot to learn about those sensors and we (as a community) need to be more aware about the details (and the physics) of how they operate. We revised the sentence and we added the following paragraph in the newly added 5.1 section of the Discussion

"Further, there has been a recent rapid increase in use of IMU sensors, but most off-the-shelf IMU sensors are not suitable for the range of forces characterising natural sediment transport, especially if the focus is on particle interaction or impacts (Maniatis et al., 2013). In addition, the physics of IMU sensors are complex and a number of common assumptions about their use do not always hold. For example, while dead-reckoning appears to allow positions to be recovered by double-integration of linear accelerations, uncertainties introduced during the production of IMUs (mostly nm scale imperfections on the alignment of the sensors) lead to extreme uncertainty in positional estimates. A second issue involves calibrating IMU accelerometers which

has often been done using free fall drop experiments. An accelerometer in free fall will measure 0 acceleration despite being subjected to the acceleration of gravity, as gravity in the context of the body frame of the accelerometer is a so-called fictitious force (Appendices A and B). Consequently, the force/impact results of a free fall drop experiment which relies solely on IMU measurements are highly dependent on how quickly the sensor is programmed to enter and wake up from the free fall detection state (Clifford, 2006). It is possible to approximate the height of the free fall using the approximate time of the free fall state. However, the measurement of the impact force needs a very detailed description of both the impact surface and the low-level code that controls all the basic operations of the sensors (on-off routines, logging, storage-handling etc.). This low-level programming is a black box for proprietary off-the-shelf sensors and for users who are not trained programmers. Finally, it is not possible to derive directional information, even for forces, for long mobile periods without complementary corrections (Kok et al., 2017) or without a detailed presentation of the reference frames involved and their initial alignment."

COMMENT # 1.12

Lines 199, 202: I think the critical drag and lift force equations should have their own equation numbers, even though they are simple equations. They are important for understanding the analyses, and it was annoying to have to go back and hunt for in-line equations.

We have revised the notation (and the terminology) to avoid confusion and the same parameters are now defined as the gravity components. Exactly the same definition with the previous version but different name so it is clear we only refer to the gravity force. They also have their own equations. From the revised manuscript (section 2):

"The vector $\vec{FG}_r$ defines the force of gravity rotated in the r frame and compensated for hydrodynamic effects (...). The magnitude of $\vec{FG}_r$ components along the drag direction ($r_x$) is given by the vector:

$$\vec{FG}_D = \vec{FG}_{rx}$$

For the direction normal to the bed ($z_r$, direction of lift hydraulic force) the $\vec{FG}_r$ component is the vector:

$$\vec{FG}_L = \vec{FG}_{zr}$$

"

COMMENT # 1.13

226: Why 50 hz? More to the point, is there a physical argument that this sampling rate is sufficiently fast to capture the forces acting on the grain? In any case, the authors need to explain why this sampling rate was chosen. Would a slower or faster rate also work?

We have now added the reference that lead us to chose this sampling rate (section 3).

"The nominal sampling frequency of the sensor (used for all the flume and field experiments presented in this work) is 50 Hz which permits constant use for approximately 5 hours (LiPo rechargeable battery). This frequency was chosen after considering displacements of 15 particle diameters per second reported by Drake et al. (1988). This is adequate for capturing the dynamics of the particles if collisions and strong interactions with the bed are excluded (different type and higher frequency piezoelectric sensors are more suitable for the full measurement of impacts)."

COMMENT # 1.14

215: I think this point should be made more prominently, and explained, earlier in the paper: that the fluid is not being measured. When I got to this point I became confused; I was still assuming that the paper would compare fluid measurements and forces on the grains, and it did not make sense that the fluid part was ignored.

We start section 2 of the revised manuscript with an elaborate (but not technical) description of what an accelerometer measures to define the type of recorded forces. Also, we introduce separate notation for the forces that are captured directly and estimated indirectly (capitals and low case letters respectively) to highlight the differences even further. Example sentences from section 2 that highlight this in the revised manuscript are:

"If the sensor moves and the sensor frame accelerations are compensated for gravity (by removing gravity from the raw accelerometer measurement) then the sensor will record the 3D components of the resultant or net force that mobilises the particle. This resultant is the force that can be observed from an observer who is static in relation to the particle."

"...The left hand elements of equations 1 and 2, describe the forces applied on the particle. Hereafter, the lower case vectors and scalars (e.g terms $\vec{f}$ and $\vec{f_{rot}}$) will refer to the interactions between hydraulic forces (turbulence) and particle forces (support forces and friction) that are not measured directly by an IMU and which cannot be decoupled using IMU measurements."

COMMENT # 1.15

233: 0.5 m long is very short relative to lengths needed to develop boundary layers (i.e. velocity profiles). How much distance was there between the test particle and the upstream end of the hemispheres? I presume that the upstream flume surface was planar other than the 1.5 mm sand? It seems to me that the possible effects of this this should be addressed or at least acknowledged somewhere.

The hemisphere bed was 4.5m downstream from the flume entrance (see Comment 1.7 and Comment 1.8).

COMMENT # 1.16

245: Give a little more explanation than just "section 4" for where these numbers came from. (in the caption to fig2 it says it used equations 9 and 10).

This is now the corresponding section 2.1 which is much smaller and easier to follow. We refer to the gravity components we describe in Comment 1.12.

COMMENT # 1.17

252: Give citations and a more complete explanation of why entrainment is defined by this displacement.

We do not think there is a need for a citation because the incipient motion threshold is explicit for this work (is for $F_{net} > 0$). The clarification about how we define the entrainment as the full dislodgment of the particle is given in section 3.1 of the revised manuscript.

"We define the entrainment point as the initiation of the vibration that dislodges the particle by one particle diameter."

COMMENT # 1.18

265 or around: I realize the slope is in table b2 but somewhere here say the key numbers, especially slope of 0.1 and flow depth of 0.15 m. Also describe something about the pocket or pockets the grain was placed. In the text says on a step, but explain what that means. Resting on 3 or 4 grains of similar size?

In the revised manuscript (section 3.3) we include key numbers and we clarify that the sensor was released over bare bedrock and that there was no initial position relevant this experiment.

"...The stream has a step-pool morphology allowing the sensor to be retrieved from pools, so the ellipsoid sensor was submerged on a bare bedrock section close to the edge of a step (average slope $S = 0.1$, cross-averaged flow depth $H = 0.1$ m, $\tau_* = 0.095$), aligned to the same orientation as the riverbed (...) and allowed to be transported until it stopped moving and remained immobile for at least 10 seconds (...). In all the experiments the sensor was entrained fully and immediately as there was no vibration *in situ*. The first one second of each transport event was removed from the data as the effect of holding and releasing the sensor were still present."

COMMENT # 1.19

Figure 2: in panel e, I think the Flcr and Fdcr lines are flipped. In panel a, the part labeled "vibration" really represents grain transport and rotation, right? See also line 291. To me vibration implies a grain wobbling or rocking back and forth in place. How would you define grain vibration as similar or different from grain transport? Is grain vibration just being defined as just transport over a distance less than one diameter? A grain rotating as it is transported?

Figure 2 is now Figure 3 and has been revised extensively to clarify the differences between that applied and the resultant (net) forces. FLcr and FDcr have been replaced (in terms of notation) with the $FG_D$ and $F\vec{G}_L$ gravitational components (Comment 1.12). In addition to the explanation we provide in the online discussion (the magnitude of the gravitational components for a given direction is not always intuitive), we only accounted for magnitudes in the previous manuscript along the x-y plane. This is now should be clear after the revisions we describe in the beginning of this document. The label "vibration" refers to pre- full dislodgment vibrations which are also captured by the sensor (from our definition of entrainment, Comment 1.17).

COMMENT # 1.20

It seems a little surprising to me that at the end of the paper, the analysis and argument is made that lift forces are more important to entrainment than drag forces, but in Figure 2 panel A and D the drag forces clearly exceed their respective thresholds much more than do the lift forces. I don't know if it is just these examples that are shown or that I'm interpreting something wrong.

Figure 3 (former 2) has examples from both the sphere and the ellipsoid in both flume and field conditions. Also in the previous manuscript the drag force was the absolute value of the addition of 2 forces. In the revised manuscript we have simplified the presentation by calculating the 1D component. When we collate the force data from the ellipsoid (Forces in Appendix E) it becomes clear that the effect of the

lift component is more pronounced for the ellipsoid than for the sphere (for both the laboratory and the field experiments). In addition, in the revised manuscripts we clarify what the "respective thresholds" are. In the previous version of Figure 3 there was an ambiguity which could miss-lead the reader and is now clarified (see introduction of this document)

COMMENT # 1.21

I'm not sure what "scale difference" means. It seems to me that the main difference is that in the flume experiment the particle was resting stably prior to entrainment, and this entrainment was analysed. In the field case, the particle was just always and fully transported, and so the data just represent particle transport over a rough bed without any data on the particles going from resting to mobile. Also the hydraulic conditions were different.

This is a point also raised by Reviewer 3. Our intention was to point out that in the field we should have different turbulence scaling than in the lab, and that this could partly explain the behaviour of the ellipsoid and specifically the dependence on the lift component which is even more pronounced in the field. We briefly discuss the differences in turbulence in the discussion section (Comment 1.9)

COMMENT # 1.22

Explain how you know that the distributions are heavy tailed. I presume this is coming from the Weibull, gamma, lognormal fits presented later in figures D1 and D2? If you're going to say heavy tailed in the results section you have to explain it there. You could just remove this mention at this location I think without losing any understanding.

This is purely observational from the histograms of Figure 6 (former 5). The fitting in the appendices concerns the calculations presented in Figure 7. We re-phrased to "right -skewed".

COMMENT # 1.23

Figure 3b, right panel: the blue line (lift) looks like an odd or poor fit to the data, because there is a whole cluster of blue data points well above the line. Are there a lot of blue points hidden by the red points? Plot the data in some way that data points are not hidden, such as smaller symbols or open symbols.

The points are much smaller in the revised version and they overlap less. Thus it is obvious that there are points close to 0 that justify the fit. This is also confirmed by the histograms for each comparison.

C<small>OMMENT</small> # 1.24

I realize that this is not the subject of this paper, and I am not sure the authors have enough data to really figure this out, but it would be interesting to know how well cumulative impulse from a given hop scales with transport distance of that hop.

This is very interesting but to calculate hop distances using our measurements we would need to derive distance from accelerometer measurement and that is very unreliable at the moment. We can workaround this with video recordings or other types of tracking. We plan experiments where position is calculated externally which could address this point.

C<small>OMMENT</small> # 1.25

A final point that I think the authors should acknowledge is that these calculations are untested. They do not know how correct these force measurements actually are. I think that is fine as long as it is stated; I would suggest saying that future work should explore and try to validate the accuracy of these measurements in some way.

We address this in text (Discussion 5.1)

"Here, we calibrated and deployed a commercial IMU sensor following standard procedures (Maniatis, 2016), but the precise corrections used are sensor specific and similar procedures should be followed again for any other IMU sensor. The calibration of force measurements is likely to be standardised and simplified in the near future as the use of IMU sensors develops further. Similar standardisation for the direction of forces is potentially further away as it requires using IMU sensors that rely on optical technology and which are currently not manufactured with physical dimensions or within a price range that is accessible for sediment transport studies (De Agostino et al., 2010)."

C<small>OMMENT</small> # 1.26   P<small>RESENTATION</small>/ G<small>RAMMAR AND WORDING</small>: R<small>EVIEWER</small> 1

- Line 25: suggest simplifying wording of "multi-variate two-phase flow defined by a range of interacting complex subprocesses. . ."

- Line 34: change "it's" to its

- 335: reword "the on the"

- "Extended analysis" is not a very informative section title; suggest changing to something different.

- 368: "of of"

- Stylistically, I suggest combining lines 114-128 into one paragraph; I think it is better than having four very short paragraphs.

All the presentation and grammar comments have been addressed in the revised version (see attached track changes manuscript)

**Reviewer # 2**

**Comment # 2.1**

The manuscript by Maniatis is based on a study that uses two custom-built smart particles (one spherical and one elliptical in shape) that contain an accelerometer and gyroscope. The manuscript focuses on (1) presenting a method for putting the accelerations extracted in the particle's frame of reference to one from the perspective of a fixed observer and (2) presenting the lift and drag forces measured while the particle was in motion in a laboratory and field experiment. The paper's strengths are the presentation of the mapping method between the two frames of reference and the introduction of the idea of inertial drag and lift impulses. The inertial drag and lift impulse are similar to the impulse idea put forward by Diplas et al. 2008 (Impulse = integral of force with respect to time applied to a particle) but is based on particle forces inferred from particle motion rather than particle forces inferred from fluid measurements. I believe the paper presents (1) useful methodology information for those using accelerometer data in particles, and (2) useful data pertaining to particle motion. At the same time, I have a few questions for the authors and some suggestions on the presentation of the work that I think should be addressed.

We sincerely thank the Reviewer for the thorough review and the positive comments.

**Comment # 2.2**

My primary suggestion, or concern, is that the authors make it explicitly clear that all of their force measurements (and impulse calculations) require that the particle be moving (if I am understanding things correctly). It would also be helpful if they expanded their discussion on the benefits and limitations of such measurements. Many of my presentation suggestions below reflect this desire for it to be clear that the forces measured are only those extracted from the particle accelerations.

Those are points we now highlight throughout the manuscript. Firstly, we define the forces in a clearer manner and we indicate the interactions that are not captured by the accelerometer (interactions f). Secondly, we clarify the threshold of motion (see introductory statement and Comment 1.20. Finally, we discuss the pros and cons of the method in the discussion (new section 5.1). From the revised manuscript:

**Comment # 2.3**

"The advantage of using an IMU sensor for capturing grain motion is that the sensor apparently solves a complex force and torque balance and removes any ambiguity for whether or not a test particle is in motion, as motion leads to the explicit thresholds $F_{net}$ and/or $T_{net}$ exceeding 0. Entrainment is captured directly and, assuming correct sensor calibration, robustly. IMUs can be a useful tool for geomorphologists since they offer a realistic prospect for monitoring particle motion during transport without invasive apparatus which is not possible with standardly used equipment, especially in field applications (e.g. PIT tracers). At the same time, it is important to recognise that that exceedance the explicit thresholds above does not always produce complete dislodgment of the particle and also does not directly describe the modes of transport in the context that is commonly assumed for sediment hydraulics (...). For a complete understanding and effective prediction of grain motion both the hydraulic and the particle forces need to be measured, analysed and decoupled from the inertial forces we measure in this study. "

COMMENT # 2.4

It seems that you have the highest number of entrainment events for an inertial impulse of zero. Is this because the primary motivating impulse came at a time before that which could be measured by the particle?

This is difficult to answer with certainty. The zero impulses are generated from forces that reached the threshold value and did not exceed it for more than 1/50th of a second (the sensor's sampling frequency). As we clarify in the revised manuscript, the threshold value for the drag direction (now the x direction and not the x-y plane) is the the noise threshold of the sensor as we only consider the norm of the net forces exceeding 0. For the lift direction is the component of gravity that balances the $f_L$ interaction. Also, not every exceedance dislodges the particle and we define the entrainment point at the start of continuous 1 particle of motion. Particle dislodgement is still stochastic for the inertial measurements. There is a very high chance that the motivating hydraulic impulse that begins the mobilisation of the particle comes before the particle starts to vibrate but we have no way to verify that with the existing dataset.

COMMENT # 2.5

Along these lines, do the potential travel paths of the particle dictate the forces measured? That is, are the FL and FD measurements sort of pre-determined by the orientation of the particle relative to others in the bed?

$F_{Lnet}$ and $F_{Dnet}$ (consequently and the interactions $f_L$ and $f_D$) would be different in different bed topography, and so would be altered in a different microtopography.

We have highlighted in the revised manuscript that the results are highly dependent on the experimental settings (Comment 1.8). At the same time, it is difficult to distinguish the most dominant control on the resistance of the particle (the orientation of the particle, the surrounding particle forces or the inertia of the particle). The orientation of the particle and the pivoting angle will dictate the mode of entrainment because the boundary condition will be different. At the same time, it is fair to assume that the trajectory of large particles will depend a lot on their moment of inertia ($I_{cm}$ in the paper) which describes the distribution of mass within the body frame of the particle and affects the direction of the resultant force. This is an exceptionally interesting question that will require a different data set in order to be answered.

COMMENT # 2.6

In equation 5, where is the contact force with the bed? Is it tied into the gravitational force terms? The critical drag and lift forces will depend on the submerged weight of the particle and the orientation of the grain within a pocket through the contact reaction force. The orientation of this force will also influence where it is more likely to have lift or drag dominate. How does it factor into equations 9 and 10? It should, shouldn't it? And 5. Along with the preceding question, how is this contact force for accounted for throughout the range of a particle's motion as it moves and interacts with the bed?

Along with the preceding question, how is this contact force for accounted for throughout the range of a particle's motion as it moves and interacts with the bed?

In addition to our answer in the interactive discussion, we now clarify that the interaction $f$ (denoted as $F_r$ in the previous manuscript) is the interaction between the hydraulic forces and the resistance or reaction forces. We realise that the previous manuscript gave the impression that we isolate somehow the hydraulic force but this is not possible to be done with and IMU (Comment 1.14 and revised section 2 in the manuscript).

COMMENT # 2.7

Please provide a figure showing the experimental setup with flow depth and bed arrangement. I would think that $\tau_b$ has little meaning in terms of mobilization with your particular laboratory setup.

We provide a detailed description of the experimental setup in the revised manuscript (Comment 1.7). We also removed all the $\tau_b$ calculations from the manuscript.

COMMENT # 2.8

Terminology in Figure 2 and elsewhere. When do vibrations turn into motion? Does entrainment start when the particle reaches a distanced travelled of 1 diameter, or does entrainment start when the particle starts to move and then continues on a path that leads to it moving 1 particle diameter? Also, what is a non-entrainment event with a measured inertial impulse? Does that correspond to a case where the particle started to move out of the pocket but then fell back down before reaching the apex?

We explain in the begining of this document that Figure 2 included a mistake which could confuse the reader. We define entrainment at the point of full dislodgment (Comment 1.17). As a result, during a vibration period the particle moves in its pocket but is not fully dislodged, as suggested by the Reviewer.

COMMENT # 2.9

In the discussion, I'd suggest non-dimensionalizing the force (maybe using sub- merged specific weight) when making the comparisons to other work. Also, how do your inertial forces compare to standard drag estimates using velocity, particle size, and a drag coefficient? What types of relative fluid/particle velocities are needed?

We now provide a non-dimensonalised force values in the discussion (Section 5.1, Lines 395-410). The comparison of inertial forces with standard drag estimates is more difficult because most of the literature is devoted to fully developed flow which is not the case for our experimental setting (small bed of hemispheres and grain diameter/ depth ratios close to 1).

COMMENT # 2.10   PRESENTATION QUESTIONS/ CONCERNS/ SUGGESTION FROM RE-
VIEWER 2

The paper is reasonably well written, but I do have some suggestions below that I think would help to improve the presentation of the work.
Abstract:

- L.1 Delete "been"

- L.4 Replace "on sediment" with "on a particle during"

- L.5-7 The sentence "Today, twenty years....  for the issue" is not needed in an abstract. Suggest deleting it

- L.9 Change "grains on" to "grain moving on"

- L.11 Change "resulting to the" to "resulting in a"

All those comments have been accepted and addressed. See attached track changes manuscript.

Comment: Introduction:

- overall I think you can shorten down the introduction

- L.38-41 I think you can remove the sentence that starts with "The term Lagrangian..." Most people know the difference between a Lagrangian and an Eulerian frame of reference

As we said in the online discussion, we tried to keep the balance between being concise and explaining all the terminology to avoid confusion.

- L.44 Delete "(the exact definition of turbulence impulse)"

- L.72 Change opening sentence to: MEMS-IMU sensors ideally measure forces at the center of mass....

- L.74 change to "acting on the grain as it moves"

- L.83 suggest changing to, "...and electrical engineering. Modeling of IMU error..."

- L.91 and 92 suggest changing resolve and resolution to "map" and "mapping" or "rotate" and "rotation"
Frames of ref

- L.123 Suggest, "... frames is non-trivial. A widely-used method..."

- L.124 Suggest changing "to apply" to "the application of"

All those comments have been addressed as suggested by the Reviewer.

Comment: L.129-end of the section. I'm torn here. I could see all of this being better suited for the appendix if the focus of the paper is on the data from the particles. However, if you want the paper to be about the mapping between the frames of reference then you should keep it here.

We followed the Reviewers advice and moved all of the tranformations to appendices

- L.157-158 Delete the beginning of the sentence that spans the text "As sediment" to "linear acceleration". Just start with ar.

Inertial measurements

- L.211 change to Ti

- L.216 suggest changing "that mobilizes the particle" to "acting on the particle once it starts to move" Lab and field experiments

- L.238 0.028 l/s is a discharge, not a rate of increase in discharge

- L.240 change "recorded" to "video of"

- Figure 4. If I'm correct, I think you reference figure 4 before referencing figure 3. It should be the other way around

- L.254 suggesting changing to, "...inertial impulses for cases were the grain started to move." Discussion

- L. 360 - I'd suggest making the extended analysis part of the discussion. Use subsection for the different components of the discussion such as comparison with other work and L.368 - delete one of the "of"s

All the comments have been addressed according to Reviewer's suggestions.

**Reviewer # 3**

**Comment # 3.1**

This manuscript is intended to quantify the dynamics of entrained sediment particles by using advanced inertial force monitoring systems. I believe this work has good contributions to this long-standing problem and it is worth being published. Overall, the materials are properly organized and relevant concepts explained concisely. I see no significant problems for its publication, but there are several minor suggestions for the authors to consider in their revision. Below, I listed my comments and suggestions as I read through the manuscript.

We want to thank the reviewer for the thorough review and the supportive comments. We did address all the suggestions in the revised manuscript as will hopefully become clear in the following responses.

**Comment # 3.2**

Regarding the Lagrangian approach, Ballio et al. (2018, "Lagrangian and Eulerian description of bed load transport") provide a unified framework to describe sediment particle motion under different frames of reference and quantification methods. The authors may want to refer to this work and highlight their particular contributions to the topic.

As we mention in the online discussion we consider this to be an important aspect of this work. The following paragraph in the discussion (section 5.2) discusses the topic and the work of Ballio et al. 2018

"This work uses a theoretical framework which has the potential to enhance the mathematical modelling of sediment transport. The Newton-Euler model of section 2, in conjunction with the quaternion transformations of Appendix A, can be read as a 3D and unrestricted Lagrangian - Eulerian model for sediment transport. In our analysis, particle dynamics are transformed from a Lagrangian domain (and the mobile body frame of the particle b) to a static Eulerian domain (frame r) which is most commonly used for the analysis of turbulent flow. Ballio et al. (2018) analyse the topic in detail and provide a comprehensive 1-D Lagrangian - Eulerian model which also accounts for the intermittency of sediment transport using a binary classification of mobile and non-mobile states. Our presentation can be used to define 3D Lagrangian dynamics, including rotation, in full and then to transform the corresponding kinematic properties to the Eulerian domain for direct comparison with the turbulent forces. We acknowledge that the verification of the 3D Lagrangian - Eulerian model is heavily dependent on the inertial measurements, and particularly the constant tracking of

relative orientation between the frames. However, it is possible to predict that future calibration experiments deploying IMUs will be used this way to parametrise simulations."

COMMENT # 3.3

L58: The issue of over-prediction of transport rates was also extendedly discussed by Bunte et al. (2004, "Measurement of coarse gravel and cobble transport using portable bedload traps") and Singh et al. (2009, "Experimental evidence for statistical scaling and intermittency in sediment transport rates"), and other related papers. The reference can be updated with these contributions.

We accepted the comments and added the references in this sentence (Lines 55-56 in the revised manuscript).

COMMENT # 3.4

Sec. 2 (Frames of reference, rotations and IMU measurements) provides details of mathematical expressions for the conversion/transformation among different frames of reference. But Fig. 1 needs some more clarity on the specifics of these frames, e.g. (x,y,z) (rx,ry,rz),(ix,iy,iz).

The figure has been revised to explain the experimental regime better (Comment 1.7). In addition, we provide an extra figure in Appendix A, dedicated to the frames of reference in the context of inertial dynamics.

COMMENT # 3.5

Based on their formulation, Eq. (9) only concerns the threshold condition in the tractive mode towards downstream, and Eq. (10) the upward movement. Yet, when approaching the strictly critical condition, particle rolling, which has even lower resistance, can be the most predominant entrainment mode. The authors can consider adding an extra formula describing the rolling threshold for providing a more complete framework.

We introduce the threshold of motion in the revised section 2. We also explain the differences in the magnitude between the tangential force applied on the surface of the particle (responsible for rotation) and the linear forces applied on its centre of mass. Finally we explain why the rolling of rigid body isn't generally the same with the rolling mode of entrainment (or rollover). From the manuscript:

"The torques are analysed in the body frame of the particle (b) and a non-directional description of the rotation threshold is given by the norm of angular accelerations ($\|\vec{\alpha_b}\|$) exceeding 0. (..) the critical condition for particle rotation is given by:

$$T_{net} = I_{cm}\|\vec{\alpha_b}\| = I_{cm}\|\frac{d\vec{\omega_b}}{dt}\| = f_{rot}\ R \geqslant 0$$

(..) Using the above equation, it is possible to estimate the magnitude of the tangential component $f_{rot}$ (Appendix C). The calculation reveals that, for the scale of the particles discussed here, the effect of the tangential force, in terms of force magnitude applied on the particle, is negligible in comparison to the linear forces. For this reason, we will only focus on the linear net force and the interaction $\vec{f}$ which we calculate as the difference between the net force and the gravitational components. However, we demonstrate the scale difference between the rotational and the linear components of particle motion in Figure 3 and Appendix C.

(..) the total kinetic energy can by calculated from these velocities as $K = \frac{1}{2}m\|v_r\|^2 + \frac{1}{2}I_{cm}\|\omega_b\|^2$ with $\frac{1}{2}m\|v_r\|^2$ being the translational and $\frac{1}{2}I_{cm}\|\omega_b\|^2$ the rotational component. The condition $K = 0$ represents the threshold of rolling for any rigid body. However, this is not a direct equivalent to the typical rolling mode of entrainment because of the differences in the definition of rotation (spinning vs orbital) and the fact that here there is no assumption about the slipping condition as K can be used to describe a rolling with slipping motion"

COMMENT # 3.6

L234. Has the 1.5 mm uniform sand glued to surface or also movable?

No, the bed was washed before the experiments and the sand was static. Also there was no sediment transport from the upstream coarser section (Comment 1.7)

COMMENT # 3.7

Fig.2 (b) describes the change of drag forces during the noted five entrainment events. The pattern shown here, however, somewhat contradicts the impulse model mentioned later in the manuscript (e.g. L278-279). If all these five events follow exactly the impulse criterion, the events of higher magnitude should persist for a shorter duration, and vice versa, to maintain at the same impulse level. In other words, these events, very likely, represent different degrees of particle mobility and not the most extreme cases of entrainment (minimal critical impulse). I think this aspect is worth mentioning in the discussion.

This comment reflects both the confusion that could cause the previous version of Figure 3 (analysed in the first section of this document) and the fact that the IMU

measurement needed to be discussed more. The Reviewer is correct: there are different degrees of mobility represented which are "masked" because in the previous manuscript we added the forces from two directions. In the revised manuscript we perform a vector calculation, which should clarify how the particle behaves and how the minimal impulse is captured (see first section of this document and revised Figure 3).

COMMENT # 3.8

The calculation of entrainment probability is not clear to me. Is it described as the ratio of entrainment duration to the total observation time, or the ratio of entrainment events to the total exceedance events observed? I assume it is the latter definition. Adding an equation will help to clarify this point.

The latter is the case (entrainment / exceedance events) but instead of using the ratio we perform a logistic regression (as also used in Maniatis et al., 2017). We believe that with the clear definition of entrainment and applied forces, we can avoid adding one more equation to this manuscript.

COMMENT # 3.9

L300. The scaling effects are usually attributed to the intermittency of particle motion dictated by turbulent flow structures. Yet, this sentence seems to suggest the role of physical scale between the laboratory flume and field stream. A clarification will be helpful (see discussion in Singh et al., 2009).

We intended to refer to the differences in turbulence (non-fully developed flow in the lab, developed (but still shallow) flow in the field). We clarify that in the discussion of the manuscript and in Comment 1.9.

COMMENT # 3.10

Comment: Fig. 4(a) can be improved by using different shading colors for FD and FL, respectively.

We did follow the Reviewer's advice.

The differences in magnitude of critical drag and lift forces between this work and literature data are attributed to the particle sizes, or corresponding mass, used differently. To resolve this issue, a dimensionless quantity, e.g. FD/particle weight, FL/particle weight, can be considered for both the present work and previous reports.

This point was raised by Reviewer 2 and we now provide the dimensionless quantities in the discussion (Section 5.1)

L325-332. The description of the negative lift forces can be more precise in relating to the threshold of motion conditions. Specifically, when the negative lift force appears, the probability of particle lifting reduces, and also, the resistance to the tractive movement increases via the enhanced intersurface friction.

We considered this comment, but since the interaction f in our work is not decoupled (it is the interaction between hydraulics, contact forces and friction) we decided to avoid a reference to friction because it could mislead the reader. We will hopefully address this comment in a future work.

Comment: L353-357. The description of the entrainment mode of rolling can be placed in the earlier section (see the previous comment) to avoid confusion.

We address this comment in Comment 3.5

In the rest of this document we present a fully annotated version of the new manuscript with all the changes highlighted as well as the output from the Acrobat Reader Compare Files tool. We used the tool to capture figure insertions and other stylistic changes that are not always captured with latexdiff (or they cannot be complied easily). The vast majority of the annotations must be visible in any pdf reader, but if the editors encounter problems reading the document please feel free to contact GM.

**Compare Results**

| Old File: | | New File: |
|---|---|---|
| **template.pdf** | versus | **template_f.pdf** |
| **42 pages (1.53 MB)** | | **49 pages (2.65 MB)** |
| 23/07/2020, 19:13:13 | | 23/07/2020, 19:19:59 |

**Total Changes**

**1334**

**Content**

417  Replacements
379  Insertions
246  Deletions

**Styling and Annotations**

75   Styling
217  Annotations

Go to First Change (page 1)

[revised manuscript text omitted]

---

## Author Response (AR2)

Authors' response (Reviewers' comments in blue)

We sincerely thank all the reviewers for the thorough and extremely useful reviews as well as the positive comments.

**Reviewer 1**

The manuscript by Maniatis et al. is greatly improved from the first version. I mostly have minor suggestions to clarify various points, including how the authors describe and define entrainment. I recommend publication after minor revisions.

Thank you very much for your time and effort.

7-8: "This paper introduces and tests a theoretical framework that connects the IMU measurements with the forces applied on a sediment grain moving on a riverbed". To me this sentence still implies that the authors independently "test" their calculations of forces, which they do not. Suggest saying something like they develop a theoretical framework for calculating drag and lift forces on grains that are in motion, based on IMU measurements.
Replaced with:

 "This paper develops a theoretical framework for calculating drag and lift forces on grains based on IMU measurements".

42: remove "and"

Done

58: remove comma after Gimbert et al citation.

Done

116: suggest changing to "… record zero acceleration (and therefore zero net force) until…"
Done

118: Could cite Garcia et al (2007) here for grain vibrations prior to entrainment.

Garcia, C., H. Cohen, I. Reid, A. Rovira, X. U´beda, and J. B. Laronne (2007), Processes of initiation of motion leading to bedload transport in gravel-bed rivers, Geophys. Res. Lett., 34, L06403, doi:10.1029/2006GL028865.

We thank the reviewer for the excellent suggestion, inserted.

142: remove comma after "equations 1 and 2". Next line, change e.g to e.g.

Done

169: suggest changing "completely immobile" to "not moving".

Done

Equation 12: An equation without a relationship operator just feels wrong to me. Add a variable for impulse and an equals sign.

That was a typo, corrected.

223: YEI capitalized, but not capitalized in line 232, be consistent.

Corrected

267-269: The definition of "entrainment point" is unclear. To me, "the initiation of the vibration that dislodges the particle by one particle diameter" unambiguously says that entrainment is defined as the moment the particle very first starts vibrating, if later on it moves by one particle diameter before stopping to vibrate. But I don't think that is what the authors actually mean, because Figure 3 shows "dislodgement" well after the initiation of vibrations, and line 285 says that there were pre- and post-entrainment vibrations (the actual definition given would mean zero vibrations before entrainment).

I am not sure, but my guess is that the authors actually define entrainment as the time when the particle has moved by one particle diameter, as implied in lines 281 and 284. If the time of entrainment is measured entirely by video, say this in the paragraph starting at line 267, because otherwise it is not clear how you know when the particle has dislodged by one b-axis. I also would remove "independently" from line 281, because it implies that entrainment was also measured in some other way, but I don't think this is true. I think entrainment time was measured from the video, and then the forces and impulses before and after this time were calculated from the IMU data.

The reviewer's observations are correct
For clarification we replaced the text in lines 266-267 with:
Line 267: 'We define entrainment as when the particle moves by one particle diameter, or b-axis length for the ellipsoid. Having identified the time when the grain has moved by this distance from the video, the timing of the vibrations which directly and continuously preceded entrainment was also determined from the video. Many periods of vibration which do not lead to entrainment were recorded by the sensor and are visible on the videos.'

Figure 3 instead uses the term dislodgement. Is this different from entrainment? Be consistent, if dislodgement is the same then use the same term; if different then define dislodgement.

Changed to Entrainment

The authors should also expand on why they define the threshold of motion as the probability of entrainment > 0.5.

Line 323 Text added:
The threshold probability of 0.5 corresponds to the impulse at which the probability changes from the particle being more likely to be at rest, to being more likely to be entrained. In this context, with this approximation we calculate a gradational threshold of entrainment (Begin and Schumm, 1979) and not an absolute one.

Done

Done

315: After reading the manuscript, I have a slightly better but still incomplete understanding of why the authors say that the ellipsoid sensor demonstrates a strong influence of lift forces. The issue is that lift forces are smaller, so it seems like they should be less important. And they're also less well correlated for I vs t. When I first read this I was just confused. I suppose lines 325-327 are the explanation. Maybe the point that I'm missing that the data presented only reflect timeseries that did lead to entrainment? Also, did the authors test whether drag and lift time series are correlated in their actual data? The figure 7 analysis assumes they are independent, but its not clear that they are independent in the actual data. Apologies if it is in the manuscript and I missed it.

Response

It is not possible to do this comparison for the actual data (point forces), because impulses need to be defined for specific time frames. In this presentation this is the time frame where the sensor was recording net forces > 0.  During those periods (over the threshold of motion) the lift forces are can be smaller in magnitude, but they are also much more frequent. This is the observation on which we base our interpretation of the influence of the lift forces on the ellipsoid.

To clarify further to the above, we added the following sentence it the captions of Figure 4 and 6. "Impulses of all the inertial forces that exceeded the gravity forces", to clarify further that the values correspond to over 0 net force impulses.

In addition, the drag and lift forces are uncorrelated as indicated in Appendix E.  This gives us some confidence for the calculation presented in Figure 5, showing that the lift threshold is lower than the drag for the ellipsoid, supporting further the control of lift impulses. To highlight this more we added a sentence in lines 452-454 which now read as

"After the normalisation, the laboratory and field results are combined into one dataset of normalised drag ($I_{Drag}$) and normalised lift ($I_{Lift}$) impulses, which are assumed to be uncorrelated. The latter is justified by the fact that the point lift and drag forces are statistically uncorrected as shown in Appendix E (Figure E2"

Figure 4a: seems like the y axis should be expanded to show the data over more than 10% of the plot area.

Figure 4a shows all the periods that net force is> 0

387: runon sentence, particles not articles. Change to "…diameter particles. The measured…"

Done

390: there is a bonus "s".
Corrected

392: expand just a bit on fully developed flows. I think you mean something like "…flows much deeper than the particle diameters, with fully developed boundary layers."
Done

413: Are the numbers flipped?
Corrected.

416: I presume instantenuous is another physics term I'm unfamiliar with? Trying to understand the physics in this paper, I sometimes have an immediate feeling of "huh?" … instantenuous. But seriously, this is my favorite new word portmanteau. And I really do like and appreciate the manuscript!

That was a typo, we mean instantaneous. We really appreciate the positive comments!

432-435: break into two sentences.

Done

**Reviewer 3**

The authors have done an excellent job responding to the questions and updating the manuscript. The changes made are all in the right direction, and I think that the paper can be published with only minor revisions. My remaining comments are given below.

Thank you very much for your time and effort.

(1) With respect to the flume setup, around L255 and follows makes me think that water can flow under the ellipsoid particle whereas water did not really flow under the sphere. If water could indeed flow under the ellipsoid particle, then I expect this had something to do with the dominance of lift. Please be clear about the exact way in which the two test particles sat in the flume with respect to the surrounding sediment.

Response:
The sphere and the ellipsoid were at comparable elevations in the bed because the gaps between the hemispheres were covered by the sand coating as we explain in the text (255-257) The protrusion of ellipsoid is approximately equal to its c-axis. (0.03, D1)  as it is fully supported by the hemispheres. This makes the protrusion of the ellipsoid smaller than the one of the sphere. Despite the ellipsoid being at full contact with the hemispheres, there is a possibility that a film of water could flow between the coating/ gap filling gravel and the particle. Because, we observe the same control of the lift force in the field experiment (where the ellipsoid was placed on bare bedrock) we think that ,for this degree of protrusion, this film of water wouldn't have a very strong effect. However, we cannot quantify that, and we added a comment in the legend of Figure 2 so the readers are aware of this issue.

"While the spherical particle was in full contact with the bed (hemispheres and coating/filling gravel), the setting could result to a film of water flowing underneath the ellipsoid. "

(2) Along with comment 1, please include the approximate height of the ellipsoid particle in Fig 2 as you have done for the sphere.

Added: Line 257

(3) Are the critical depths in Fig 2 measured or estimated based on Shields? If they are estimated based on Shields, please put the actual measured depths in Fig 2 rather than the calculated critical depths.

The depths are measured at the point of entrainment and all the hydraulics are calculated for the same conditions. This is now clarified in Figure 2 and table D1. We also replaced critical depth with measured depth in lines 275-277.

(4) I expect that the relative importance of lift vs drag on a particle has as much to do with how the particle sits in the bed then it does with particle shape. Wouldn't you expect that lift is more important in mobilizing a sphere if the top of the particle of interest is down near the top of the surrounding bed particles rather than being perched and exposed up on the bed top? Could you argue that the importance of lift for the ellipsoid has to do with protrusion as much as it does particle shape? If so, please discuss, or at least comment on protrusion and its link to your test particles and outcomes.

We have now clarified that the sphere and the ellipsoid are in comparable elevations in the bed. We completely agree that the lift forces will exert a much higher control for lower protrusion values, but our settings (in both the flume laboratory and the field) correspond to highly exposed to the flow particles. In this context, we added a comment in the discussion:

Line 379:

"In addition, it is useful to note that grain protrusion is not discussed in this work, despite being an important control on grain motion and particularly entrainment (e.g. Dey and Ali, 2018), since the presented laboratory and field experiments only correspond to particles that are highly exposed to the turbulent flow."

**Reviewer 2**

I served as one of the reviewers of this manuscript at its first submission. The authors has considered my suggestions seriously and applied them to this revision. After major improvements has been made, I think this manuscript is ready for publication in the Earth Surface Dynamics.

Thank you very much for your time and effort.

---

## Author Response (AR3)

Author's response to technical corrections 7/11/2020

- We addressed all the technical corrections from the associated editor apart from the change of colours in Figure 5 since the colour matching is not intended.

More specifically

Throughout: Please capitalize "Time" in all figure labels.
**Done**

Figure 4 - Bring labels of "event duration" and "impulse" inside the graphs (top left corner) for readability; remove grey box behind legend on last panels.
**Done**

Figure 5 – please add the red and blue vertical lines to the Figure legend. I'd recommend also changing the blue to better match the other values (if the intention is to match colors?)

**Lines added in the legend, colour matching is not intended.**

Figure 6 – Bring labels of "event duration" and "impulse" inside the graphs (top left corner) for readability; remove grey box behind legend on last panel.

**- Done**

Figure 7 – Colors are difficult to see – especially the RHS zone in light blue – I suggest adding an inset legend and increasing the stroke of the red dashed lines. The "thres" superscripts a bit confusing here – I would instead just add the word threshold behind your variable names.

**- RHS colours changed, stroke increased, superscripts added behind the variables. Also legend added**